

# 1 ForestScan: a unique multiscale dataset of tropical forest structure across 3
# 2 continents including terrestrial, UAV and airborne LiDAR and in-situ forest census
# 3 data

Cecilia Chavana-Bryant[1,2], Phil Wilkes[1,2,24], Wanxin Yang[1,2], Andrew Burt[3], Peter Vines[19], Amy C. Bennett[4], Georgia C.
Pickavance[4], Declan L. M. Cooper[1,25], Simon L. Lewis[1,4], Oliver L. Phillips[4], Benjamin Brede[5], Alvaro Lau[11], Martin Herold[5],
Iain M. McNicol[6], Edward T.A. Mitchard[6,18], David A. Coomes[8], Toby D. Jackson[8], Löic Makaga[9], Heddy O. Milamizokou
Napo[9], Alfred Ngomanda[15], Stephan Ntie[9], Vincent Medjibe[9], Pacôme Dimbonda[9], Luna Soenens[10], Virginie Daelemans[23],
Laetitia Proux[13], Reuben Nilus[12], Nicolas Labrière[20], Kathryn Jeffery[14], David F.R.P. Burslem[21], Dan Clewley[16], David
Moffat[16], Lan Qie[22], Harm Bartholomeus[11], Gregoire Vincent[7], Nicolas Barbier[7], Geraldine Derroire[13], Katharine
Abernethy[14,15], Klaus Scipal[17] and Mathias Disney[1,2]
[1]Department of Geography, University College London, London, WC1E 6BT, UK
[2]NERC National Centre for Earth Observation, UCL Geography, London, WC1E 6BT, UK
[3]Sylvera Ltd., London, EC1Y 4TW, UK
[4]School of Geography, University of Leeds, Leeds, LS2 9JT, UK
[5]Section 1.4 Remote Sensing and Geoinformatics, GFZ Helmholtz Centre for Geosciences, Potsdam, 14473, DE
[6]School of GeoSciences, University of Edinburgh, Edinburgh, EH9 3JN, UK
[7]AMAP, Univ. Montpellier, CIRAD, CNRS, INRAE, IRD, Montpellier, 34398, FR
[8]Plant Science and Cambridge Conservation Initiative, University of Cambridge, Cambridge, CB2 3QZ, UK
[9]Agence Nationale des Parcs Nationaux (ANPN), PO Box 20379, Libreville, GA
[10]Q-ForestLab, Department of Environment, Ghent University, Coupure Links 653, B-9000, Ghent, BE
[11]Laboratory of Geo-Information Science and Remote Sensing, Wageningen University & Research, 6708 PB Wageningen,
NL
[12]Forest Research Centre, Sabah Forestry Department, P.O. Box 1407, Sabah, 90715, MY
[13]CIRAD, UMR EcoFoG (AgroParistech, CNRS, INRAE, Université des Antilles, Université de Guyane), Campus
Agronomique, Kourou, 20040, FG
[14]Faculty of Natural Sciences, University of Stirling, Stirling, FK9 4LA, UK
[15]Institut de Recherche en Ecologie Tropicale, IRET/CENAREST, Libreville, PO Box 13354, GA
[16]Plymouth Marine Laboratory, Plymouth, PL1 3DH, UK
[17]ESA Centre for Earth Observation (ESA-ESRIN), Frascati, 00044, IT
[18]Space Intelligence Ltd. 93 George Street, Edinburgh, EH2 3ES, UK
[19]8 Havelock Terrace, Plymouth, PL2 1AT, UK
[20]Centre de Recherche sur la Biodiversité et l'Environnement (CRBE), UMR 5300 CNRS-IRD-INP-UT3, Toulouse, 31062
cedex 9, FR
[21]School of Biological Sciences, University of Aberdeen, Aberdeen, AB24 3UU, UK
[22]College of Health and Science, Department of Life Sciences, University of Lincoln, Lincoln, LN6 7TS, UK
[23]Gembloux Agro-Bio Tech Liège University, Passage des déportés 2, B-5030 Gembloux, BE
[24]Kew Wakehurst, Ardingly, West Sussex, RH17 6TN, UK
[25]Centre for Biodiversity and Environment Research, Department of Genetics, Evolution and Environment, University College
London, London, WC1E 6BT, UK
*Correspondence to*: Dr Cecilia Chavana-Bryant (c.chavana-bryant@ucl.ac.uk)



## Abstract

The ForestScan project was conceived to evaluate new technologies for characterising forest structure and biomass at Forest Biomass Reference Measurement Sites (FBRMS). It is closely aligned with other international initiatives, particularly the Committee on Earth Observation Satellites (CEOS) Working Group on Calibration & Validation (WGCV) AGB cal/val protocols, and is part of GEO-TREES, an international consortium dedicated to establishing a global network of Forest Biomass Reference Measurement Sites (FBRMS) to support EO and encourage investment in relevant field-based observations and science. ForestScan is the first demonstration of what can be achieved more broadly under GEO-TREES, which would significantly expand and enhance the use of EO-derived AGB estimates.

We present data from the ForestScan project, a unique multiscale dataset of tropical forest 3D structural measurements, including terrestrial LiDAR scanning (TLS), unmanned aerial vehicle LiDAR scanning (UAV-LS), airborne LiDAR scanning (ALS), and in-situ tree census and ancillary data. These data are critical for the calibration and validation of earth observation (EO) estimates of forest biomass, as well as providing broader insights into tropical forest structure.

Data are presented for three FBRMS: FBRMS-01: Paracou, French Guiana; FBRMS-02: Lopé, Gabon; and FBRMS-03: Kabili-Sepilok, Malaysia. Field data for each site include new 3D LiDAR measurements combined with plot tree census and ancillary data, at a multi-hectare scale. Not all data types were collected at all sites, reflecting the practical challenges of field data collection. We also provide detailed data collection protocols and recommendations for TLS, UAV-LS, and plot census measurements for each site, along with requirements for ancillary data to enable integration with ALS data (where possible) and upscaling to EO estimates. We outline the requirements and challenges for field data collection for each data type and discuss the practical considerations for establishing new FBRMS or upgrading existing sites to FBRMS standard, including insights into the associated costs and benefits.

## 1. Introduction

Our capability to estimate forest structure and above-ground biomass (AGB) has rapidly advanced, leveraging new remote sensing observations from ground, air, and space. This progress underscores the importance of quantifying and understanding terrestrial carbon sources and sinks, the response of global forests to climate change, and conservation and restoration efforts at local to global scales. These new measurements broadly fall into the following categories:

1) Terrestrial laser scanning (TLS) provides highly detailed (centimetre-scale) 3D structural measurements across hectare scales, enabling non-destructive AGB estimates that are independent of, yet complementary to, empirical allometric model estimates (e.g. Calders et al., 2022; Demol et al., 2024).





2)  Unmanned aerial vehicle laser scanning (UAV-LS) has evolved from highly specialised and expensive surveying platforms to more operational, low-cost systems that offer coverage of several to thousands of hectares, with hundreds to thousands of points per square metre from above. These data can be used to estimate forest canopy height, basal area, tree crown size and shape, vertical structure, and AGB via allometric model functions of tree properties, including height, diameter at breast height (DBH), and crown shape (Brede et al., 2022a; Kellner et al., 2019) However, as UAV-LS systems proliferate, the need for intercalibration between sensors increases, due to differences in scanner and laser properties such as power, wavelength, divergence, and scan rate, which result in notable variations in penetration and object detection rates (Vincent et al., 2023).

3)  Airborne laser scanning (ALS) has been a well-established tool in forestry and forest ecology since the 1990s. ALS is routinely used to estimate forest height, structure, and AGB at stand level via empirical models and at regional to national scales via allometric models (Duncanson et al., 2019; Jucker et al., 2017).

4)  Spaceborne Light Detection and Ranging (LiDAR) (e.g. GEDI, ICESat, and ICESat-2) can provide estimates of forest height in non-continuous footprints of tens to hundreds of metres, underpinning most large-scale AGB maps, particularly in the lowland tropics (Avitabile et al., 2011; Avitabile et al., 2016; Saatchi et al., 2011). Various satellite missions have also provided empirical evidence for correlations between the radar signal and AGB for AGB < 250 Mg ha$^{-1}$ (Askne and Santoro, 2012), but the ESA BIOMASS mission, scheduled for launch in 2025, is the only mission specifically targeting higher biomass tropical forests (Quegan et al., 2019; Ramachandran et al., 2023).

The current challenge is to consistently make and process plot-based measurements in support of EO-derived AGB, combine them, integrate them with long-term ground-based inventory approaches, and optimally use them with EO data. There is increasing recognition that the value of large-scale EO approaches to assessing AGB and forest structure largely depends on robust calibration and validation data (Duncanson et al., 2019; Nature Editorial, 2022; Ochiai et al., 2023). This knowledge and capability gap have led to calls for concerted international funding and coordination to establish long-term Forest Biomass Reference Measurement Sites (FBRMS), with a particular focus on tropical forests (Labrière et al., 2023; Schepaschenko et al., 2019).

Here, we present a new dataset from the European Space Agency (ESA) funded ForestScan project, which contributes to this aim and provides access to data from the first three FBRMS of the GEO-TREES network. The project has collected data, including TLS, UAV-LS, ALS, and census data, covering three FBRMS across the tropics. We describe these data, related data collection and processing protocols and tools, and make brief recommendations for future data collection for FBRMS.



## 2. Methodology

### 2.1 ForestScan Forest Biomass Reference Measurement Sites (FBRMS)

Three Forest Biomass Research Monitoring Sites (FBRMS) were selected based on discussions among the team, the European Space Agency (ESA), external collaborators, and various criteria, including the availability of well-established plots, the representativity of tropical forest types and climates, established collaborations, agreements and logistical support with in-country partners, and the availability of previously collected data, particularly census data, as well as Airborne Laser Scanning (ALS) and Terrestrial Laser Scanning (TLS) data. The chosen sites were:

- FBRMS-01: Paracou Research Station, French Guiana
- FBRMS-02: Station d'Etudes des Gorilles et Chimpanzés, Lopé National Park, Gabon
- FBRMS-03: Kabili-Sepilok, Malaysian Borneo

Earth System
Open Access Science
Discussions Data

**FBRMS-01: Paracou Research Station, French Guiana**

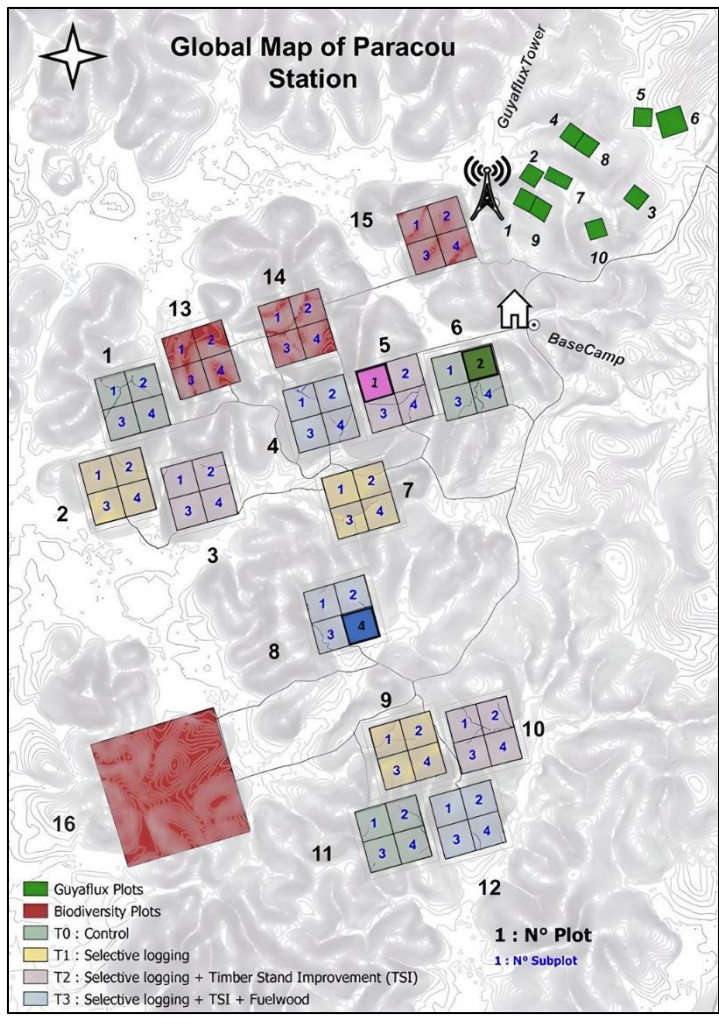

**Figure 1:** Map of FBRMS-01: Paracou Research Station, French Guiana (image: Laetitia Proux, UMR EcoFoG). The
location of ForestScan plots FG5c1 (pink), FG6c2 (green) and FG8c4 (blue) has been highlighted.
The Paracou research station is located near Sinnamary in the northern part of French Guiana, at a latitude of 5°18′N and a
longitude of 52°53′W. It is established on a long-term concession of the French National Centre for Space Studies (CNES)
and is managed by Cirad-UMR EcoFoG. The station experiences an equatorial climate characterised by two main climatic
periods: a well-marked dry season from mid-August to mid-November and a long rainy season, often interrupted by a short
drier period between March and April. The station receives approximately 3,000 mm of rainfall annually (mean annual
precipitation from 2004 to 2014: 3102 mm) and has a mean annual temperature of 25.7°C.



The core area of the Paracou research station (approximately 500 ha) is predominantly covered by lowland terra firme
rainforest. This old-growth forest has experienced no major human disturbance, although there are signs of pre-Columbian
activities. Species richness is high, with more than 750 woody species recorded, and 150-200 tree species per hectare with a
diameter at breast height (DBH) above 10 cm. A few dominant botanical families characterise the vegetation: Fabaceae,
Chrysobalanaceae, Lecythidaceae, Sapotaceae, and Burseraceae. The local heterogeneity of the floristic composition is mainly
driven by soil drainage. Aboveground biomass (AGB), measured on trees with a DBH ≥ 10 cm, ranges from 286.10 to 450
Mg/ha.
Following an initial large-scale inventory in the early 1980s, 12 permanent plots with an area of 6.25 ha each were established
in 1984. The positioning of the plot corners, perimeter, and inner trail (delimiting the four subplots) was verified about 10
years later by a professional land surveyor who confirmed the accuracy of the positioning. Initial tree positioning within plots
was done using two tape measures on perpendicular sides of subplots of 12.5 x 12.5 m at the time of plot establishment. Trees
recruited after that are positioned relative to the trees present at the time of plot establishment. Nine of the 12 permanent plots
were logged, with six receiving additional silvicultural treatment via one of three different treatment modalities between 1986
and 1988. This resulted in a disturbance gradient with a loss of AGB ranging from 18 to 25% for treatment 1, 40 to 52% for
treatment 2, and 48 to 58% for treatment 3. In the early 1990s, three new 6.25 ha plots and one 25 ha plot were established,
forming a total of about 120 ha of forest censused annually (undisturbed/control plots), every two years (disturbed plots), or
every five years (25 ha plot). All 6.25 ha permanent plots are subdivided into four subplots with relative tree coordinates
recorded within each subplot (see Fig. 1). Trees and palms with DBH ≥ 10 cm are mapped, identified, tagged with a field
number unique to their subplot, and periodically measured. This results in a large database covering more than 70,000 trees.
Understory woody vegetation (1-10 cm DBH) has been monitored on 64 subplots of 50 m² per plot (plots 1-12) since the early
1990s, and in a 9 ha permanent plot currently being established in plot 16. Since 2003, the station has had a 57 m flux tower
measuring greenhouse gas fluxes. An N, P, NP fertilisation experiment has been ongoing since 2015.
**FBRMS-02: Station d'Etudes des Gorilles et Chimpanzés, Lopé National Park, Gabon**

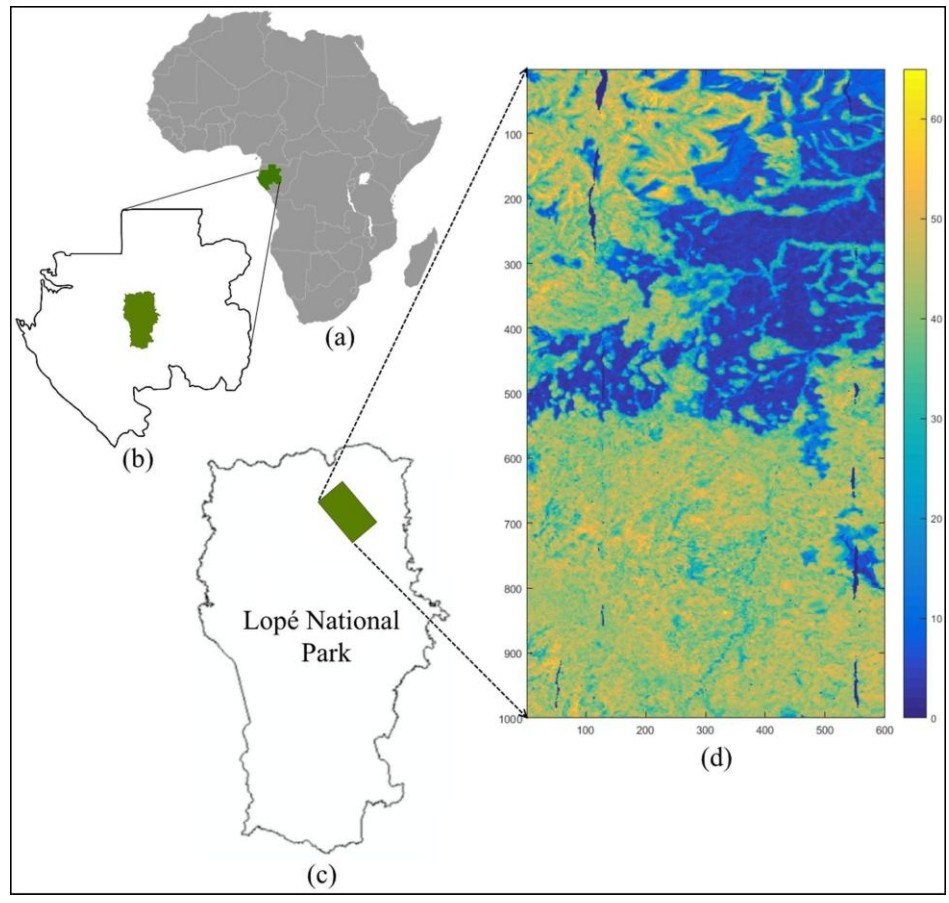


**Figure 2:** Location of FBRMS-02: Station d'Etudes des Gorilles et Chimpanzés, Lopé National Park, Gabon, and a EO-
derived map of forest canopy height across the savanna-forest mosaic. Reproduced under a Creative Commons licence from
Pourshamsi et al. (2021).

Lopé National Park is a 5000 km² protected area in central Gabon (coordinates: -0.5° latitude, 11.5° longitude; see Fig. 2),
comprising predominantly intact old-growth moist tropical forest. The northern part of the park features a savanna-forest
mosaic, an anthropogenically maintained remnant of the landscape from the Last Glacial Maximum. The broader landscape is
designated as a UNESCO World Heritage Site.

The transition from savanna to old-growth forest in the northern part of the park is characterised by six distinct forest types
(Cuni-Sanchez et al., 2016; White et al., 1995): (i) savanna, (ii) colonising forest, (iii) monodominant Okoume forest, (iv)
young Marantaceae forest, (v) mixed Marantaceae forest, and (vi) old-growth forest.

A substantial and varied body of literature has emerged from research conducted in Lopé National Park (Agence Nationale
Des Parcs Nationaux, 2025). More than 100 long-term censused forest plots have been established within the park, contributing
significant ground data for the calibration and validation of Earth Observation (EO) instruments (i.e. Duncanson et al., 2022;
Saatchi et al., 2019). These plots also support various other research activities, such as the Global Ecosystem Monitoring
(GEM) Network, an initiative aimed at understanding forest ecosystem functions and traits (Malhi et al., 2021).
**FBRMS-03: Kabili-Sepilok, Malaysian Borneo**




**Figure 3**: Map and location of the 36 x 1 ha forest plots established across the three distinct forest types found in FBRMS-
03: Kabili-Sepilok, Malaysian Borneo. Map adapted with permission from Sabah Forestry Department (Sabah Forestry
Department, n.d.) to show the location of ForestScan plots SEP-11 (Sandstone forest), SEP-12 (Alluvial forest) and SEP-30
(Kerangas forest).
The Kabili-Sepilok Forest Reserve is located on the Sandakan Peninsula in North-East Sabah, Malaysia, and encompasses
approximately 4,300 hectares of intact old-growth tropical forest. Sepilok has been protected since its establishment by the





Sabah Forest Department in 1931. The elevation ranges from 50 to 250 metres above sea level. This topographic variation,
combined with edaphic differences, results in three distinct forest types: (i) lowland mixed dipterocarp forest overlaying
alluvial soil in the valleys, (ii) sandstone hill forest on hillsides and crests, and (iii) lowland mixed dipterocarp and kerangas
forest at higher elevations (Sabah Forestry Department, n.d.).

Between 1995 and 2000, the Ecology Section of the Sabah Forestry Department established 36 one-hectare censused forest
stands across these forest types, as illustrated in Fig. 3.
**2.2 Data**
**2.2.1 Tree census**
Quality-controlled, tree-by-tree data on identity (tag number and species) and diameter size for all sampled plots in each of the
three FBRMS were collected using global standard tropical forest plot inventory protocols (Forestplots.Net et al., 2021). This
ensured a consistent, full species-level census for all plot trees with a diameter equal to or greater than 10 cm at each FBRMS.
Censuses provide tree-by-tree records that can potentially be linked to laser-scanning approaches. Species identity exerts
critical control on tree biomass via its strong influence on wood density. Laser-scanning techniques can provide excellent
measures of dimensions (e.g., height, volume) but require wood density estimates to convert tree volume into tree biomass
(see Fig. 4). Census data also provide tree-by-tree measurements of tree diameter and whole forest basal area. Finally, because
they are independent of constantly changing sensor technologies, when sustained over time, the core measurement protocols
in forest plots deliver long-term consistency for tracking forest biomass change, growth, mortality, demography, and their
trends over decades.

Census data for FBRMS plots in Gabon and Malaysia are available via ForestPlots.net (https://forestplots.net/, Forestplots.Net
et al., 2021; Lopez-Gonzalez et al., 2011). ForestPlots.net is an internet-based facility with functionality to support all aspects
of forest plot data management, including archiving, quality control, sharing, analysis, and data publishing via stable URLs
(DOIs). ForestPlots.net currently supports the data management needs of more than 2,000 contributors working with 7,000
plots across 23 participating tropical networks. Data access requires potential users to provide details of their planned use and
agreement to abide by requirements for the inclusion of all contributing researchers. This encourages maximum inclusivity of
data originators and is recognised as a key part of what is required to maintain long-term investment in people and infrastructure
that enables continued measurements in these areas (De Lima et al., 2022).

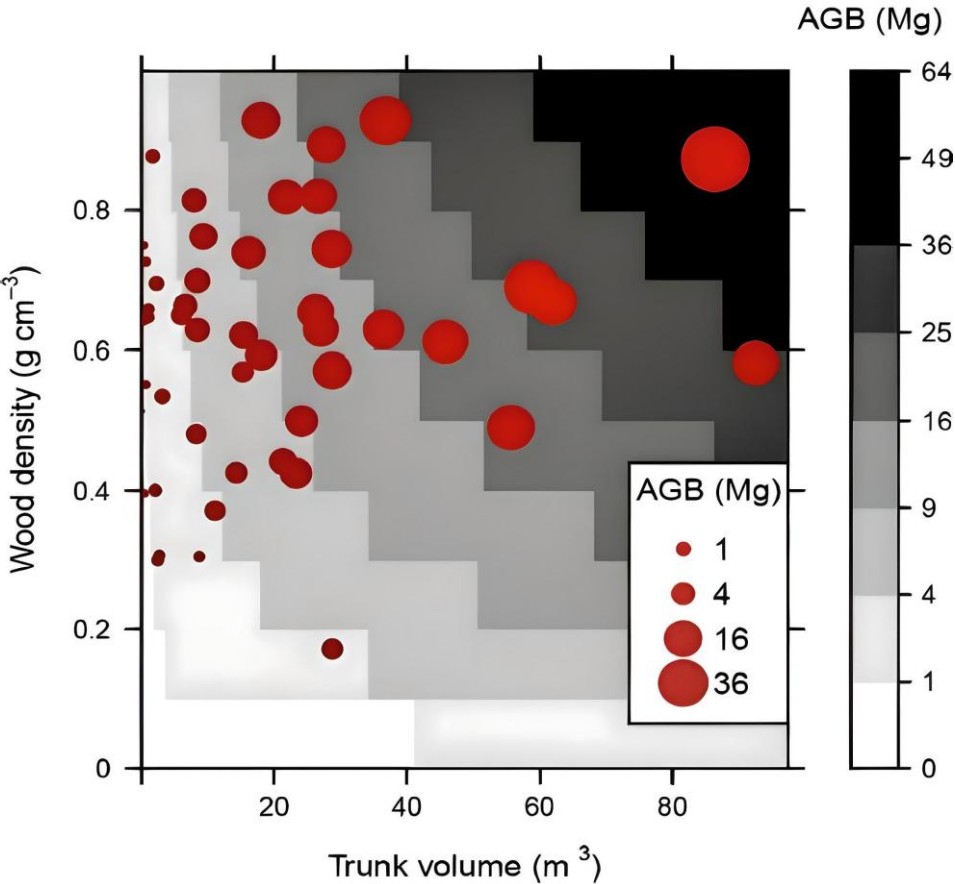

**Figure 4:** Wood density and volume independently control biomass, as shown by direct measurements of biomass of tropical forest trees. Red dots represent 51 trees destructively sampled and weighed in Amazonian Peru by Goodman *et al.* (2014). Dot areas are proportional to the actual, destructively measured aboveground biomass (AGB) of each tree, plotted against their trunk volume and directly measured wood density. Trunk volume was estimated as basal area multiplied by tree height. The grayscale background depicts a quasi-continuous allometric estimate of AGB for combinations of tree volume and wood density, using the Chave *et al.* (2014) allometric equation solved for each combination of diameter and wood density, and with tree height estimated using a three-parameter Weibull model fitted to all trees in the Goodman *et al.* (2014) dataset. Figure from Phillips *et al.* (2019).

**Tree census: FBRMS-01: Paracou, French Guiana**

In the Paracou FBRMS, tree censuses are conducted by two teams of three to five permanent field staff using Qfield on field tablets (since 2020, field computers were used prior to this). Tree girth is measured with a measuring tape at 1.3 m, except when buttresses necessitate a higher measurement point. The point of measurement (POM) is marked with paint to ensure the exact same point of measurement between censuses. POM and its potential changes are recorded. New recruits (i.e., trees that





have exceeded 10 cm DBH since the previous survey) are recorded and identified by vernacular names by the field team. Their
position is measured relative to initial trees. Dead trees and the cause of their death are recorded. Data are checked for errors
after field census using an R script. Any abnormal measurement (e.g., girth showing abnormal increase/decrease, missing
value) is then rechecked in the field in the weeks following the initial census.

Botanical identification campaigns are periodically carried out by one or two experienced botanists. When identification is not
possible in the field, samples are collected and examined at the herbarium of EcoFoG in Kourou or at the French Guiana IRD
herbarium in Cayenne. The plant classification system used is APG IV.

Plot descriptions for the Paracou FBRMS plots FG5c1, FG6c2 and FG8c4  are accessible via the Guyafor DataVerse
(https://dataverse.cirad.fr/dataset.xhtml?persistentId=doi:10.18167/DVN1/94XHID).
This internet-based data repository provides plot descriptions and datasets downloadable as CSV files, together with the
corresponding metadata, referenced by a DOI (Derroire et al., 2023). The ForestScan Project data package, including the latest
tree census data used in our analysis and collected in August 2023 for FBRMS plot FG5c1, in June 2023 for plot FG6c2, and
in        September        2023        for        plot        FG8c4,        is        accessible        via
https://dataverse.cirad.fr/dataset.xhtml?persistentId=doi:10.18167/DVN1/94XHID (Derroire et al., 2025).
**Tree census: FBRMS-02: Lopé, Gabon**
In the Lopé FBRMS, tree census data was collected at 12 plots in 2017 for the ESA AfriSAR campaign. During June - July
2022, these 12 plots plus one additional 1 ha plot (LPG-02) were re-censused, making a total of 10 x 1 ha forest plots, plus 3
x 1 ha plots in savanna. The 10 ha plots included LPG-01, OKO-01, OKO-02 and OKO-03, the 4 x 1 ha FBRMS plots where
TLS was collected in 2017 and 2022.
**Tree census: FBRMS-03: Kabili-Sepilok, Malaysian Borneo**
In the Kabili-Sepilok FBRMS, tree census data was collected during 2020 - 2022 for a total of 9 x 4 ha plots covering most of
the long-term plots at this site. These 4 ha plots included SEP-11, SEP-12, and SEP-30, the 3 x 1 ha FBRMS plots where TLS
was collected in 2017. A 2 ha plot, one of the oldest in the global tropics, dating back to 1958 (RP-17 = SEP-06, sandstone
forest) was also censused. The 2020-2022 census was overdue as these plots had not been censused since 2013.

Plot meta-data, including geography, institution, personnel and historical context, as well as tree-level census attributes (tag,
identity, diameter, point of measurement, stem condition, height, sub-plot, and, where measured x, y coordinates of 5 x 5 m
subplots) and multi-census attributes (tree demography and measurement trajectory and protocols, including growth, point of
measurement changes, recruitment, mortality, and mortality mode) were recorded for all Gabon and Malaysia FBRMS plots.


The ForestScan Project data package, includes data from the 2022 tree census collected during February and March for the
Gabon FBRMS plots and the Malaysian FBRMS plots census data collected in October 2020 for FBRMS plot SEP-11, in
March 2020 for plot SEP-12, and in June 2021 for plot SEP-30. This data package can be accessed via
https://doi.org/10.5521/forestplots.net/2025_2 (Chavana-Bryant et al., 2025a).
**2.2.2 Terrestrial LiDAR Scanning (TLS)**
Terrestrial LiDAR data was collected to provide state-of-the-art estimates of tree- and stand-scale above-ground biomass
(AGB) for each Forest Biomass Research Monitoring Site (FBRMS). These LiDAR measurements, collected using the
protocol described in the following sections, produce millimetre-accurate 3D point clouds/digital twins representing the forest
at each FBRMS. Terrestrial Laser Scanning (TLS) chain sampling (Wilkes et al., 2017), as illustrated and described in Fig. 5,
was employed at all three FBRMS. This data was processed to construct explicit Quantitative Structural Models (QSMs)
describing individual trees within each FBRMS with a stem diameter ≥ 10 cm. Tree- and stand-scale AGB estimates were then
calculated from the volumes of these models, using wood density values derived from published sources based on species
identification from botanical surveys.

(a)    (b)

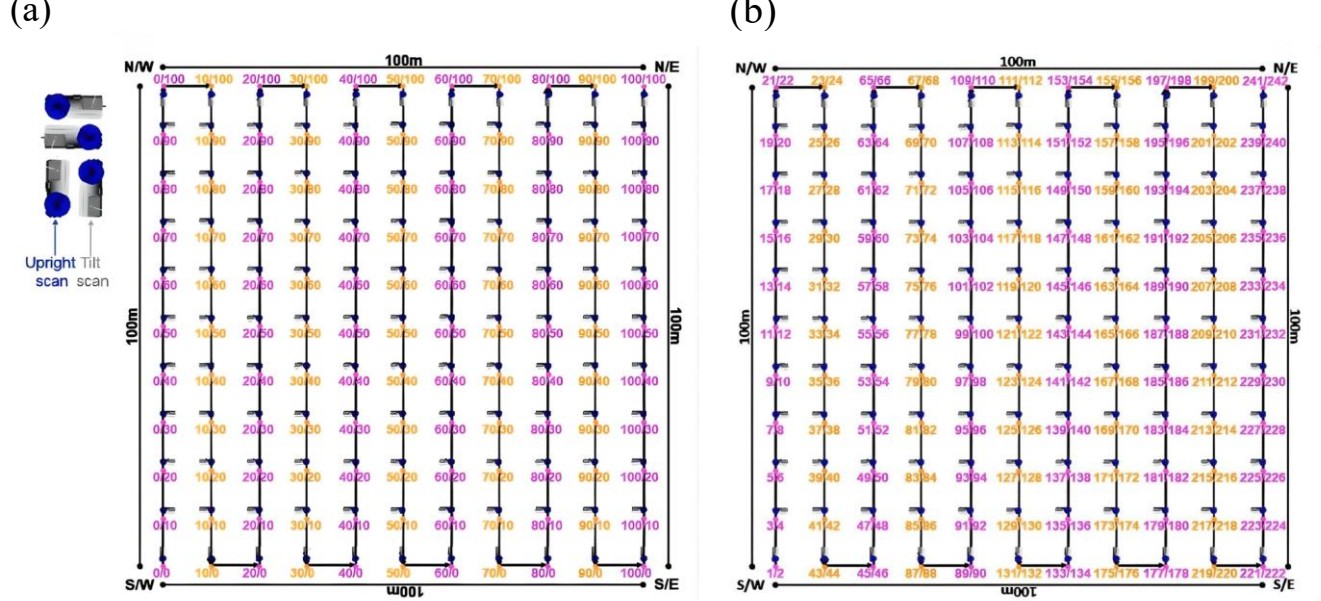


**Figure 5a & b:** Terrestrial LiDAR Scanning (TLS) chain sampling was employed to capture high-quality LiDAR data suitable
for accurate tree- and stand-scale above-ground biomass (AGB) estimation. Chain sampling was deployed over a 10 m
Cartesian grid, resulting in 11 sampling lines with 11 scan positions along each line (i.e., 0 – 10) within 1 ha forest plots.
Sampling lines were established in a south-to-north direction (standard practice) and colour-coded using flagging tape, with
the ID of each scan position written in permanent marker. Scan positions were identified by their line number and grid position,
as shown in 5b (left). Due to the scanner's 100° field of view, capturing a complete scene at each scan position required two



scans—upright and tilted. Consequently, 242 scans were collected from 121 positions at each 1 ha forest plot. The order in
which the 242 individual scans were collected at each plot is depicted in 5c (right). The first scan at each plot was collected at
the southwest corner, i.e., scan position 0,0 (unless impeded by obstacles such as streams, large tree falls, etc., or if the plot
was oriented differently). To facilitate scan registration, all tilt scans along the first sampling line were oriented towards the
same sampling position along the next sampling line, and all other tilt scans along plot edges were oriented towards the inside
of the plot so that the previous scan location was within the tilt-scan field of view. Depending on the density of the canopy
understory, terrain, and wind conditions (ideally, low to zero wind and no rain or mist/fog), a team of three experienced TLS
operators required 1–2 full working days to set up the chain sampling grid and 3–5 days to complete the scanning of a 1 ha
plot.
**TLS: FBRMS-01: Paracou, French Guiana**
Terrestrial LiDAR Scanning (TLS) data was collected in Paracou over two separate periods due to interruptions caused by the
COVID-19 pandemic. In 2019, censused plot FG6c2 was scanned during October and November (Brede et al., 2022a). The
scanning was conducted over a 200 x 200 m² area (equivalent to 16 quarter-hectare plots) covering two FG6 subplots, resulting
in 21 x 21 scan lines with 10 m grid spacing. A RIEGL VZ-400 scanner (RIEGL Laser Measurement Systems GmbH, 2025)
was used, with retro-reflective targets placed between scan positions to facilitate coarse registration (Wilkes et al., 2017).
In 2022, three 1 ha censused plots were scanned in Paracou during September and October using a RIEGL VZ-400i scanner
(RIEGL Laser Measurement Systems GmbH, 2025). These plots were selected to represent the disturbance gradient found at
this site, as shown in Table 1 below. All three plots were also scanned with ALS and plot FG6c2 also scanned with UAV-LS.

**Table 1:** Overview of plots scanned in 2022 with TLS in Paracou, French Guiana.

| Plot ID | Subplot | Logging treatment | Description | AGB | Lat | Long |
|---------|---------|-------------------|-------------|-----|-----|------|
| FG6c2 | 2 | Control | Old-growth, lowland, Terra firme rainforest | High | 5.27 | -52.92 |
| FG5c1 | 1 | T2 | Old-growth, lowland, Terra firme rainforest with mid-level logging disturbance | Mid | 5.27 | -52.92 |
| FG8c4 | 4 | T3 | Old-growth, lowland, Terra firme rainforest with high-level of logging disturbance | Low | 5.26 | -52.93 |


TLS data for all three Forest Biomass Research Monitoring Sites (FBRMS) were collected using a RIEGL VZ-400 laser
scanner or its newer model, the VZ-400i, which has very similar technical specifications and includes Global Navigation
Satellite System (GNSS) Real-Time Kinematic (RTK) positioning (RIEGL Laser Measurement Systems GmbH, 2025). RTK



GNSS facilitates TLS data acquisition by replacing the labour-intensive and time-consuming task of placing and continuously
relocating retro-reflective targets between scan positions as required by the RIEGL VZ-400 scanner. Common targets between
adjacent scan locations were later identified and used to create a registration chain that integrates the 3D point cloud of a
scanned plot. GNSS RTK has replaced the use of common targets, enabling the absolute (latitude, longitude, and altitude) and
relative (between base and rover GNSS) positioning of individual scans with centimetre precision, which makes the auto-
registration of scans in real-time possible. This GNSS-enabled auto-registration significantly reduces the time and effort
required to both collect and register TLS data. Furthermore, data collected with the VZ-400i are backwards compatible with
data from the older VZ-400 scanner, allowing for consistent processing and comparison over time.

Both the RIEGL VZ-400 and VZ-400i scanners are time-of-flight, multiple-return, waveform instruments operating in the
near-infrared. These instruments have generally been used with an angular resolution of 0.04° in dense forests, resulting in
approximately 22.4 million emitted pulses per scan (i.e., 5.42 billion per hectare). While angular resolution can be increased,
scanning time also increases linearly, this choice is therefore a compromise. Up to seven returns can be resolved per pulse,
with a nominal ranging accuracy of 5 mm. The laser itself is characterised by a beam divergence of 0.35 mrad, and the diameter
of the beam at emission is 7 mm (e.g., the diameter of the beam at a range of 50 m would be 21 mm). The pulse repetition rate
can be set between 300 and 1200 kHz, but higher scan rates use lower power returns. In this study, a rate of 300 kHz was used,
with each scan taking approximately 3 minutes to complete at this rate.
**TLS: FBRMS-02: Lopé, Gabon**
TLS data was collected in Lopé over two separate periods due to interruptions caused by the COVID-19 pandemic. In 2016,
four 1 ha censused plots were scanned during July and August. The four sampled plots, shown in Table 2, were selected to
represent the diversity of forest types found within this site. A RIEGL VZ-400 scanner (RIEGL Laser Measurement Systems
GmbH, 2025) was used, with retro-reflective targets positioned between scan locations to facilitate coarse registration (Wilkes
et al., 2017). In 2022, the same four plots were rescanned using a RIEGL VZ-400i with GNSS RTK-enabled auto-registration,
eliminating the need for retro-reflective targets between scan positions.

**Table 2:** Overview of plots scanned with TLS in Lopé National Park, Gabon.

| Plot ID | Description (local plot name / forest type) | Lat | Long |
|---------|---------------------------------------------|------|-------|
| LNL-07 | OKO-01 / Maturing secondary Okoumé forest | -0.19 | 11.58 |
| LNL-08 | OKO-02 / Maturing secondary Okoumé-Sacoglottis forest | -0.19 | 11.58 |
| LNL-09 | OKO-03 / Maturing secondary Okoumé forest | -0.19 | 11.57 |



| LPG-01 | Angak / Old-growth forest | | -0.17 | 11.57 |
|--------|---------------------------|--|-------|-------|

**TLS: FBRMS-03: Kabili-Sepilok, Malaysian Borneo**

TLS data was collected for three 1 ha forest plots at this FBRMS during March 2017. The three sampled plots, shown in Table 3, were selected to represent the three distinct forest types found within this site. TLS data was collected Subplot 2 was scanned with TLS for plots SEP-11 and SEP-12 and subplot 3 in SEP-30.

A RIEGL VZ-400 scanner (RIEGL Laser Measurement Systems GmbH, 2025) was used, with retro-reflective targets positioned between scan locations to facilitate coarse registration (Wilkes et al., 2017).

**Table 3:** Overview of plots scanned with TLS in Kabili-Sepilok Forest Reserve, Malaysia. Note: subplot 2 was

| Plot ID | Subplot | Description (local plot name / forest type) | Lat | Long |
|---------|---------|---------------------------------------------|------|--------|
| SEP-11 | 2 | 292/3 / Sandstone forest | 5.86 | 117.94 |
| SEP-12 | 2 | 292/1 / Alluvial forest | 5.86 | 117.93 |
| SEP-30 | 3 | 508/4 / Kerangas forest | 5.86 | 117.97 |

**TLS data processing**

Terrestrial LiDAR data was collected and processed to provide state-of-the-art estimates of tree- and plot-scale structural attributes and above-ground biomass (AGB) for each ForestScan FBRMS. TLS data was processed using the *TLS2trees* processing pipeline (Wilkes et al., 2023). *TLS2trees* is a set of free and open-source software (FOSS) Python command line tools designed to automate tree extraction from TLS point clouds, utilising high-performance computing (HPC) facilities, particularly GPUs (Wilkes et al., 2023). By automating the previously time-consuming process of manual tree extraction, *TLS2trees* has overcome a major processing bottleneck.

*TLS2trees* employs several methods: (1) semantic segmentation to classify point clouds into leaf, wood, woody debris, and ground point classes; (2) instance segmentation to separate point clouds into sets representing individual trees; and (3) estimation of woody volume using a QSM approach. TLS data was processed to construct explicit quantitative structural models (QSMs) describing individual trees with a stem diameter at breast height ≥ 10 cm within each ForestScan plot. Tree- and stand-scale AGB estimates were then calculated from the volumes of these models, using wood density values derived from published sources based on species identification from botanical surveys conducted during the census measurements. Smaller stems (DBH < 10 cm) are estimated to contribute approximately 10% of AGB in a 1 ha plot (Chave et al., 2014).






The five main processing steps required to retrieve structural attributes from the acquired TLS data are described below. These
processing steps demand significant computational resources -a full 1 ha plot can take 3.4 to 4 days to process from start to
finish on an HPC cluster, running on multiple CPUs.
**1. Individual scan registration into plot-level point cloud**
This process was conducted in a near-automated manner using the RIEGL VZ-400i's new GNSS RTK positioning capabilities
and the enhanced RIEGL RiSCAN software (versions 2.14 - 2.17). The integrated Auto Registration 2 (AR2) process utilises
GNSS RTK data to update the scanner's position and orientation, even in tilt mode, enabling real-time automated coarse
registration during scanning. When auto-registration fails, unregistered scans can be identified, adjusted, and their positioning
fine-tuned to ≤ 2 cm accuracy using Multi Station Adjustment 2 (MSA2).

The use of GNSS significantly enhances the utility and accessibility of TLS by drastically reducing both data acquisition and
processing time. This is achieved by (1) as previously mentioned, replacing the previous labour-intensive and time-consuming
practice of using common retro-reflective targets to link adjacent scan positions into a registration chain (Wilkes et al., 2017),
and (2) reducing the manual processing registration time by an experienced user to 1 - 2 days per hectare, which is less than
half the time required when using retro-reflective targets.

Following registration, a plot-level point cloud was generated for each of the scanned FBRMS plots, comprising 242 individual
scan-level point clouds, potentially containing more than 5.42 billion points. A small section of the plot-level point cloud
collected from a forest stand in Paracou, French Guiana, is shown in Fig. 6.

Earth System
Science
Data

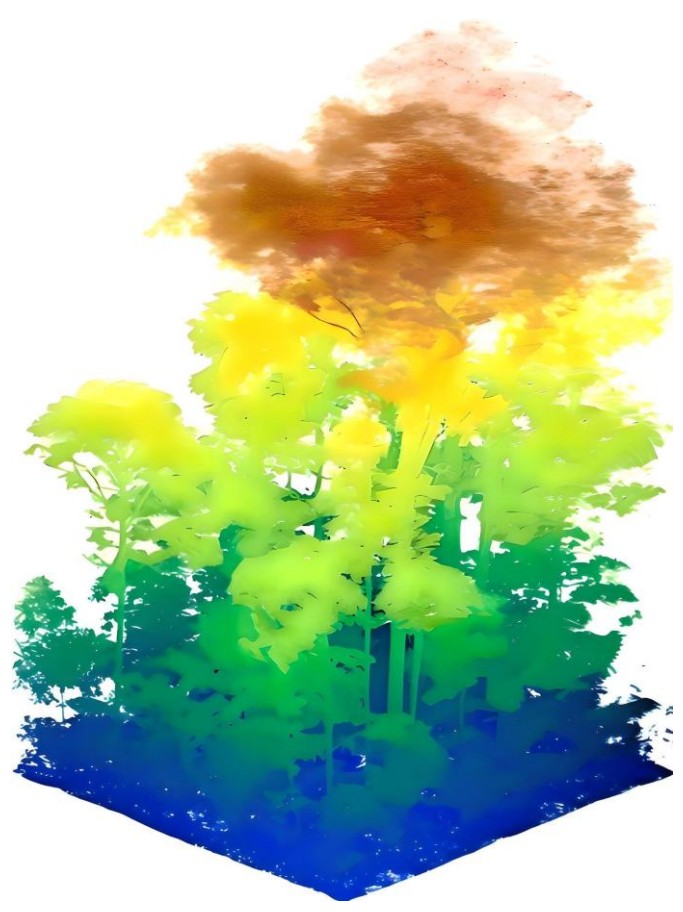


**Figure 6:** A section of plot-level point cloud coloured by height (0 - 45m) from plot FG6c2 in FBRMS-01: Paracou.

The subsequent four processing steps were performed in a semi-automated manner using the *TLS2trees* processing pipeline
(Wilkes et al., 2023), as described below.

**2. Pre-processing of plot-level point clouds**

Pre-processing is accomplished through a three-step procedure. Initially, the point clouds are clipped to the plot extent, with a
10 m buffer added around the plot, and projected onto a 10 m x 10 m grid to create a set of tiled point clouds. These 10 m tiled
point clouds are then converted from the RIEGL proprietary file format .rxp to .ply format. The final pre-processing step
involves generating a tile index to map the spatial location of the tiled point clouds.

**3. Semantic segmentation: wood-leaf separation**

*TLS2trees* implements a modified version of the Forest Structural Complexity Tool (FSCT) semantic segmentation method by
Krisanski et al. (2021) to classify points within tiled point clouds into homogeneous groups or classes of different biophysical



components: leaf, wood, coarse woody debris, or ground. An example of the wood and leaf classes extracted from tree-level
point clouds is illustrated in Fig. 7 below.

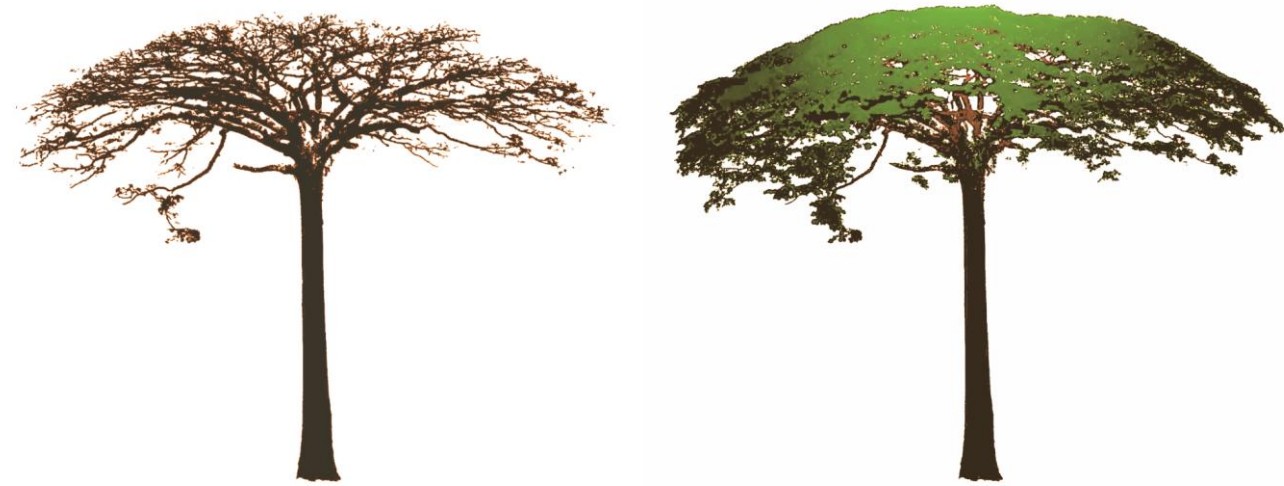

**Figure 7:** Wood (brown) and leaf (green) points in tree-level point cloud for the largest *Baillonella toxisperma* (Maobi) tree
(~ 40 m in height with an almost circular canopy ~50 m wide) in plot LPG-01 in FBRMS-02: Lopé, Gabon.
**4.  Instance segmentation: individual tree separation**
This modelling step identifies and segments individual trees via a 2-step process. The Dijksta's shortest path method first
groups all points identified as wood into a set of individual woody stems to which points identified as leaf are then assigned.
A small group of trees automatically segmented from a plot in Gabon are shown in Fig. 8.

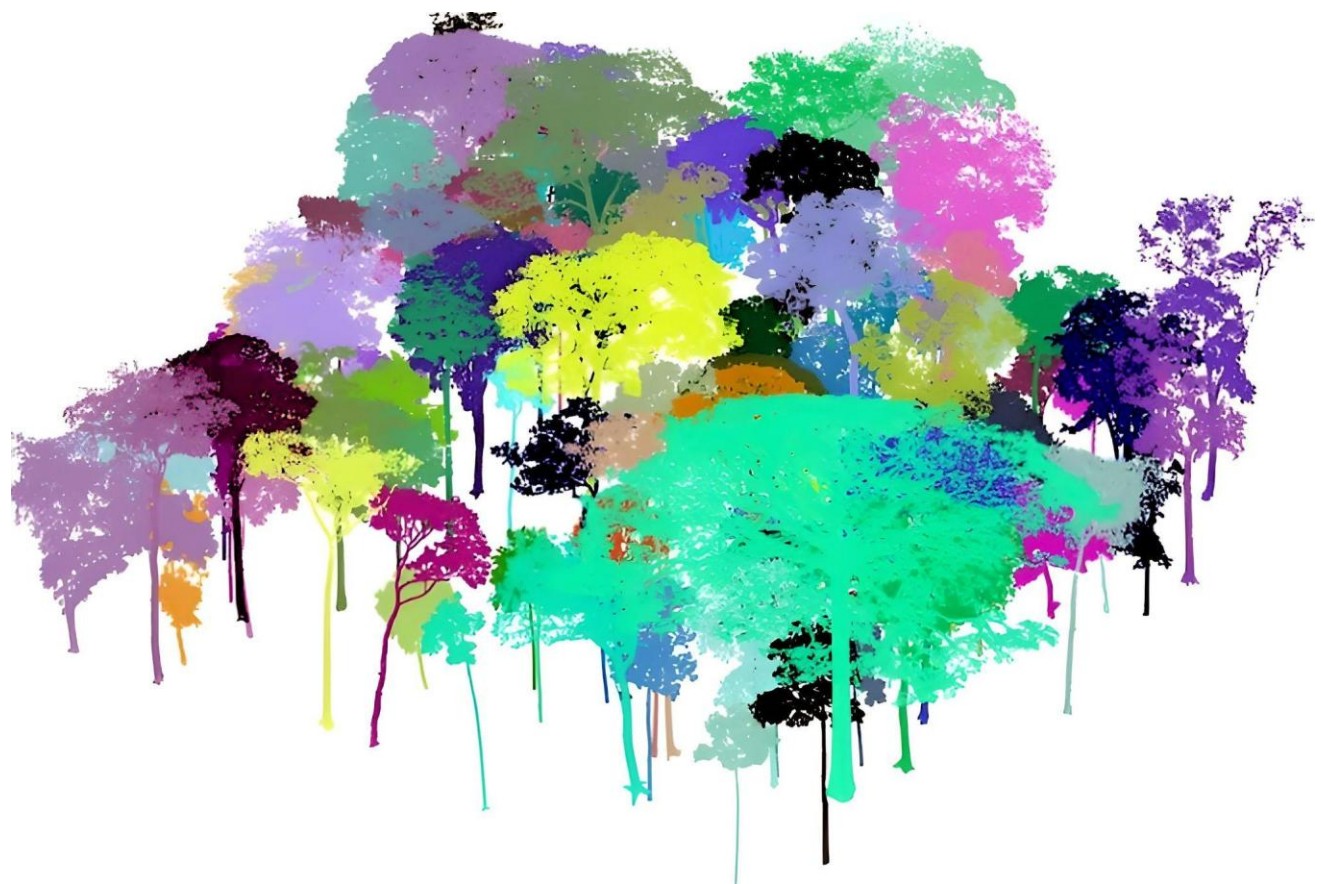

**Figure 8:** Individual tree-level point clouds acquired from plot LPG-01 in FBRMS-02: Lopé, Gabon.

## 5.  TreeQSM: quantitative structural models and results

Quantitative structural models (QSMs) were constructed in a near-automated manner from each woody tree-level point cloud using the TreeQSM software package version 2.3 (Raumonen et al., 2013), which employs cylinders to reconstruct underlying woody surfaces, as illustrated in Fig. 9. The QSM fitting process involves: (i) reducing each point cloud to a series of patches, (ii) assessing the arrangement and neighbour relationships between patches, and (iii) robustly fitting cylinders onto common patches.

The overall fit of the cylinders is controlled by three parameters, which are iterated into 125 different parameter sets, each generating five models. This results in a total of 625 potential models per segmented tree. An optimal model is then selected by minimising the point-to-cylinder surface distance (Burt et al., 2019; Martin-Ducup et al., 2021). Estimates of morphological and topological traits such as volume, length, and surface area metrics, along with their optimal mean and standard deviation, are generated from the five models that share the same parameters as the optimal model.





**Figure 9:** QSMs derived from the point clouds shown in Fig. 8.

The final modelling outputs for each tree are saved into a "tree-attributes.csv" report file, which is generated at the end of the modelling exercise. This file also includes tree and plot level carbon and AGB estimates, the last of which are based on a mean pantropical wood density value of 0.5 g/cm³ estimated from the DRYAD global database of tropical forest wood density (Zanne et al., 2009). FBRMS plot AGB was also estimated using DRYAD-derived regional mean wood densities as shown in Table 4.

Figures of all individually segmented trees arranged by tree DBH size (largest to smallest DBH) are also generated for each FBRMS plot, examples of which can be seen in Fig. 10. Figure 11 provides a comparison of the distribution of diameter at breast height (DBH) measurements collected by tree census and TLS methods at each of the 10 ForestScan FBRMS 1 ha plots.

**TLS datasets**

The following terrestrial LiDAR-derived products are available for each of the 10 ForestScan FBRMS plots:





1. Raw terrestrial LiDAR data from each scan, stored in the RXP data stream format developed by RIEGL.

2. Transformation matrices necessary for rotating and translating the coordinate system of each scan, into the coordinate system of the first scan. Stored in DAT format.

3. Pre-processed terrestrial LiDAR data:
   a. full-resolution 10m tiled plot point clouds stored in polygon PLY format.
   b. downsampled 10m tiled plot point clouds stored in polygon PLY format.
   c. A tile_index file (maps the spatial location of the tiled point clouds) stored in DAT format.
   d. Bounding geometry files setting plot boundaries with and without a buffer surrounding the plot. Stored in shapefile SHP, DBF, SHX and CPG formats.

4. Downsampled 10m tiled plot point clouds segmented into leaf, wood, ground points or coarse woody debris. Stored in polygon file format PLY format.

5. Wood-leaf separated tree-level point clouds stored in polygon PLY format.

6. QSM files:
   a. **in_plot** CSV (for plots processed with *TLS2trees*) lists all trees to be modelled with QSMs as they are located inside the plot boundary.
   b. **out_plot** CSV (for plots processed with *TLS2trees*) lists all trees NOT to be modelled as they are located outside the plot boundary.
   c. **plot_boundary** CSV (for plots processed with *TLS2trees*) shows the location of all in_plot trees within each plot boundary.
   d. **QSM processing files** (.MAT Matlab).
   e. **QSMs** derived from each woody tree-level point cloud, (.MAT Matlab).

7. Tree-attributes file (.CSV) containing biophysical parameters derived from both the point clouds and QSMs: stem diameter, tree height, tree-level volume and AGB with uncertainty, plot-level AGB and associated uncertainty.

8. Figures of all individually segmented trees arranged by tree DBH size (largest to smallest DBH) for each FBRMS plot (see Figure 9) (PNG image format).

9. GNSS coordinates (geographical coordinate system: WGS84 Cartesian) for all scans stored in KMZ zip-compressed format. These files are available for the seven French Guiana and Gabon FBRMS plots.

These TLS ForestScan FBRMS 1 ha plot datasets are freely available via the Centre for Environmental Data Analysis (CEDA) with URLs and DOIs provided in section 4. Data access.





**Table 4:** Summary statistics for the 10 FBRMS ForestScan TLS plot datasets. AGB was estimated using different wood
densities based on the DRYAD global database of tropical forest wood density (Zanne et al., 2009): 1) the *TLS2Trees*
pantropical mean wood density, 2) a regional mean wood density for Tropical Africa (TAF) for our FBRMS plots in Gabon
(GA), 3) a regional mean wood density for South-East Asia (TS-EA) for our FBRMS plots in Malaysia (MY), 4) a regional
mean wood density for South America (TSA), 5) a Guyana community-mean wood density for FBRMS plots in French Guiana
(GF), and 6) an allometric AGB estimates for all FBRMS plots based on Chave et al. (2014).

| Plot ID | Site | Census trees (≥10 cm DBH) | TLS2trees plot summary | | | | TLS2trees Carbon estimation | | TLS2trees AGB estimations | | | Tropical Africa (TAF) / Tropical South America (TSA) / Tropical South-East Asia (TS-EA) AGB estimations | | | Guyana AGB estimations | | | 2014 Allometric AGB estimation |
|---|---|---|---|---|---|---|---|---|---|---|---|---|---|---|---|---|---|---|
| | | | TLS trees (#) | TLS vs Census trees (%) | TLS plot area (ha) | TLS plot volume (m³) | Plot C (t) | C per ha (t/ha) | Wood density (g/cm³) | Plot AGB (t) | AGB per ha (t/ha) | Wood density (g/cm³) | Plot AGB (t) | AGB per ha (t/ha) | Wood density (g/cm³) | Plot AGB (t) | AGB per ha (t/ha) | Plot AGB (t) |
| OKO-01 | GA | 388 | 397 | 2.58 | 1.08 | 829.05 | 195.24 | 181.60 | 0.5 | 414.52 | 385.57 | 0.60 | 495.77 | 459.05 | | | | 378.62 |
| OKO-02 | GA | 472 | 473 | 0.21 | 1.02 | 625.45 | 147.29 | 143.97 | 0.5 | 312.72 | 305.67 | 0.60 | 374.02 | 366.69 | | | | 351.35 |
| OKO-03 | GA | 339 | 355 | 4.72 | 1.04 | 959.59 | 225.98 | 218.19 | 0.5 | 479.79 | 463.26 | 0.60 | 573.83 | 551.76 | | | | 372.82 |
| LPG-01 | GA | 340 | 275 | -19.12 | 1.05 | 477.88 | 112.54 | 107.16 | 0.5 | 238.94 | 227.52 | 0.60 | 285.77 | 272.17 | | | | 459.85 |
| FG5c1 | GF | 1110 | 804 | -27.57 | 1.06 | 529.67 | 124.74 | 117.62 | 0.5 | 264.83 | 249.73 | 0.63 | 334.75 | 315.80 | 0.73 | 386.66 | 409.86 | 327.30 |
| FG6c2 | GF | 902 | 832 | -7.76 | 1.10 | 751.13 | 176.89 | 161.48 | 0.5 | 375.57 | 342.86 | 0.63 | 474.72 | 431.56 | 0.73 | 548.33 | 603.16 | 421.90 |
| FG8c4 | GF | 1116 | 1090 | -2.33 | 1.09 | 625.80 | 147.38 | 135.76 | 0.5 | 312.90 | 288.24 | 0.63 | 395.50 | 362.85 | 0.73 | 456.83 | 497.95 | 286.10 |
| SEP-11 | MY | 584 | 659 | 12.84 | 1.05 | 961.36 | 226.40 | 214.67 | 0.5 | 480.68 | 455.78 | 0.57 | 551.82 | 579.41 | | | | 499.91 |
| SEP-12 | MY | 469 | 380 | -18.99 | 1.13 | 765.51 | 180.28 | 158.98 | 0.5 | 382.76 | 337.53 | 0.57 | 439.40 | 496.53 | | | | 443.45 |
| SEP-30 | MY | 787 | 986 | 25.29 | 1.03 | 374.66 | 88.23 | 85.25 | 0.5 | 187.33 | 181.01 | 0.57 | 215.05 | 221.50 | | | | 311.54 |






**Figure 10:** Examples of the largest trees (up to 30 trees) arranged in decreasing DBH size (1.3 m trunk height) for each of the 10 ForestScan FBRMS plots. The upper limit of the Y axis varies and ranges from 30 m to 60 m maximum tree size between plots.



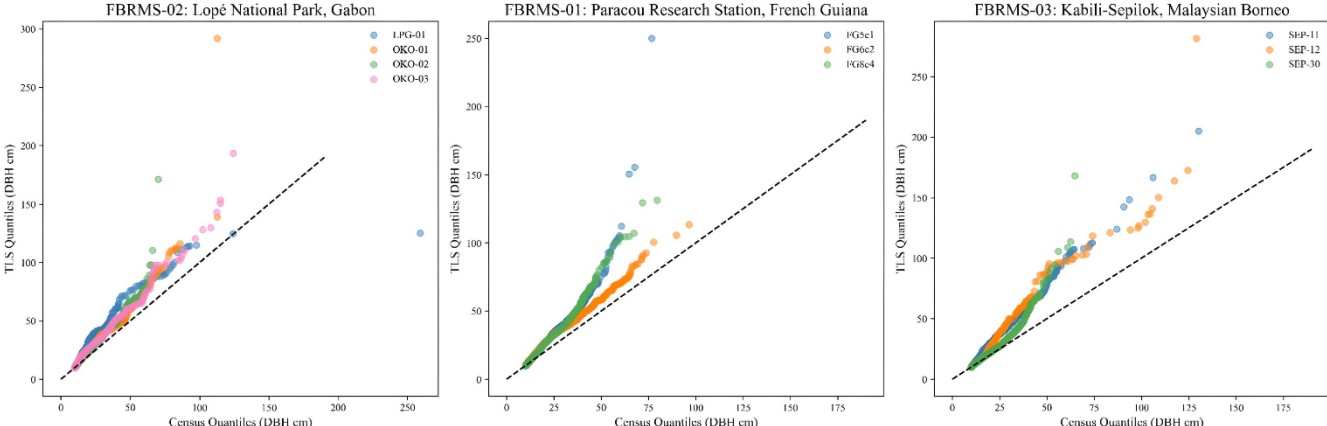

**Figure 11:** Quantile-Quantile (QQ) plots comparing the distribution of diameter at breast height (DBH) measurements collected by tree census and TLS methods at each of the 10 ForestScan FBRMS 1 ha plots. TreeQSM measures DBH at the standard height of 1.3 m for each TLS-extracted tree, whereas census DBH measurements are routinely adapted to account for tree buttresses found among larger trees. Generally, census and TLS DBH measurements are in good agreement; however, deviations for larger DBH values can be improved by adapting the DBH extraction of large buttressed trees once these trees are matched to their census counterparts. The 1:1 reference line (dotted black line) represents perfect agreement between census and TLS-extracted DBH measurements.



### 2.2.3 UAV-borne laser scanning (UAV-LS)

Unlike TLS, there are currently no best practice guidelines for UAV-LS data acquisition for forest characterisation. Therefore, flight plans and parameters were implemented on a case-by-case basis, considering the site, instrument, sensor, and application. An important consideration in this respect is whether visual line of sight (VLOS) needs to be maintained, i.e., the visibility of the platform by the pilot throughout the mission. Regulations on this vary nationally and are changing rapidly as technology evolves and the use of UAVs expands. In Europe, for example, a risk-based approach has been introduced, allowing beyond VLOS (BVLOS) when risks are negligible. Given the remote nature of the ForestScan FBRMSs, the likelihood of severe incidents involving non-crew persons is very low.

Another important consideration is the availability of take-off and landing areas. Vertical take-off and landing (VTOL) platforms (e.g., quadcopters and octocopters) require smaller areas and are more flexible, while fixed-wing platforms may require substantial take-off and landing sites, although they offer greater area coverage and flight duration. The actual take-off area for VTOL platforms is highly dependent on the skills and confidence of the pilot. However, a very small take-off area surrounded by tree crowns typically also means low chances for VLOS operation, unless an above-canopy platform such as a cherry-picker is available.

In the context of VTOL and VLOS operations, viewshed analysis based on already acquired ALS data has proved useful. ALS point clouds can be used to derive initial Digital Surface Models (DSM), which can identify possible take-off positions. Viewshed analysis can then use the DSM to simulate the visibility of the UAV from the take-off position.

During data collection, attention should also be paid to acquiring access to GNSS observables from permanent base stations (e.g., CORS network) or to collecting observables with a temporary base station (e.g., Emlid Reach RS+ or RS2). A base station should be positioned less than 15 km from the survey area. For some platforms, Real-Time Kinematic (RTK), and therefore radio connection, between the UAV and base station can be an added constraint.

Finally, the external framework for UAV operations comprises the legal regulations to operate UAVs, which are taught during pilot licence training. Consideration should be given to the legal issues involved in acquiring permission to use the airspace. Many aeronautical authorities have adopted the practice of regarding UAVs as regular airspace users comparable to crewed aircraft. In certain areas, this can have significant implications for planning, particularly regarding permissions that must be obtained and licences required by the pilots. Special attention should be paid to airports, as they are surrounded by controlled traffic regions (CTR). Flying within CTRs is only possible with special licences and equipment (transponder, radio). New technical developments are underway to equip UAVs with transponders, making CTR operations more feasible in the future. Additionally, military airspace (particularly relevant to FBRMS-01) requires thorough preparation and prior communication





with the relevant authorities. Unlike civil airspace, low-flying exercises can be conducted in military airspace; however, the
military has the right to completely block areas for exercises, even at short notice.
Our UAV-LS data collections used three different LiDAR systems built by RIEGL at FBRMS-01 and FBRMS-02. All systems
are based on the time-of-flight principle and capable of multi-return registration with the miniVUX-1DL being a specific
downward-looking sensor designed for fixed-wing UAVs. Technical specifications for all three UAV-LS sensor systems are
provided in Table 5 below.
**Table 5:** UAV-LS sensor systems used at ForestScan FBRMS-01 and FBRMS-02.

| Characteristic | miniVUX-1UAV | VUX-1UAV | miniVUX-1DL |
|---|---|---|---|
| Max Pulse Repetition Rate [kHz] | 100 | 550 | 100 |
| Wavelength [nm] | 905 | 1550 | 905 |
| FOV [°] | 360 | 330 | 46 |
| Ranging accuracy / precision [mm] | 15 / 10 | 10 / 5 | 15 / 10 |
| Max range [m] | 330 @ ρ ≥ 80% | 1050 @ ρ ≥ 80% | 260 @ ρ ≥ 80% |
| Weight [kg] | 1.55 | 3.5 | 2.4 |
| Inertial Meassurement Unit (IMU) | Applanix APX20 | Applanix AP20 | Applanix APX15 |
| Operated by | AMAP | Wageningen University | University of Edinburgh |
| Operated on | DJI M600 | RiCOPTER | DELAIR DT26X |
| Flight location | FBRMS-01: Paracou | FBRMS-01: Paracou | FBRMS-02: Lopé |


**UAV-LS: FBRMS-01: Paracou, French Guiana**

UAV-LS data was collected in October 2019 using two different scanning systems as shown in Table 5. As detailed in Fig. 12
and Table 6 below, a total of 11 flights were conducted using the RIEGL VUX-1UAV mounted on a RIEGL RiCOPTER UAV
and flown over the same 200 x 200 m² sub-area that was scanned with TLS. Six of these flights covered the entire plot sub-
area with 20 m spacing between flight lines at an altitude of 120 m above ground level (AGL). The remaining five flights
covered only the north-east 100 x 100 m² area with a criss-cross pattern to maximise the diversity of viewing angles into the
canopy. These latter flights were conducted at a lower altitude of 90 m AGL to increase point density; however, the entire plot
could not be covered without losing visual line of sight (VLOS).




**Figure 12:** UAV-LS acquisitions over FBRMS-01: Paracou with flight trajectories covering ForestScan plot FG6c2. Inset
map data © OpenStreetMap contributors, available from https://www.openstreetmap.org.


**Table 6:** Overview of 2019 VUX-1 UAV-LS flights in FBRMS-01: Paracou. Flight pattern N-S = flight in lines oriented from
North to South, etc. and criss-cross = multiple flight directions.

| Plot ID | Date & Time (UTC) | Flight pattern | Height AGL [m] | Velocity [m/s] | Pulse Repetition Rate [kHz] |
|---------|-------------------|----------------|----------------|----------------|------------------------------|
| P6 200 | 191018_114105 | Manual | 115 | 4 | 550 |
| P6 200 | 191018_132827 | N-S | 110 | 6 | 550 |
| P6 200 | 191018_143654 | E-W | 105 | 7 | 550 |





| P6 200 | 191018_175753 | NW-SE | 115 | 6 | 550 |
| P6 200 | 191018_192314 | NE-SW | 105 | 6 | 550 |
| P6 200 | 191019_163412 | N-S | 120 | 6 | 300 |
| P6 200 | 191020_184540 | N-S | 120 | 6 | 100 |
| P6 100 | 191019_121041 | criss-cross | 95 | 4 | 550 |
| P6 100 | 191019_124109 | criss-cross | 85 | 4 | 550 |
| P6 100 | 191019_181957 | criss-cross | 95 | 4 | 550 |
| P6 100 | 191019_194142 | criss-cross | 90 | 4 | 550 |


UAV-LS data was also collected over several plots during the same mission as that using the RiCOPTER but using a separate
UAV-LS system -a Yellowscan Vx20 containing a RIEGL Mini-VUX scanner and Applanix 20 IMU- mounted on a DJI
M600. Scanning was performed using automated flight plans with the UgCS route planning software in grid mode. Flight
details can be found in Table 7 below. To allow for comparisons with the VUX system, coincident acquisitions were performed
over Plot 6, Arbocel (a few kilometres west of the Paracou site and covering ForestScan plot FG6c2), and the Plantation area
(500 metres north of the Paracou site). The two sites acquired outside of Paracou correspond to contrasting vegetation: young
secondary forest for Arbocel and plantations, for which field data can be obtained. A full description of the UAV-LS data
collection for this FBRMS is provided in Brede et al. (2022b).

**Table 7:** Overview of miniVUX UAV-LS flights in FBRMS-01 Paracou.

| Plot ID | Date & Time (UTC) | Interline [m] | Direction [°] | Speed [m/s] | Alt [m] |
|---|---|---|---|---|---|
| Arbocel | YS-20191017-185811 | 50 | | 5 | 50 AGL |
| Arbocel | YS-20191017-201741 | 50 | | 5 | 50 AGL |
| Paracou P6 | YS-20191018-124006 | 20 | 345 | 5 | 80 AGL |
| Paracou P6 | YS-20191018-131043 | 20 | 345 | 5 | 80 AGL |
| Paracou P6 | YS-20191018-183057 | 20 | 120 | 5 | 80 AGL |
| Paracou P6 | YS-20191018-185416 | 20 | 120 | 5 | 80 AGL |



| Paracou P6 | YS-20191018-200932 | 20 | 165 | 5 | 145 amsl |
|---|---|---|---|---|---|
| Paracou P6 | YS-20191019-115917 | 20 | 75 | 5 | 145 amsl |
| Paracou P6 | YS-20191019-190345 | 20 | 75 | 5 | 80 AGL |
| Paracou P6 | YS-20191020-191757 | 40 | 345 | 3 | 100 amsl |
| Paracou P7 | YS-20191020-142306 | 50 | 75 & 345 | 5 | 100 amsl |
| Paracou P8 | YS-20191020-113907 | 50 | 75 & 345 | 5 | 105 amsl |
| Paracou P10 | YS-20191020-123019 | 50 | 75 & 345 | 5 | 105 amsl |
| Paracou P15 | YS-20191020-180828 | 50 | 75 & 345 | 5 | 100 amsl |
| Paracou P4&5 | YS-20191019-172347 | 50 | 345 | 5 | 100 amsl |
| Paracou Tower (tropiscat) | YS-20191019-162557 | 50 | 0 | 5 | 80 AGL |
| Paracou Tower (tropiscat) | YS-20191019-181021 | 50 | 90 | 5 | 105 amsl |
| Plantation | YS-20191021-142058 | 75 | 100 | 5 | 125 amsl |
| Plantation | YS-20191021-150450 | 75 | 135 | 5 | 125 amsl |


**UAV-LS data processing**

Trajectometry was post-processed in POSPac UAV (V8.3) using single station DGNSS corrections from a local SXBlue base station or the Kourou IGN network. Raw LAS points were exported using the Yellowscan CloudStation software with the 'line adjustment' option. Further improvement of inter-line matching was performed using BayesMap software to account for an undetected defect in roll angle recording on the scanner unit. Merging and processing of each flight were conducted with LAStools software (https://github.com/LAStools/LAStools v 1.0-1.4) for point cloud classification using the 'lasground' function with the options '-step 15' and '-wilderness', and for the generation of DTM, DSM, and CHM at 1 m resolution. This UAV-LS dataset is freely available via the Centre for Environmental Data Analysis (CEDA) with DOIs provided in section 4. Data access.

553

**UAV-LS: FBRMS-02: Lopé, Gabon**

UAV-LS data was collected in June 2022, concurrently with TLS data acquisition at this FBRMS. Data was acquired using a DELAIR DT26X equipped with a RIEGL miniVUX-1DL (Mcnicol et al., 2021) as seen in Figure 13 below. This platform differs from the one used at FBRMS-01: Paracou in that it is designed for large-scale data acquisitions (thousands of hectares) and is capable of operating beyond the VLOS, with an average flight speed of 17 m/s (61 km/h). Flights were conducted in perpendicular lines at a nominal altitude of 120 m above the ground surface, with an average flight line spacing of 20 m (based on 70–80% overlap). Each one-hour flight covered approximately 120–200 hectares with an estimated point density of 400 points per square metre. To obtain the required densities, several flights were conducted over the core plots from different angles (depending on wind conditions) to maximise the diversity of viewing angles into the canopy.

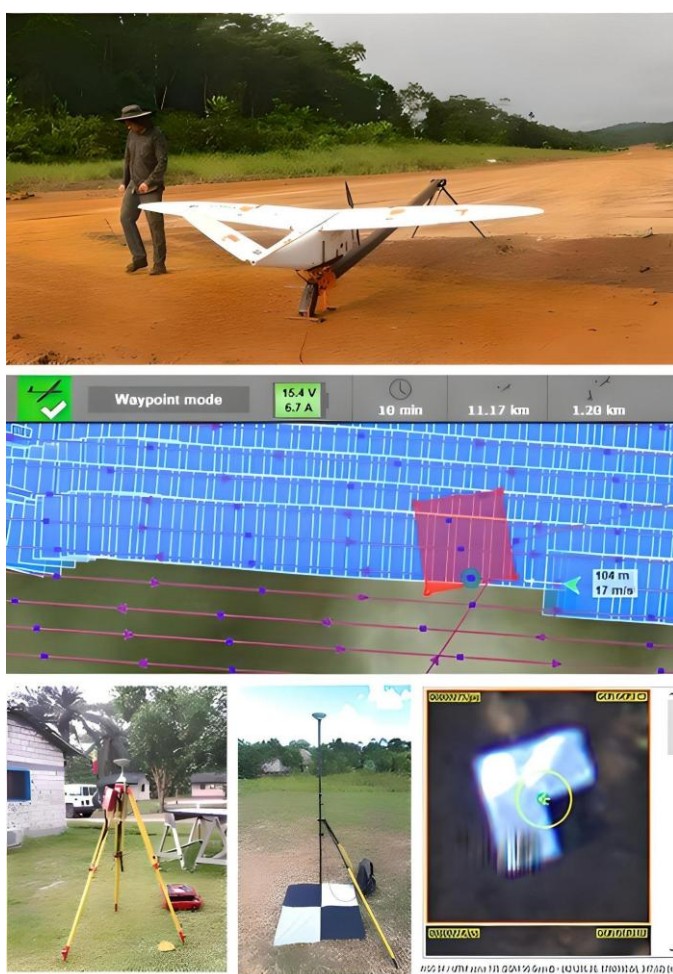

**Figure 13:** UAV-LS acquisitions at FBRMS-02: Lopé using a fixed-wing system. This UAV employs a conventional take-off and landing (CTOL) procedure, with launch aided by a catapult. Once airborne, the UAV is controlled from a laptop connected



to the UAV via an antenna. The flight trajectory is corrected to centimetre precision using data collected from a static GNSS
receiver placed within 10 km of the UAV operating area. Additional refinements and corrections are possible via ground
control points located across the study area, the positions of which are measured using a 'rover' GNSS receiver (image
originally published in McNicol et al. (2021)).

**UAV-LS data processing**

UAV-LS data processing commences with the post-processing of the platform's trajectory based on GNSS observations (rover)
in conjunction with the Inertial Measurement Unit (IMU) and Global Navigation Satellite System (GNSS) observables from a
base station, i.e., Post Processed Kinematic (PPK) processing (Brede et al., 2017). LiDAR waveforms must be interpreted to
produce discrete returns in the scanner's own coordinate system. The post-processed trajectory can then be combined with the
ranging information to generate point clouds in a global coordinate system. This processing pathway ensures global registration
of the point clouds with survey-grade accuracy in the best-case scenario. If necessary, flight lines can be further fine-registered
based on point cloud features, typically using automatic feature finding similar to RIEGL's Multi-station Adjustment (MSA)
routine for TLS. Software packages for processing are usually provided or offered by the vendor of the UAV-LS system. The
end product of this process is the globally registered point cloud.
The point cloud can be treated as an ALS point cloud, allowing the application of standard processing steps such as ground
point detection, DEM/DSM/CHM generation, and individual tree detection. The final step of tree detection remains an ongoing
development because UAV-LS has a much higher point density than ALS but typically cannot detect trunks as clearly as TLS
(Chen et al., 2021; Terryn et al., 2022; Torresan et al., 2020; Yan et al., 2020). This UAV-LS dataset is freely available via the
Centre for Environmental Data Analysis (CEDA) with DOIs provided in section 4. Data access.

**2.2.4 Airborne Laser Scanning (ALS)**

**FBRMS-01: Paracou, French Guiana**

Airborne laser scanning (ALS) data was collected over Paracou in November 2019. The data cover 10 km², including all field
plots and areas covered by terrestrial laser scanning (TLS) and unmanned aerial vehicle LiDAR scanning (UAV-LS). The data
collection was conducted by the private company Altoa using a BN2 aircraft flying at approximately 900 m altitude at a speed
of approximately 180 km/hr (that is, 50 m.s$^{-1}$). The LiDAR instrument used was a RIEGL LMS-Q780, with a minimum pulse
density of 15 points/m² and a mean pulse density of 40 points/m². The lateral overlap between two flight lines was 80%, with
a scan angle of +/- 30 degrees. During the same campaign, additional data was gathered over Nouragues Research Station in
French Guiana. This supplementary data was collected using identical scanning parameters and has been incorporated into the
ForestScan data archive.



Airborne LiDAR point cloud data for Paracou are provided in a local coordinate reference system (EPSG:2972) and freely
available via the Centre for Environmental Data Analysis (CEDA) with DOIs provided in section 4. Data access. Canopy
height models for both Paracou and Sepilok are described in Jackson et al. (2024) and available at https://doi.org/10.908679.
**FBRMS-03: Kabili-Sepilok, Malaysia**
The collection and processing of ALS data for this FBRMS are detailed in Jackson *et al.* (2024). The data was collected in
February 2020 by Ground Data Solutions using a RIEGL LMS-Q560 scanner with a scanning angle of +/- 30 degrees from a
helicopter flying at an altitude of 350 m above the forest canopy and at a speed of approximately 100 km/h (ca. 30 m.s$^{-1}$). This
dataset includes LiDAR and RedGreenBlue (RGB) imagery data collected from a helicopter over two forest sites in Sabah,
Malaysia, in February 2020. The 27 square kilometres covered by the Sepilok Reserve were fully scanned on 15 February
2020. In the Danum Valley, scanning between 19 and 22 February, 2020 covered two adjacent areas: a protected zone (20
square kilometres) and a reduced impact logging zone (9 square kilometres). These areas were selected due to the availability
of prior airborne LiDAR data collected in 2013 and 2014.

The point cloud data for this FBRMS have approximately 42 pulses per square metre and are available in LAS (LASer) format,
as well as RGB data summary rasters in .tif format. The raster images were processed with LAStools using default parameters.
Canopy Height Model (CHM), Digital Surface Model (DSM), Digital Terrain Model (DTM), and pulse density (pd) data are
also included. The RGB data are provided in .jpg format and organised by flight date. The data was georeferenced using ground
control points and provided in the 'WGS 84 / UTM 50N' coordinate reference system (EPSG:32650). This UAV-LS dataset
is freely available via the Centre for Environmental Data Analysis (CEDA) with DOIs provided in section 4. Data access.
**3. Aligning and matching datasets**
**3.1 Matching TLS to census data: stem maps**
A key step in estimating above-ground biomass (AGB) from tree-level terrestrial laser scanning (TLS) point clouds is the
selection of wood density (WD) for converting volume to mass. WD represents a significant source of uncertainty in the
indirect estimation of AGB, whether through allometry and census diameter at breast height (DBH), Earth observation (EO)-
derived canopy height, TLS-estimated volume, or other methods (Phillips et al., 2019). If the censused trees in each plot can
be matched to their TLS counterparts, literature estimates of species-specific WD (or field-measured values, if available) can
be used. In the absence of such a match, plot-level mean WD values are employed, as is common in most EO-derived estimates
that rely on large-scale allometric models (e.g. Chave et al., 2014). Research by Momo et al. (2020), Burt et al. (2020), and
Demol et al. (2021) has demonstrated that significant bias can occur in TLS-derived AGB estimates due to within-tree WD
variations when literature-derived species average WD values are used. However, Momo et al. (2020) suggest there is sufficient
correlation between vertical gradients and basal WD to allow for empirical corrections.




While it is preferable to match TLS trees to census trees, this process is not straightforward and is currently only possible
manually (if at all) after TLS data acquisition and co-registration. Once registered, a slice through the TLS plot-level point
cloud can be generated, enabling the identification of individual trees from their stem profiles. This stem map can be provided
in hard copy or digital format (e.g., high-resolution PDF) to the census team, who can then revisit the plot, moving through it
in the same manner as during the census—starting at the plot's southeast corner or 0,0 and moving up and down by 10 m
quadrants—annotating the TLS stem map with each tree census ID. This process can be conducted separately or as part of an
existing census but is best performed simultaneously or as soon as possible after TLS collection to minimise changes and
facilitate collaboration between TLS and census teams. Despite success with this approach in some plots (e.g., Gabon 2016),
experience has shown that significant understory, terrain variation, and/or changes and tree falls between census and TLS data
collection (e.g., ~2 years between census and TLS data collection for FBRMS-03 plots, and significant tree falls and changes
due to a storm between census and TLS data collection  in FBRMS plot LPG-01 in Gabon) make this process very challenging,
particularly for smaller stems (in the 10-20 cm DBH range).
**3.2 Aligning TLS to UAV-LS data (and other spatial data)**
Through its accurate global registration via PPK processing, UAV-LS can be regarded as a high-quality geometric reference
for registration. For the purpose of comparison with accurate ALS data or satellite observations, a registration of TLS to the
UAV-LS point cloud is highly recommended. The integration of GNSS directly into TLS data collection now ensures that
registered plot-level point clouds are aligned within a global coordinate system. This significantly facilitates the co-registration
of TLS and UAV-LS point clouds, given that GNSS accuracy is typically within 1 metre. Historically, placing all LiDAR point
clouds within accurate global coordinate systems necessitated dedicated survey measurements of plot corners or TLS locations
via GNSS, a process often hindered by signal attenuation in dense forests. Consequently, GNSS surveying of plot corner
locations is not a standard component of forest census protocols, although it should be considered essential for plots intended
for EO calibration and validation purposes. The reduced cost of RTK GNSS equipment and its subsequent routine integration
into TLS workflows have made this more feasible, despite the challenges in obtaining fixed positions, and maintaining radio
link with a base positioned on a well-known point under deep forest canopy cover. While this may not benefit ALS directly,
UAV-LS is likely to serve as a valuable intermediary between TLS (and census data) and ALS. The requirement for global
GNSS positioning also extends to other spatial datasets.
**3.3 Aligning TLS and UAV-LS to ALS data**
ALS presents another challenge; despite the use of high-quality GNSS, m-scale geolocation discrepancies with UAV-LS and
TLS data may still occur. The co-location of LiDAR datasets from different sensors and at varying scales -TLS, UAV-LS, and
ALS- remains challenging, with no standard or 'turn-key' solution available. Manual intervention and processing are often
required, varying for each site and sensor combination. For plot-level estimation of above-ground biomass (AGB), co-location



is less critical, but at finer scales (e.g., for matching to tree-scale census data), this issue can potentially be mitigated through
manual co-registration by identifying common tie points.

ALS point cloud characteristics depend on various acquisition parameters that should be controlled as much as possible. Point
density, point density regularity, and scan angles may be regularised within or across campaigns during post-processing.
Homogeneous scanning density and scanning angles enable the extraction of more stable statistics from the point clouds,
thereby improving AGB prediction performance. To meaningfully compare point clouds across different sites or dates, other
parameters should be kept constant as far as possible. These include the pulse transmitted power (which typically co-varies
with Pulse Repetition Rate) and the flight altitude (which affects pulse irradiance and footprint size, and consequently, LiDAR
pulse penetration) (Vincent et al., 2023).

## 4. Recommendations for data collection in FBRMS

Building on this first case study, we make the following general recommendations for data collection of tropical forest plot
census, TLS, UAV-LS and ALS data for the specific application of estimating AGB and upscaling to Earth Observation
estimates. These recommendations follow from the CEOS LPV AGB protocol and subsequent requirements identified for the
GEO-TREES initiative.
● **Consistent data acquisition and processing:** in order to facilitate the comparison of AGB estimates between sites,

dates, teams, etc. care should be taken to collect and process data as consistently as possible. This may sound obvious

but is particularly important as the use of TLS and UAV-LS for AGB estimation (and even ALS in some cases) are

currently primarily research-led (as opposed to fully operational). As new methods and tools are developed, including

newer versions of existing software, care should be taken to ensure backwards compatibility of the resulting AGB

estimates. This means either re-processing older data, or at the very least, some form of cross-comparison of original

and new methods. In our experience, listed below are some of the areas where care is needed to ensure data

consistency and reduce bias and uncertainty:

● **TLS data acquisition** - comparison between sites and plots is made much easier by using the same census,

TLS, UAV-LS and ALS data acquisition and processing protocols. Even within the forest plot census

community there are slightly different protocols and processes between different plot networks. This is even

more variable for different types of LiDAR data. We note that much of the TLS work in tropical forests

aimed at volume reconstruction and AGB estimation has been carried out with RIEGL VZ series TLS

instruments. We make no comment as to what is 'the best' instrument - there are various cost/benefit trade-

offs to be made. Equipment has to be robust to withstand tropical forest work (and humidity). LiDAR range

needs to be in the 100s of metres to ensure points are returned from tall canopies. Phase-shift TLS systems

can be light and have very rapid scan rates, but suffer from 'ghosting' of multiple returned hits along a beam

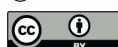



| 693 | path. Mobile Laser Scanning (MLS) systems offer rapid coverage, and require minimal input for registration |
| 694 | by using simultaneous location and mapping (SLAM), but tend to have lower range and precision due to the |
| 695 | uncertainty in absolute location resulting from SLAM. It is likely that these systems will become more |
| 696 | powerful and precise, offering a possible alternative to static tripod-mounted TLS in the future for AGB |
| 697 | applications. Specific issues to consider are TLS power. For example, the RIEGL VZ-400 and newer VZ- |
| 698 | 400i systems (both used here) have different recording sensitivities i.e. down to -30 dB for the newer VZ- |
| 699 | 400i, whereas the VZ-400 only recorded to -20 dB. This can have a significant impact on the number of |
| 700 | returns, particularly from further away and higher in the canopy and should be taken into consideration when |
| 701 | comparing results between older and newer TLS instruments. Choices are also possible in terms of power |
| 702 | settings: lower power settings reduce scan times & extend battery time, but also significantly reduce the |
| 703 | quality of resulting point clouds, particularly higher in the canopy. Here, TLS data was collected using the |
| 704 | highest LiDAR power (300 kHz) for RIEGL scanners VZ-400 and VZ-400i, trading off longer scan times |
| 705 | for a fixed angular resolution to maximise coverage at the tops of tall trees. However, recent work by |
| 706 | Verheltz et al. (2024) suggests that using lower power, but with higher angular resolution, can achieve better |
| 707 | coverage in tall forests for the same scan duration (3 mins per scan). More generally, comparing |
| 708 | measurements made with scanners of varying power, sensitivity, resolution etc. will compound uncertainties |
| 709 | (particularly biases) in the resulting estimates of AGB and so should be avoided or minimised as far as |
| 710 | possible. This is particularly important for large-scale site-to-site comparison required for EO biomass |
| 711 | product cal/val (e.g. for global FBRMS comparisons). |
| 712 | ● **TLS processing -** broadly, TLS data acquisition and processing in tropical forests has gradually converged |
| 713 | towards something of a consensus, albeit this is still an active area of research and will vary depending on |
| 714 | the team, site and application. Specific issues to consider are the way in which trees are extracted from plot- |
| 715 | scale point clouds. Currently, the most accurate method for doing this is by manual clearing of each tree |
| 716 | using a tool such as CloudCompare (h). However, this is a time-consuming and somewhat subjective process |
| 717 | that is not fully replicable - different people will produce slightly different results. Automated pipelines using |
| 718 | machine learning/deep learning (ML/DL) offer a more rapid and repeatable approach (e.g. Krisanski et al., |
| 719 | 2021; Wilkes et al., 2023), however, their resulting tree extraction accuracy is harder to assess given that the |
| 720 | 'true' structure of trees is unknown. Manually-extracted trees can be used to assess automated tree extraction |
| 721 | accuracy, as well as forming the training data to enable improvements in the underlying ML/DL approaches. |
| 722 | Developing locally-trained / optimised ML/DL models is likely to improve this approach further. Moving |
| 723 | from individual tree point clouds to volume estimates it is also important to use consistent QSM-fitting |
| 724 | approaches. For example, there are systematic differences between older and newer versions of TreeQSM, |
| 725 | currently the most widely-used QSM fitting software (Demol et al., 2024; Raumonen et al., 2013). |



Quantifying the uncertainty in tree-level estimates of volume will depend on this processing chain, which
will then determine the plot-level uncertainty when upscaling.
• **UAV-LS acquisition and processing -** due to the wide range of platforms and LiDAR payloads being used
(as well as local UAV and safety regulations), there is currently little consensus in terms of both acquisition
and processing of UAV-LS data. There are a wide range of flight choices (particularly altitude), instrument
settings (scan angle), and survey systems (overlap, duration, etc.) that are a function of platform
performance, cost, etc. The impact of some of these choices is discussed in Brede et al. (2022b) where the
benefits of higher power, multiple returns and overlapping flights in detecting canopy structure are
highlighted. UAV-LS is not a like-for-like replacement for TLS, thus, the ability to compare these two
different sources of LiDAR data will be facilitated by accurate geo-location (see above). This can be
achieved by using ground targets with surveyed locations that can be identified in the UAV-LS data (e.g.
reflective sheets/tarps, umbrellas, commercial UAV targets etc). This presupposes that there are sufficient
gaps in the canopy for targets to be seen, which is not always true. During data collection attention should
be paid to also either have access to GNSS observables from permanent base stations (e.g. CORS network)
or collect observables with a temporary base station (e.g. Emlid Reach RS+ or RS2). A base station should
be positioned less than 15 km away from the survey area. An important consideration for UAV-LS data
collection is whether visual line of sight (VLOS) needs to be maintained, i.e. visibility of the platform by the
pilot during the whole mission. If so, this can impact the choice of take-off, flight plan, etc. which in turn
may influence the choice of platform. Fixed-wing platforms have a much greater area coverage and flight
duration than VTOL platforms, but by necessity, must operate beyond VLOS (BVLOS). They also require
far more space to take off and land than VTOL platforms.
• **ALS acquisition and processing -** while ALS has been used operationally for forest applications for several
decades, its application for AGB estimates specifically is still less well-defined. In particular, this is true
when considering tree-scale rather than plot-level estimates. Practically, ALS surveys are almost always
outsourced (from the plot PIs, census and TLS, UAV teams) to commercial or agency (e.g. NASA, ESA,
NERC) providers. In the former case, there may be limited input from the end user over the platform,
instrument and acquisition parameters, or the way in which the data are processed to the resulting final
delivery. In ESA, NERC, NASA acquisitions, there tends to be more input from the users, but there may be
other restrictions in terms of when and where flights can be made. We recommend a pulse density of 10 m$^{-2}$
or higher and a swath angle of +/-15 degrees or smaller. Most importantly, consistency over time of the
other acquisition parameters should be sought to enable meaningful temporal analysis of ALS point cloud.
In most cases, the 3D point cloud will be processed to generate a 2D canopy height model for further analysis.
This post-processing can have important effects on the results, we therefore, recommend users follow a
standardized procedure such as Fischer et al. (2024).





•   **Accurate (cm-scale) GNSS locations for 1ha FBRMS plot corners (or at the least the nominal origin 0, 0**
**coordinate for each plot):** this makes comparison and merging of any subsequent measurements much easier. It is
important to note that this is not a standard requirement of forest census measurements and requires specialist
surveying equipment e.g. GNSS RTK base station + rover configuration. It is also challenging under heavy forest
cover. Given that such setups are required (ideally) for TLS and UAV-LS, plot corner surveying is potentially best
carried out by these teams.
•   **Linking TLS trees to their census counterparts:** ideally, a permanent 10 x 10m subplot grid would be established
within each 1 ha forest plot. Census teams can then follow the same chain sampling pattern used in TLS data collection
(see Figure 2.1.4b & c) and identify the tree IDs found within each 10 x 10 m quadrants as they move through the
plot. However, placing a 10 x 10 m sub-grid is not always straightforward (or even desirable) as it may require rebar
posts, which can be expensive and are likely to be removed or damaged by e.g. elephants in West African plots
particularly. An alternative approach is to label some trees with temporary numbered QR-type markers that can be
read automatically from the lidar point cloud data. The markers can be printed on A4 waterproof paper, attached to
trees with known census ID, and then identified in the TLS data using a tool such as qrDAR (Wilkes et al., 2017). If
the 20 or so largest trees are labelled in this way, distributed across a 1 ha plot, this makes subsequent tree matching
between census and TLS data much easier as there are known 'anchor trees' for the survey team to work from.
**5. Data Access**
All LiDAR ForestScan datasets and one tree census dataset are freely available from the CEDA archive
(https://archive.ceda.ac.uk) via the listed DOIs in Table 8a (Chavana-Bryant et al., 2025b). As previously mentioned in section
2.2.1, tree census data for FBRMS plots in Paracou, French Guiana is available as a data package via
https://dataverse.cirad.fr/dataset.xhtml?persistentId=doi:10.18167/DVN1/94XHID (Derroire et al., 2025). Tree census data
for FBRMS plots in Gabon and Malaysia are available as a data package via https://doi.org/10.5521/forestplots.net/2025_2
(Chavana-Bryant et al., 2025a). Access, licensing and citation details for ForestScan tree inventory/census datasets: FBRMS-
01 (French Guiana), FBRMS-02 (Gabon) and FBRMS-03 (Malaysian Borneo) are provided in Table 8b.

**Table 8a:** Access, licensing and citation details for 17 LiDAR and one tree inventory/census ForestScan dataset available in
the CEDA archive. When using any of the ForestScan datasets, this paper must also be cited.

| ForestScan Datasets / Data license type | URLs | DOIs | Citable as |
|---|---|---|---|
| ForestScan Collection | https://catalogue.ceda.ac.uk/uuid/88a8620229014e0ebacf0606b302112d | 10.5285/88a862022 9014e0ebacf060 6b302112d | Chavana-Bryant, C.; Wilkes, P.; Yang, W.; Burt, A.; Vines, P.; Bennett, A.C.; Pickavance, G.C.; Cooper, D.L.M.; Lewis, S.L.; Phillips, O.L.; Brede, B.; Lau, A.; |

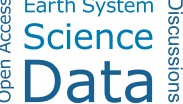

| | | | Herold, M.; McNicol, I.M.; Mitchard, E.T.A.; Coombes, D.; Jackson, T.D.; Makaga, L.; Milamizokou Napo, H.O.; Ngomanda, A.; Ntie, S.; Medjibe, V.; Dimbonda, P.; Soenens, L.; Daelemans, V.; Proux, L.; Nilus, R.; Labrière, N.; Jeffery, K.; Burslem, D.F.R.P.; Clewley, D.; Moffat, D.; Qie, L.; Bartholomeus, H.; Vincent, G.; Barbier, N.; Derroire, G.; Abernethy, K.; Scipal, K.; Disney, M. (2025): ForestScan Collection. NERC EDS Centre for Environmental Data Analysis, *20 January 2025*. DOI:10.5285/88a8620229014e0ebacf0606 b302112d. https://dx.doi.org/10.5285/88a8620229014e0ebacf0606b302112d |
|---|---|---|---|
| ForestScan Project: Terrestrial Laser Scanning (TLS) of FBRMS-01: Paracou, French Guiana 1ha plot FG5c1, September to October 2022<br><br>License type: CC BY 4.0 http://creativecommons.org/licenses/by/4.0/ | https://catalogue.ceda.ac.uk/uuid/656ac8ee1d42443f9addcbce28c1b137 | 10.5285/656ac8ee1d42443f9addcbce28c1b137 | Chavana-Bryant, C.; Wilkes, P.; Yang, W.; Burt, A.; Vines, P.; Bennett, A.C.; Pickavance, G.C.; Cooper, D.L.M.; Lewis, S.L.; Phillips, O.L.; Brede, B.; Lau, A.; Herold, M.; McNicol, I.M.; Mitchard, E.T.A.; Coombes, D.; Jackson, T.D.; Makaga, L.; Milamizokou Napo, H.O.; Ngomanda, A.; Ntie, S.; Medjibe, V.; Dimbonda, P.; Soenens, L.; Daelemans, V.; Proux, L.; Nilus, R.; Labrière, N.; Jeffery, K.; Burslem, D.F.R.P.; Clewley, D.; Moffat, D.; Qie, L.; Bartholomeus, H.; Vincent, G.; Barbier, N.; Derroire, G.; Abernethy, K.; Scipal, K.; Disney, M. (2025): ForestScan Project : Terrestrial Laser Scanning (TLS) of FBRMS-01: Paracou, French Guiana 1ha plot FG5c1, September to October 2022. NERC EDS Centre for Environmental Data Analysis, *28 March 2025*. DOI:10.5285/656ac8ee1d42443f9addcbce28c1b137. https://dx.doi.org/10.5285/656ac8ee1d42443f9addcbce28c1b137 |
| ForestScan Project: Terrestrial Laser Scanning (TLS) of FBRMS-01: Paracou, French Guiana 1ha plot FG6c2, September to October 2022<br><br>License type: CC BY 4.0 http://creativecommons.org/licenses/by/4.0/ | https://catalogue.ceda.ac.uk/uuid/931973db09af41568853702efe135f29 | 10.5285/931973db09af415688537 02efe135f29 | Chavana-Bryant, C.; Wilkes, P.; Yang, W.; Burt, A.; Vines, P.; Bennett, A.C.; Pickavance, G.C.; Cooper, D.L.M.; Lewis, S.L.; Phillips, O.L.; Brede, B.; Lau, A.; Herold, M.; McNicol, I.M.; Mitchard, E.T.A.; Coombes, D.; Jackson, T.D.; Makaga, L.; Milamizokou Napo, H.O.; Ngomanda, A.; Ntie, S.; Medjibe, V.; Dimbonda, P.; Soenens, L.; Daelemans, |





| | | | V.; Proux, L.; Nilus, R.; Labrière, N.; Jeffery, K.; Burslem, D.F.R.P.; Clewley, D.; Moffat, D.; Qie, L.; Bartholomeus, H.; Vincent, G.; Barbier, N.; Derroire, G.; Abernethy, K.; Scipal, K.; Disney, M. (2025): ForestScan Project : Terrestrial Laser Scanning (TLS) of FBRMS-01: Paracou, French Guiana 1ha plot FG6c2, September to October 2022. NERC EDS Centre for Environmental Data Analysis, *28 March 2025*. DOI:10.5285/931973db09af41568853702efe135f29. https://dx.doi.org/10.5285/931973db09af41568853702efe135f29 |
|---|---|---|---|
| ForestScan Project: Terrestrial Laser Scanning (TLS) of FBRMS-01: Paracou, French Guiana 1ha plot FG8c4, September to October 2022<br><br>License type: CC BY 4.0 http://creativecommons.org/licenses/by/4.0/ | https://catalogue.ceda.ac.uk/uuid/40f0f38023ac40f6b40bbf96e4dc5258 | 10.5285/40f0f38023ac40f6b40bbf96e4dc5258 | Chavana-Bryant, C.; Wilkes, P.; Yang, W.; Burt, A.; Vines, P.; Bennett, A.C.; Pickavance, G.C.; Cooper, D.L.M.; Lewis, S.L.; Phillips, O.L.; Brede, B.; Lau, A.; Herold, M.; McNicol, I.M.; Mitchard, E.T.A.; Coombes, D.; Jackson, T.D.; Makaga, L.; Milamizokou Napo, H.O.; Ngomanda, A.; Ntie, S.; Medjibe, V.; Dimbonda, P.; Soenens, L.; Daelemans, V.; Proux, L.; Nilus, R.; Labrière, N.; Jeffery, K.; Burslem, D.F.R.P.; Clewley, D.; Moffat, D.; Qie, L.; Bartholomeus, H.; Vincent, G.; Barbier, N.; Derroire, G.; Abernethy, K.; Scipal, K.; Disney, M. (2025): ForestScan Project : Terrestrial Laser Scanning (TLS) of FBRMS-01: Paracou, French Guiana 1ha plot FG8c4, September to October 2022. NERC EDS Centre for Environmental Data Analysis, *28 March 2025*. DOI:10.5285/40f0f38023ac40f6b40bbf96e4dc5258. https://dx.doi.org/10.5285/40f0f38023ac40f6b40bbf96e4dc5258 |
| ForestScan Project : Terrestrial Laser Scanning (TLS) of FBRMS-02: Station d'Etudes des Gorilles et Chimpanzés, Lopé National Park, Gabon 1ha plot LPG-01, June to July 2022<br><br>License type: CC BY 4.0 http://creativecommons.org/licenses/by/4.0/ | https://catalogue.ceda.ac.uk/uuid/8ea2c697ee53430a84825384bfdcf06a/ | 8ea2c697ee53430a84825384bfdcf06a | Chavana-Bryant, C.; Wilkes, P.; Yang, W.; Burt, A.; Vines, P.; Bennett, A.C.; Pickavance, G.C.; Cooper, D.L.M.; Lewis, S.L.; Phillips, O.L.; Brede, B.; Lau, A.; Herold, M.; McNicol, I.M.; Mitchard, E.T.A.; Coombes, D.; Jackson, T.D.; Makaga, L.; Milamizokou Napo, H.O.; Ngomanda, A.; Ntie, S.; Medjibe, V.; Dimbonda, P.; Soenens, L.; Daelemans, V.; Proux, L.; Nilus, R.; Labrière, N.; Jeffery, K.; Burslem, D.F.R.P.; Clewley, |



| | | | D.; Moffat, D.; Qie, L.; Bartholomeus, H.; Vincent, G.; Barbier, N.; Derroire, G.; Abernethy, K.; Scipal, K.; Disney, M. (2025): ForestScan Project : Terrestrial Laser Scanning (TLS) of FBRMS-02: Station d'Etudes des Gorilles et Chimpanzés, Lopé National Park, Gabon 1ha plot LPG-01, June to July 2022. NERC EDS Centre for Environmental Data Analysis, *28 March 2025*. DOI:10.5285/8ea2c697ee53430a84825384 bfdcf06a. https://dx.doi.org/10.5285/8ea2c 697ee53430a84825384bfdcf06a |
|---|---|---|---|
| ForestScan Project : Terrestrial Laser Scanning (TLS) of FBRMS-02: Station d'Etudes des Gorilles et Chimpanzés, Lopé National Park, Gabon 1ha plot OKO-01, June to July 2022<br><br>License type: CC BY 4.0 http://creativecommons.org/licens es/by/4.0/ | https://catalogue.c eda.ac.uk/uuid/45 ae3437f82f4e4fb7 5f9a5c26a194ba | 10.5285/45ae3437 f82f4e4fb75f9a5c 26a194ba | Chavana-Bryant, C.; Wilkes, P.; Yang, W.; Burt, A.; Vines, P.; Bennett, A.C.; Pickavance, G.C.; Cooper, D.L.M.; Lewis, S.L.; Phillips, O.L.; Brede, B.; Lau, A.; Herold, M.; McNicol, I.M.; Mitchard, E.T.A.; Coombes, D.; Jackson, T.D.; Makaga, L.; Milamizokou Napo, H.O.; Ngomanda, A.; Ntie, S.; Medjibe, V.; Dimbonda, P.; Soenens, L.; Daelemans, V.; Proux, L.; Nilus, R.; Labrière, N.; Jeffery, K.; Burslem, D.F.R.P.; Clewley, D.; Moffat, D.; Qie, L.; Bartholomeus, H.; Vincent, G.; Barbier, N.; Derroire, G.; Abernethy, K.; Scipal, K.; Disney, M. (2025): ForestScan Project : Terrestrial Laser Scanning (TLS) of FBRMS-02: Station d'Etudes des Gorilles et Chimpanzés, Lopé National Park, Gabon 1ha plot OKO-01, June to July 2022. NERC EDS Centre for Environmental Data Analysis, *28 March 2025*. DOI:10.5285/45ae3437f82f4e4fb75f9a5c2 6a194ba. https://dx.doi.org/10.5285/45ae3 437f82f4e4fb75f9a5c26a194ba |
| ForestScan Project : Terrestrial Laser Scanning (TLS) of FBRMS-02: Station d'Etudes des Gorilles et Chimpanzés, Lopé National Park, Gabon 1ha plot OKO-02, June to July 2022<br><br>License type: CC BY 4.0 http://creativecommons.org/licens es/by/4.0/ | https://catalogue.c eda.ac.uk/uuid/ff4 b43475c9641cca1 dad2c8be8dadaf | 10.5285/ff4b4347 5c9641cca1dad2c 8be8dadaf | Chavana-Bryant, C.; Wilkes, P.; Yang, W.; Burt, A.; Vines, P.; Bennett, A.C.; Pickavance, G.C.; Cooper, D.L.M.; Lewis, S.L.; Phillips, O.L.; Brede, B.; Lau, A.; Herold, M.; McNicol, I.M.; Mitchard, E.T.A.; Coombes, D.; Jackson, T.D.; Makaga, L.; Milamizokou Napo, H.O.; Ngomanda, A.; Ntie, S.; Medjibe, V.; Dimbonda, P.; Soenens, L.; Daelemans, V.; Proux, L.; Nilus, R.; Labrière, N.; Jeffery, K.; Burslem, D.F.R.P.; Clewley, |



| | | | D.; Moffat, D.; Qie, L.; Bartholomeus, H.; Vincent, G.; Barbier, N.; Derroire, G.; Abernethy, K.; Scipal, K.; Disney, M. (2025): ForestScan Project : Terrestrial Laser Scanning (TLS) of FBRMS-02: Station d'Etudes des Gorilles et Chimpanzés, Lopé National Park, Gabon 1ha plot OKO-02, June to July 2022. NERC EDS Centre for Environmental Data Analysis, *28 March 2025*. DOI:10.5285/ff4b43475c9641cca1dad2c8be8dadaf. https://dx.doi.org/10.5285/ff4b43475c9641cca1dad2c8be8dadaf |
|---|---|---|---|
| ForestScan Project : Terrestrial Laser Scanning (TLS) of FBRMS-02: Station d'Etudes des Gorilles et Chimpanzés, Lopé National Park, Gabon 1ha plot OKO-03, June to July 2022<br><br>License type: CC BY 4.0 http://creativecommons.org/licenses/by/4.0/ | https://catalogue.ceda.ac.uk/uuid/8ed3ddec76b8470285bdb2ea643f54bc | 10.5285/8ed3ddec76b8470285bdb2ea643f54bc | Chavana-Bryant, C.; Wilkes, P.; Yang, W.; Burt, A.; Vines, P.; Bennett, A.C.; Pickavance, G.C.; Cooper, D.L.M.; Lewis, S.L.; Phillips, O.L.; Brede, B.; Lau, A.; Herold, M.; McNicol, I.M.; Mitchard, E.T.A.; Coombes, D.; Jackson, T.D.; Makaga, L.; Milamizokou Napo, H.O.; Ngomanda, A.; Ntie, S.; Medjibe, V.; Dimbonda, P.; Soenens, L.; Daelemans, V.; Proux, L.; Nilus, R.; Labrière, N.; Jeffery, K.; Burslem, D.F.R.P.; Clewley, D.; Moffat, D.; Qie, L.; Bartholomeus, H.; Vincent, G.; Barbier, N.; Derroire, G.; Abernethy, K.; Scipal, K.; Disney, M. (2025): ForestScan Project : Terrestrial Laser Scanning (TLS) of FBRMS-02: Station d'Etudes des Gorilles et Chimpanzés, Lopé National Park, Gabon 1ha plot OKO-03, June to July 2022. NERC EDS Centre for Environmental Data Analysis, *28 March 2025*. DOI:10.5285/8ed3ddec76b8470285bdb2ea643f54bc. https://dx.doi.org/10.5285/8ed3ddec76b8470285bdb2ea643f54bc |
| ForestScan Project : Terrestrial Laser Scanning (TLS) of FBRMS-03: Kabili-Sepilok, Malaysian Borneo 1ha plot SEP-11, March 2017<br><br>License type: CC BY 4.0 http://creativecommons.org/licenses/by/4.0/ | https://catalogue.ceda.ac.uk/uuid/37b039605e9b4bb5a89371fd7f5b7ba1 | 37b039605e9b4bb5a89371fd7f5b7ba1 | Chavana-Bryant, C.; Wilkes, P.; Yang, W.; Burt, A.; Vines, P.; Bennett, A.C.; Pickavance, G.C.; Cooper, D.L.M.; Lewis, S.L.; Phillips, O.L.; Brede, B.; Lau, A.; Herold, M.; McNicol, I.M.; Mitchard, E.T.A.; Coombes, D.; Jackson, T.D.; Makaga, L.; Milamizokou Napo, H.O.; Ngomanda, A.; Ntie, S.; Medjibe, V.; Dimbonda, P.; Soenens, L.; Daelemans, V.; Proux, L.; Nilus, R.; Labrière, N.; Jeffery, K.; Burslem, D.F.R.P.; Clewley, |





| | | | |
|---|---|---|---|
| | | | D.; Moffat, D.; Qie, L.; Bartholomeus, H.; Vincent, G.; Barbier, N.; Derroire, G.; Abernethy, K.; Scipal, K.; Disney, M. (2025): ForestScan Project : Terrestrial Laser Scanning (TLS) of FBRMS-03: Kabili-Sepilok, Malaysian Borneo 1ha plot SEP-11, March 2017. NERC EDS Centre for Environmental Data Analysis, *28 March 2025*. DOI:10.5285/37b039605e9b4bb5a89371fd 7f5b7ba1. https://dx.doi.org/10.5285/37b0 39605e9b4bb5a89371fd7f5b7ba1 |
| ForestScan Project : Terrestrial Laser Scanning (TLS) of FBRMS-03: Kabili-Sepilok, Malaysian Borneo 1ha plot SEP-12, March 2017<br><br>License type: CC BY 4.0 http://creativecommons.org/licens es/by/4.0/ | https://catalogue.c eda.ac.uk/uuid/bb 81c82352524df99 ddd411f6ca2ec81 | bb81c82352524df 99ddd411f6ca2ec 81 | Chavana-Bryant, C.; Wilkes, P.; Yang, W.; Burt, A.; Vines, P.; Bennett, A.C.; Pickavance, G.C.; Cooper, D.L.M.; Lewis, S.L.; Phillips, O.L.; Brede, B.; Lau, A.; Herold, M.; McNicol, I.M.; Mitchard, E.T.A.; Coombes, D.; Jackson, T.D.; Makaga, L.; Milamizokou Napo, H.O.; Ngomanda, A.; Ntie, S.; Medjibe, V.; Dimbonda, P.; Soenens, L.; Daelemans, V.; Proux, L.; Nilus, R.; Labrière, N.; Jeffery, K.; Burslem, D.F.R.P.; Clewley, D.; Moffat, D.; Qie, L.; Bartholomeus, H.; Vincent, G.; Barbier, N.; Derroire, G.; Abernethy, K.; Scipal, K.; Disney, M. (2025): ForestScan Project : Terrestrial Laser Scanning (TLS) of FBRMS-03: Kabili-Sepilok, Malaysian Borneo 1ha plot SEP-12, March 2017. NERC EDS Centre for Environmental Data Analysis, *28 March 2025*. DOI:10.5285/bb81c82352524df99ddd411f 6ca2ec81. https://dx.doi.org/10.5285/bb81c 82352524df99ddd411f6ca2ec81 |
| ForestScan Project: Terrestrial Laser Scanning (TLS) of FBRMS-03: Kabili-Sepilok, Malaysian Borneo 1ha plot SEP-30, March 2017<br><br>License type: CC BY 4.0 http://creativecommons.org/licens es/by/4.0/ | https://catalogue.c eda.ac.uk/uuid/ff2 17c783e3f4c66a4 891d2b5807ee6e | ff217c783e3f4c66 a4891d2b5807ee6 e | Chavana-Bryant, C.; Wilkes, P.; Yang, W.; Burt, A.; Vines, P.; Bennett, A.C.; Pickavance, G.C.; Cooper, D.L.M.; Lewis, S.L.; Phillips, O.L.; Brede, B.; Lau, A.; Herold, M.; McNicol, I.M.; Mitchard, E.T.A.; Coombes, D.; Jackson, T.D.; Makaga, L.; Milamizokou Napo, H.O.; Ngomanda, A.; Ntie, S.; Medjibe, V.; Dimbonda, P.; Soenens, L.; Daelemans, V.; Proux, L.; Nilus, R.; Labrière, N.; Jeffery, K.; Burslem, D.F.R.P.; Clewley, D.; Moffat, D.; Qie, L.; Bartholomeus, H.; Vincent, G.; Barbier, N.; Derroire, G.; |



| | | | Abernethy, K.; Scipal, K.; Disney, M. (2025): ForestScan Project : Terrestrial Laser Scanning (TLS) of FBRMS-03: Kabili-Sepilok, Malaysian Borneo 1ha plot SEP-30, March 2017. NERC EDS Centre for Environmental Data Analysis, *28 March 2025*. DOI:10.5285/ff217c783e3f4c66a4891d2b5 807ee6e. https://dx.doi.org/10.5285/ff217c 783e3f4c66a4891d2b5807ee6e |
|---|---|---|---|
| ForestScan: Terrestrial Laser Scanning (TLS) of FBRMS-01: Paracou, French Guiana 1ha plot IRD-CNES (Tropiscat), October 2021<br><br>License type: CC BY 4.0<br>http://creativecommons.org/licens es/by/4.0/ | https://catalogue.c eda.ac.uk/uuid/b1 cd34f6af7941a3b1 429ac52a3f6b28 | 10.5285/b1cd34f6 af7941a3b1429ac 52a3f6b28 | Vincent, G.; Villard, L. (2025): ForestScan: Terrestrial Laser Scanning (TLS) of FBRMS-01: Paracou, French Guiana 1ha plot IRD-CNES, October 2021. NERC EDS Centre for Environmental Data Analysis, *28 March 2025*. DOI:10.5285/b1cd34f6af7941a3b1429ac5 2a3f6b28. https://dx.doi.org/10.5285/b1cd3 4f6af7941a3b1429ac52a3f6b28 |
| ForestScan Project: Unpiloted Aerial Vehicle LiDAR Scanning (UAV-LS) and Terrestrial Laser Scanning (TLS) data of FBRMS-01: Paracou, French Guiana plot 6, 10th October to 15th November 2019<br><br>License type: CC BY 4.0<br>http://creativecommons.org/licens es/by/4.0/ | https://catalogue.c eda.ac.uk/uuid/32 5a4dde60d142049 339e0c84816aac1 | 10.5285/325a4dde 60d142049339e0c 84816aac1 | Brede, B.; Barbier, N.; Bartholomeus, H.; Derroire, G.; Lau, A.; Lusk, D.; Herold, M. (2025): ForestScan Project: Unpiloted Aerial Vehicle LiDAR Scanning (UAV-LS) and Terrestrial Laser Scanning (TLS) data of FBRMS-01: Paracou, French Guiana plot 6, 10th October to 15th November 2019. NERC EDS Centre for Environmental Data Analysis, *28 March 2025*. DOI:10.5285/325a4dde60d142049339e0c8 4816aac1. https://dx.doi.org/10.5285/325a 4dde60d142049339e0c84816aac1 |
| ForestScan Project: Multiple Unpiloted Aerial Vehicle LiDAR Scanning (UAV-LS) data acquisitions of FBRMS-01: Paracou, French Guiana, plots 4, 5, 6, 8, IRD-CNES (Tropiscat) and Flux-Tower area, October 2019<br><br>License type: CC BY 4.0<br>http://creativecommons.org/licens es/by/4.0/ | https://catalogue.c eda.ac.uk/uuid/00 5f2e0aebc24ed98a 9772a0ba3798e2 | 10.5285/005f2e0a ebc24ed98a9772a 0ba3798e2 | Barbier, N.; Vincent, G. (2025): ForestScan Project: Multiple Unpiloted Aerial Vehicle LiDAR Scanning (UAV-LS) data acquisitions of FBRMS-01: Paracou, French Guiana, plots 4, 5, 6, 8, IRD-CNES and Flux-Tower area, October 2019. NERC EDS Centre for Environmental Data Analysis, *28 March 2025*. DOI:10.5285/005f2e0aebc24ed98a9772a0 ba3798e2. https://dx.doi.org/10.5285/005f2 e0aebc24ed98a9772a0ba3798e2 |



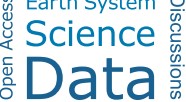

| | | | |
|---|---|---|---|
| ForestScan project: Unpiloted Aerial Vehicle LiDAR Scanning (UAV-LS) data of FBRMS-02: Station d'Etudes des Gorilles et Chimpanzés, Lopé National Park, Gabon, June 2022<br><br>License type: CC BY 4.0<br>http://creativecommons.org/licenses/by/4.0/ | https://catalogue.ceda.ac.uk/uuid/7a4649cabd3e4afb8cd31cfd7d95ac8e | 10.5285/7a4649cabd3e4afb8cd31cfd7d95ac8e | McNicol, I.M.; Mitchard, E.T.A. (2025): ForestScan project: Unpiloted Aerial Vehicle LiDAR Scanning (UAV-LS) data of FBRMS-02: Station d'Etudes des Gorilles et Chimpanzés, Lopé National Park, Gabon, June 2022. NERC EDS Centre for Environmental Data Analysis, *28 March 2025*. DOI:10.5285/7a4649cabd3e4afb8cd31cfd7d95ac8e. https://dx.doi.org/10.5285/7a4649cabd3e4afb8cd31cfd7d95ac8e |
| ForestScan: Aerial Laser Scanning (ALS) of FBRMS-01: Paracou, French Guiana, November 2022<br><br>License type: CC BY 4.0<br>http://creativecommons.org/licenses/by/4.0/ | https://catalogue.ceda.ac.uk/uuid/7bef89a9dc404683a46642625a024a4b | 10.5285/7bef89a9dc404683a46642625a024a4b | Vincent, G. (2025): ForestScan: Aerial Laser Scanning (ALS) of FBRMS-01: Paracou, French Guiana, November 2022. NERC EDS Centre for Environmental Data Analysis, *28 March 2025*. DOI:10.5285/7bef89a9dc404683a46642625a024a4b. https://dx.doi.org/10.5285/7bef89a9dc404683a46642625a024a4b |
| Aerial LiDAR French Guiana Paracou, November 2019<br><br>License type: CC BY 4.0<br>http://creativecommons.org/licenses/by/4.0/ | https://catalogue.ceda.ac.uk/uuid/1d554ff41c104491ac3661c6f6f52aab | 10.5285/1d554ff41c104491ac3661c6f6f52aab | Jackson, T.D.; Vincent, G.; Coomes, D.A. (2023): Aerial LiDAR data from French Guiana, Paracou, November 2019. NERC EDS Centre for Environmental Data Analysis, *20 December 2023*. DOI:10.5285/1d554ff41c104491ac3661c6f6f52aab. https://dx.doi.org/10.5285/1d554ff41c104491ac3661c6f6f52aab |
| Aerial LiDAR French Guiana Nouragues, November 2019<br><br>License type: CC BY 4.0<br>http://creativecommons.org/licenses/by/4.0/ | https://catalogue.ceda.ac.uk/uuid/7bdc5bfc06264802be34f918597150e8 | 10.5285/7bdc5bfc06264802be34f918597150e8 | Jackson, T.D.; Vincent, G.; Coomes, D.A. (2023): Aerial LiDAR data from French Guiana, Nouragues, November 2019. NERC EDS Centre for Environmental Data Analysis, *20 December 2023*. DOI:10.5285/7bdc5bfc06264802be34f918597150e8. https://dx.doi.org/10.5285/7bdc5bfc06264802be34f918597150e8 |
| Airborne LiDAR and RGB imagery from Sepilok Reserve and Danum Valley in Malaysia in 2020<br><br>License type: OGL UK 3.0<br>https://www.nationalarchives.gov.uk/doc/open-government-licence/version/3/ | https://catalogue.ceda.ac.uk/uuid/dd4d20c8626f4b9d99bc14358b1b50fe | 10.5285/dd4d20c8626f4b9d99bc14358b1b50fe | Coomes, D.A.; Jackson, T.D. (2022): Airborne LiDAR and RGB imagery from Sepilok Reserve and Danum Valley in Malaysia in 2020. NERC EDS Centre for Environmental Data Analysis, *03 October 2022*. DOI:10.5285/dd4d20c8626f4b9d99bc14358b1b50fe. https://dx.doi.org/10.5285/dd4d20c8626f4b9d99bc14358b1b50fe |





| | | | |
|---|---|---|---|
| ForestScan: Tree census data (diameter and species name) of FBRMS-01: Paracou, French Guiana 1ha plot IRD-CNES (Tropiscat), October 2021<br><br>License type: CC BY 4.0 http://creativecommons.org/licenses/by/4.0/ | https://catalogue.ceda.ac.uk/uuid/5e78ff91e9cd4143bfa3b7358efd2607 | 10.5285/5e78ff91e9cd4143bfa3b7358efd2607 | Vincent, G.; Martin, O.; Engel, F. (2025): ForestScan: Tree census data (diameter and species name) of FBRMS-01: Paracou, French Guiana 1ha plot IRD-CNES, October 2021. NERC EDS Centre for Environmental Data Analysis, *28 March 2025*. DOI:10.5285/5e78ff91e9cd4143bfa3b7358efd2607. https://dx.doi.org/10.5285/5e78ff91e9cd4143bfa3b7358efd2607 |


Table 8b: Access, licensing and citation details for ForestScan tree inventory/census datasets: FBRMS-01 (French Guiana), FBRMS-02 (Gabon) and FBRMS-03 (Malaysian Borneo). When using any of the ForestScan datasets, this paper must also be cited.

These datasets are provided as curated data packages made available by the ForestPlots consortium and the French Agricultural Research Centre for International Development (CIRAD) open-access portal – DataVerse. Both archival platforms operate under a fair use policy, governed by the Creative Commons Attribution-NonCommercial-ShareAlike 4.0 International Licence (CC BY-NC-SA 4.0) (see https://forestplots.net/en/join-forestplots/working-with-data and https://dataverse.org/best-practices/dataverse-community-norms). These policies reflect a strong commitment to equitable and inclusive data collection, funding, and sharing practices, as outlined in the ForestPlots code of conduct (https://forestplots.net/en/join-forestplots/code-of-conduct).

Tropical forest plot census data provide unique insights into forest structure and dynamics but are challenging and often hazardous to collect, requiring sustained investment and logistical support in remote regions with limited infrastructure. A persistent challenge to equitable research is that those who collect these data are often least able to exploit the resulting large-scale datasets. This issue is particularly acute in the context of commercial data exploitation, including by artificial intelligence and large-scale data mining enterprises. To address this, the ForestPlots community has developed data-sharing agreements that promote fairness and inclusivity, as detailed in de Lima et al. (2022).

| ForestScan Datasets / Data license type | URLs | DOIs | Citable as |
|---|---|---|---|
| ForestScan: Plot descriptions for FBRMS-01: Paracou, French Guiana, 1ha plots FG5c1, FG6c2 and FG8c4<br><br>License: CC BY-NC-SA 4.0 http://creativecommons.org/licenses/by-nc-sa/4.0/ | https://dataverse.cirad.fr/dataset.xhtml?persistentId=doi:10.18167/DVN1/94XHID | 10.18167/DVN1/94XHID | Derroire, G., Hérault, B., Rossi, V., Blanc, L., Gourlet-Fleury, S., Schmitt, L., 2025, "ForestScan", 10.18167/DVN1/94XHID, CIRAD Dataverse, V1 |
| ForestScan: Tree census data for FBRMS-01: Paracou, French Guiana, 1ha plots FG5c1, FG6c2 and FG8c4<br><br>License: CC BY-NC-SA 4.0 http://creativecommons.org/licenses/by-nc-sa/4.0/ | https://dataverse.cirad.fr/dataset.xhtml?persistentId=doi:10.18167/DVN1/94XHID | 10.18167/DVN1/94XHID | Derroire, G., Hérault, B., Rossi, V., Blanc, L., Gourlet-Fleury, S., Schmitt, L., 2025, "ForestScan", 10.18167/DVN1/94XHID, CIRAD Dataverse, V1 |





| ForestScan: Tree census data for FBRMS-02: Lope, Gabon, 1ha plots LPG-01, OKO-01, OKO-02 and OKO-03 and FBRMS-03: Kabili-Sepilok, Malaysian Borneo, plots SEP-11, SEP-12 and SEP-30<br><br>License: CC BY-NC-SA 4.0 http://creativecommons.org/licenses/by-nc-sa/4.0/ | https://doi.org/10.5521/forestplots.net/2025_2 | 10.5521/forestplots.net/2025_2 | Chavana-Bryant, C., Wilkes, P., Yang, W., Burt, A., Vines, P., Bennett, A.C., Pickavance, G., Cooper, D.L.M., Lewis, S.L., Phillips, O.L., Brede, B., Lau, A., Herold, M., McNicol, I.M., Mitchard, E.T.A., Barbier, N., Vincent, G., Coomes, D.A., Jackson, T., Makaga, L., Milamizokou Napo, H.O., Ngomanda, A., Ntie, S., Medjibe, V., Dimbonda, P., Soenens, L., Daelemans, V., Bartholomeus, H., Majalap, N., Nilus, R., Labrière, N., Burslem, D.F.R.P., Qie, L., Derroire, G., Proux, L., Abernethy, K., Jeffery, K., Clewley, D., Moffat, D., Scipal, K. and Disney, M. ForestScan: a unique multiscale dataset of tropical forest structure across 3 continents including terrestrial, UAV and airborne LiDAR and in-situ forest census data. ESSD. 2025 |

## 6. Author contributions

All authors provided input towards the writing of this manuscript.

C.Ch.-B. wrote the manuscript with significant input from M.D.

C.Ch.-B. collected, cleaned, processed and curated TLS data.

C.Ch.-B. developed the data repositories and ensured data integrity with support from M.D., the CEDA data management team and the ForestPlots and DataVerse database management team.

P.W. assisted in the collection of TLS data in FBRMS-02: Lopé, Gabon and its processing.

W.Y. assisted in the collection of TLS data in FBRMS-01 Paracou, French Guiana.

A.B., and T.J. collected TLS data in FBRMS-03: Kabili-Sepilok, Malaysian Borneo.

H.O.M.N. and L.M. provided field logistics and assisted in the collection of TLS data in FBRMS-02: Lopé, Gabon

L.S. and V. D. helped collect TLS in FBRMS-02: Lopé, Gabon.

K.A., S.N. & A.N. provided logistics and research permit support for FBRMS-02: Lopé, Gabon.

P.V. assisted in the processing of TLS data and developing the TLS2trees Processing Scripts.

A.C.B. collected census data in FBRMS-01 Paracou, French Guiana and in FBRMS-02: Lopé, Gabon with assistance from D.L.M.C.

V.M., P.D, H.O.M.N. and K.J collected the field census data for LPG-01

N.L., P.D., H.O.M.N. and K.J. collected the field census data for OKO-01, OKO-02 and OKO-03 in Lopé, Gabon.

T.J., D.C. and G.V. planned and funded the ALS data collection in FBRMS-01, Paracou French Guiana.

T.J. & D.C. planned and funded the ALS data collection in FBRMS-03, Kabili-Sepilok, Malaysian Borneo.





I.M.M. arranged, collected and processed the UAV-LS data collected over FBRMS-02: Lopé, Gabon.
B.B., A.L. and H.B. collected, cleaned, processed and curated TLS and UAV-LS data collected at Paracou, French Guiana.
N.B., G.V. collected, cleaned, processed and curated TLS and UAV-LS data collected at Paracou, French Guiana.
**7. Competing interests**
A.B. is an employee and/or shareowner of Sylvera Ltd. All other authors declare that they have no conflict of interest.
**8.  Acknowledgements**
We are indebted to the long-term work of many researchers in funding, establishing and maintaining the field plots that were
used in this study. It is not possible to carry out meaningful cal/val measurements of tropical forest biomass for earth
observation studies without the logistical support and expertise of the plot PIs and their teams. We thank Dr Noreen Majalap
for logistical and research permit support in FBRMS-03, Kabili-Sepilok, Malaysian Borneo. We also thank the Sabah
Biodiversity Council for their support with airborne laser scanning data collection in Kabili-Sepilok, access license number:
JKM/MBS.1000-2/2 JLD.9 (122). We thank Esther Conway and her team for their outstanding support in developing the
ForestScan CEDA dataset collection. We thank Dr Aurora Levesley and Gaëlle Jaouen for their generous support in developing
the ForestPlots and DataVerse tree census data packages. Specific data collection activities were funded by the European Space
Agency under ESA/ contract No. 4000126857/20/NL/AI. Work in French Guiana benefited from the Investissement d'Avenir
grants of the ANR, France (CEBA: ANR-10-LABX-0025). M.D., P.W., C.Ch.-B., W.Y. acknowledge capital funding for TLS
equipment from UCL Geography and the NERC National Centre for Earth Observation (NCEO).  T.J. and D.C. acknowledge
the funding for airborne laser scanning (ALS) data collection over FBRMS-01 Paracou, French Guiana in 2019 and FBRMS-
03: Kabili-Sepilok, Malaysian Borneo during February 2020 as part of a NERC project grant (NE/S010750/1). I.M.M. was
partly funded by a European Research Council Starting Grant (757526) awarded to E.T.A.M. Work in Lopé was supported by
core funding from Total Gabon and the EU-ACP ECOFAC VI grant to the Gabon National Parks Agency for logistics, staff
and site operations.



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
