# Peer review of "ForestScan: a unique multiscale dataset of tropical forest structure across 3 continents including terrestrial, UAV and airborne LiDAR and in-situ forest census data"

_Earth System Science Data, 2025_

## Referee Comment (RC2)

**Review**

ForestScan: a unique multiscale dataset of tropical forest structure across 3 continents including terrestrial, UAV and airbourne LiDAR and in-situ forest census data

Dear Authors,

Thank you for the opportunity to review your manuscript for possible publication in ESSD. I have provided a number of comments throughout, both as general observations and as specific notes referring to particular lines or figures. I hope you find these suggestions constructive.

**General points**

Title: consider this adjustment → ForestScan: a unique multiscale dataset of tropical forest structure integrating terrestrial, UAV, and airborne LiDAR with in-situ forest **inventory** data across three continents

Use of the word "census": Consider changing this to "inventory" in every instance (also in the title), I find inventory is more commonly used in forest research and remote sensing fields. In any case you can use census as a keyword.

There are lots of authors, and I suspect lots of contributions – sections need to be unified in style and language. Language is generally good, but often sections could be more concise. Please consider

Acronyms: please check every acronym throughout the whole manuscript, there is a lot of repetition – acronym after the first instance only (e.g., DBH, TLS, ALS, AGB, EO, VOLS, ....).

Please pay attention to the technological limitations (e.g. accuracy of LiDAR – refer to Morhart et al. 2024 <a href="https://doi.org/10.1007/s10342-023-01651-z">https://doi.org/10.1007/s10342-023-01651-z</a>), this is touched upon in line 311. Also consider elements of quality assessment of data processing (e.g., wood/leaf class extraction as shown in Fig. 7). I would like to see a section devoted to discussing possible methodological/processing errors within the dataset. I would also like the methodological/equipment limitations to be discussed, this is touched on in line 271. I would argue that the tilted scan is nice to have but an extra step (and all the extra workload) that might not be essential (dependent on the size of the trees scanned). Did you scan outside the plot to better capture the trees on the plot boundaries?

Plot/sub-plot numeration becomes confusing between FBRMS and their individual subplots – consider making this more transparent/unified.

Consider adding a summary table in section 2 outlining the study sites, country, coordinates, area, climate, ... this could considerably shorten the section which is a bit wordy. I would also like to know something about the stands that were scanned, tree density, mean DBH, top height, .....

Figures 1, 2 and 3 would be nicer if they had a unified style and similar content. For example, Figure 1 is really detailed and figure 2 is very vague. Some of the figures are missing basic mapping elements. It looks like they have been provided by separate people from the author consortium. Make sure the legends and captions are clear and correlate with the text body (e.g., lines 141 and 142 refer to treatment 1,2 and 3 – I guess this is T1, 2 and 3 in Fig 1? Also Line 230.

Mapping could also show the areas within each FBRMS that was scanned with each method. This is relevant, for example, to lines 589/590.

The methods sections (lines 281 – 332) could be condensed, call it data collection – the same or similar methods were applied for all TLS campaigns. One table for scanners used and their settings, one table for plot overviews. This part is very repetitive. E.g., Lines 318 and 328 (use of targets).

Line 766-775: Linking TLS trees to their census (inventory) counterparts. Plot marking is a valid point. Your examples are specific (e.g. Elephants). We too have tree tags that are destroyed by birds pecking at their shiny surface, we know the problem! We are also trialling QR codes on alu/foam board, plots can also be marked with a steel ground stake that can be found with a metal detector. The idea of anchor trees is good.

**Specific points**

Lines 44-46: define acronyms EO, AGB, cal/val, ...

Lines 78-80: Better description needed – particularly on the need for better intercalibration.

Line 86: the acronym for Spacebourne LiDAR is not LiDAR. Also be careful in this section ESA Biomass is Radar not LiDAR – it has also already been launched (line 90), update needed.

Line 93: make → undertaken.

Lines 107-108: selection based on discussions is vague (people will always discuss), stick to the criteria.

Lines 112-114: Bullet points → note the acronym after the site name in brackets. E.g., Paracou Research station, French Guiana (FBRMS-01). Also see the general comment above about plot codes/names.

Line 122: define Cirad-UMR EcoFoG.

Line 125: thousand separators, additionally please check all.

Lines 135: remove "in the early 1980's".

Line 138: done  $\rightarrow$  carried out.

Line 148: 9 → nine.

Lines 135-149: lots of (possibly) irrelevant detail that could be cut to make the section more focused (e.g., flux tower, fertilisation experiments). Please consider revising. Compare with the description of FBRMS 02 and 03 where there is much less.

Figure 2: a, b, c and d need defining in the caption (but consider first my general comment above).

Line 190: replace "laser-scanning" with "LiDAR" approaches

Lines 190-192: Sentence "species identity...." needs a rephrase. This and the next sentence can be made more precise. I think you mean the determination of tree species is critical since wood density x TLS derived volume = biomass.

Figure 4: remove – it is secondary information – just direct the reader to the right place.

Lines 201-202: Remove the sentence about ForestPlot.net – it is not needed.

Line 218: at 1.3m – this is DBH you have defined it before.

Lines 221-228: do you mean trees were recorded by their common names and then (as written in lines 226 – 228) trained botanists returned to identify species? If so try to combine these paragraphs.

Line 228: Explain APG IV

Line 233: Remove "referenced by a DOI"

Line 231 and 236: ask the journal how to display the hyperlinks, especially since the second is linked to a reference.

Line 240: Plot numbering/labelling confusion for the reader – link to map figure

Lines 238-241: much less detail shown here than FBRMS 01, why?

Lines 241 and 245: TLS was conducted not collected (TLS point cloud data was collected).

Line 242: "most" = vague.

Line 244: Same plot numbering problems as above.

Line 251: Gabonese and Malaysian FBRMS plots

Line 260: insert - chain sampling "protocols"

Line 261: QSMs needs describing

Figure 5a and b can be combined into one grid. Consider using axis labels to define scan position, the legend to the left must be much bigger to be legible.

Line 273: figure 5c?? missing.

Line 279: Working day is arbitrary – person hours give a better idea of workload.

Line 280: Maybe give the time needed for one scan.

Line 284: 16 quarter ha plots – that's seems like a complicated way of expressing area.

Line 319: give detail on the RTK equipment

Tables 1, 2 and 3: the Lat. Long. coordinates are very approximate.

Lines 338-339: delete bottleneck sentence. It is true but not needed here.

Lines 351: Maybe specs are needed for HPC cluster and CPUs? Time reference (ca. 4 days) is ok here as the reader knows no break is needed.

Line 362: Precision of person hours vs. days (see comments above).

Line 366: "potentially containing more than 5.42 billion points" impressive, but too accurate and irrelevant to the reader, please delete.

Line 366: Insert  $\rightarrow$  one small "exemplary" section of ....

Line 447, 552, 586, 599 and 616: delete "data access".

Lines 455-459: please condense.

Figure 10: a nice figure but too much content, if only representative maybe just choose one plot row for e.g., 10 trees.

Line 476: an overview table might help the reader interpret the similarities and differences between flights (across all plots and flights) – Table 5 should be used and referred to earlier, and could be expanded for the other plots.

Line 477: VLOS stipulation is a repeat.

Line 480-481: irrelevant, I would not suggest anything other than adherence to the flight rules given.

Line 487-488: delete cherry-picker (above canopy platform is sufficient as a description).

Figure 12: legend entries need defining e.g., AOI, DSM

Table 6 and 7: is something up with the UTC date and time in the tables??

Table 6: define AGL (assumed above ground level).

Line 555: please rephrase and be specific that you are referring to FBRMS02.

Line 556: DELAIR DT26X drone platform

Line 561: "different" can you be specific?

Figure 13: Please label the panels and use fitting captions, I am not sure what all photos are showing

me.

Line 609: ALS

Line 611: approximately = vague.

Line 620: WD, I don't think this is the first instance?

Lines 664 & 668: rephrase – much as/far as possible.

Line 673: EO

Line 677: "This may sound obvious" too chatty.

Line 687: replace "types" with "sources"

Line 715: clearing → cleaning?

Line 716: what is (h)?

---

## Author Comment (AC1)

**ForestScan Manuscript: essd-2025-67**

**Reviewer 1 — Hannah Weiser**

**GENERAL RESPONSE**

We would like to express our gratitude to both reviewers of our manuscript for their thoughtful and constructive comments. Their feedback has been invaluable in improving the clarity, quality, and overall presentation of our manuscript.

Our detailed responses to each of their points are provided below, as shown in the accompanying tables. We have carefully considered all suggestions and implemented revisions where appropriate. In cases where we have retained the original wording, we have explained our reasoning. We hope that our responses satisfactorily address our reviewers' concerns and demonstrate our commitment to producing a rigorous and transparent manuscript.

**GENERAL COMMENTS**

| | |
|---|---|
| **1)** The dataset is very extensive with three study sites and three LiDAR platforms plus census data, making it difficult to review and also difficult to follow the manuscript and find relevant information. While I do support describing the dataset as a whole and including the various sub-datasets, the authors are advised to improve the organization of the dataset (cf. 5. Data Access) and the description and presentation, e.g., by improving and adding figures and tables. This will make it easier for readers/users from different fields to find the relevant information and to access the data they are interested in. | Most suggested new tables have been added and previous Tables 8a & b in section 5. Data Access have been separated into three tables arranged by FBRMS site with column 1 displaying each site's datasets, Acquisition date and Data license type; column 2 displaying the Data type (tree census, TLS, UAV-LS or ALS); and column 3 displaying the Citable as information which now includes both the URL and DOI for each dataset. There are now a total of 11 tables. Site figures have also been completely redesigned to standardise them. These new site figures include several panels providing continental, country and/or regional (Research station/National park/Forest reserve) perspectives together with their detailed site maps which now display TLS plots + census plots + UAV-LS coverage + ALS coverage and the dates of their collection. More detailed descriptions of these additions are provided in responses for later comments. |
| **2)** The manuscript should state more clearly which analyses were undertaken and which can be addressed (by others) in the future — especially in | Section 3 has been renamed "3. Recommendations for aligning and matching datasets". For clarification, we have added the two below paragraphs at the beginning this section:

We provide data that are internally consistent in terms of pre-processing, geo-referencing, and exported in formats compatible with |

| | |
|---|---|
| Section 3 (Aligning and matching datasets). For this, it may help to modify the section title. | open-source tools. Any further processing will depend largely on the intended application, such as individual tree analysis or plot-level studies.

For TLS data, all point clouds within a single plot are co-registered into one unified point cloud. These are subsequently processed into individual tree point clouds, to which quantitative structural models (QSMs) are fitted to estimate volume.

UAV-LS and ALS datasets are geo-referenced in each case, however, they are not explicitly aligned or matched to one another. Alignment can be performed, but it requires manual identification of control points in each dataset and, as noted above, will depend on the intended use of the resulting data. |
| **3)** The authors should provide users with more information on the quality of the dataset (e.g., accuracies, error estimates or possible error sources, etc.). | We agree and have made some effort to do this for each dataset. See responses below to specific comments. |

**SPECIFIC COMMENTS**

| | |
|---|---|
| **1)** It would be interesting to get more information about possible error sources as well as accuracy estimates, e.g., of alignment of scan positions and flight strips. | We have added/amended text in subsection 2.2.2 TLS data processing 1. Individual scan registration into plot-level point cloud for TLS data and section 3. Recommendations for aligning and matching datasets for UAV-LS and ALS:

**1. Individual scan registration into plot-level point cloud**
This process was carried out using retro-reflective targets positioned between scan locations to facilitate coarse registration for data collected with the RIEGL VZ-400 or in a near-automated manner using the RIEGL VZ-400i's GNSS RTK positioning capabilities in conjunction with the enhanced RIEGL RiSCAN Pro software (versions 2.14–2.17). The integrated Auto Registration 2 (AR2) function employs GNSS RTK data to update the scanner's position and orientation, including in tilt mode, thereby enabling real-time automated coarse registration during scanning. Major registration errors are easily detected, typically occurring during pre-processing in RiSCAN Pro when individual scans fail to register (i.e., no coherent solution is found) or are incorrectly positioned, which is visually apparent. In cases where coarse registration/auto-registration fails, unregistered scans can be identified, adjusted, and refined using Multi Station Adjustment 2 (MSA2). Following this workflow, the co-registration of all TLS point clouds achieves sub-centimetre accuracy, as confirmed through post-registration inspection. Wind and occlusion are key sources of uncertainty for the scan registration process, |

highlighting the necessity of scanning under low or zero wind conditions and capturing both tilt and upright scans at each location.

**3. Recommendations for aligning and matching datasets**
UAV-LS and ALS datasets are geo-referenced in each case, As positional accuracy depends on the IMU and GNSS measurements, which can introduce errors manifesting as height biases between individual flightlines. Although we did not observe such discrepancies in our data, a rigorous comparison with ground control points would be required to confirm this definitively -a step we have not undertaken. These datasets have are not been explicitly aligned or matched to one another. Alignment can be performed, but it requires manual identification of control points in each dataset and, as noted above, will depend on the intended use of the resulting data.

| | |
|---|---|
| **2)** Do you have a way of judging the quality of the segmentation of the TLS data and the quality of the reconstructed QSMs? Also here, can you state limitations and possible sources of error? | We have included details about this in subsection 5. TreeQSM: The overall QSM fit is controlled by three parameters, which are iterated into 125 different parameter sets, each generating five models. This yields a total of 625 candidate models per segmented tree. The optimal model is then selected by minimising the point-to-cylinder surface distance (Burt et al., 2019; Martin-Ducup et al., 2021). Estimates of morphological and topological traits such as volume, length, and surface area metrics, along with their mean and standard deviation, are derived from the five models that share the same parameters as the optimal model. This approach provides an estimate of the uncertainty associated with the resulting volume (Wilkes et al., 2023). In a HPC system, QSMs for a 1 ha plot can take up to 2 days to complete.

Uncertainty estimates are reported for each ForestScan FBRMS plot and included alongside the final modelling outputs for every tree in a 'tree-attributes.csv' file, generated at the end of the modelling process. Sources of error in QSM fitting can arise from data acquisition (e.g., wind, leaf occlusion, understory vegetation) and from assumptions inherent in segmentation and fitting processes. Wilkes et al. (2017) discuss issues related to data acquisition and methodological choices, while Morhart et al. (2024) quantify their effects on branch size and volume under controlled conditions. Although these impacts are difficult to assess without reference (harvest) data, Demol et al. (2022)  show that, where TLS and harvest data have been compared, agreement is generally within a few percent of AGB per tree. The report CVS file also includes tree- and plot-level carbon and AGB estimates, the latter based on a mean pantropical wood density value of 0.5 g cm$^{-3}$ derived from the DRYAD global database of tropical forest wood density (2009). Plot-level AGB was also estimated using DRYAD-derived regional mean wood densities and is presented in Table 5. |
| **3)** Study site figures: It would be easier for the reader if all the study site figures would be coherent | We thank the reviewers for these valuable suggestions. We have completely redesigned all three site figures to ensure consistency and improve clarity: |

| | |
|---|---|
| in style (font, colours, etc.) and content (overview map, map elements, legend, background map type to show elevation, etc.). Please find some further detailed comments below. | **Comment 3 (Coherent style and content):**
All three site maps now follow a standardised three-panel format: (a) continental overview showing the country location, (b) country/regional view showing the research station, and (c) detailed site map with high-resolution satellite imagery. All figures use consistent fonts, legend styles, colour schemes (orange for FBRMS-01 TLS plots, treatment-specific colours for census plots, yellow tones for ALS coverage, purple for UAV-LS coverage), scale bars, north arrows, and boundary styling. |
| **4)** Consider adding a world map that shows a marker for all the three sites on the three continents. | **Comment 4 (World map showing all three sites):**
Each site map now includes a continental-scale overview panel that clearly shows the country location. The three figures together provide complete geographic context across South America (French Guiana), Africa (Gabon), and Southeast Asia (Malaysia), making it straightforward for readers to understand the global distribution of our study sites across three continents. |
| **5)** Fig. 1 would benefit from an overview map and a scale bar. It might also be better to mark the three ForestScan plots all by the same colour very prominently and label them by their plot ID directly in the figure. | **Comment 5 (Fig. 1 - Paracou improvements):**
We have added the three-panel layout with South America overview and French Guiana regional maps, plus a scale bar in the detailed panel. The three FBRMS-01 TLS plots (plot 5-subplot 1, plot 6-subplot 2, plot 8-subplot 4) are now prominently displayed in bright orange with white borders and are directly labelled with their subplot IDs. A comprehensive legend shows all plot types, experimental treatments, and data coverage areas (ALS and UAV-LS). |
| **6)** Fig. 2 needs bigger labels, and a label for the colour scale would be useful. Also, it would be useful if the census and TLS plots as well as UAV areas were shown here. | **Comment 6 (Fig. 2 - Lopé improvements):**
We have significantly increased all label sizes for better readability and the elevation image is not used. All census plots are now clearly shown as semi-transparent coloured polygons according to their treatment type, TLS plots are prominently marked in orange with white borders, and the UAV-LS coverage area is shown as a semi-transparent overlay. |
| **7)** Fig. 3: It is confusing that the legend mentions "4 ha" plots but the caption says "1 ha" plots, please clarify this. Please also add to the caption that there are always 4 plots in the corners of a "plot area". | **Comment 7 (Fig. 3 - Malaysia plot clarification):**
We have clarified in both the figure caption and legend that the field sites consist of 4 ha "plot groups," each containing four 1 ha census plots arranged in a 2×2 grid. The caption now explicitly states: "Each 4 ha plot group contains four 1 ha subplots (shown as individual coloured squares)." The individual 1 ha subplots are clearly visible in the detailed satellite imagery view, making the spatial arrangement unambiguous.
These comprehensive revisions ensure all site maps are publication-ready with consistent, professional presentation that enhances reader comprehension of the study locations and data collection infrastructure. |
| **Data** | |
| **8)** Tree census: I think a tabular overview or a timeline would make sense here so the user can quickly find the census times for each FBRMS. | Please see our response to General Comment 1) |

| | |
|---|---|
| **9)** Generally: If feasible, a timeline for all measurements (per study site) would be useful. | Please see our response to General Comment 1) |

**Terrestrial laser scanning (TLS)**

| | |
|---|---|
| **10)** Can you also add tables with scanner specifications (cf. Table 5 for UAV-LS)? | Done, Table 1: TLS sensor systems used at ForestScan FBRMS has been added to subsection 2.2.2 Terrestrial LiDAR Scanning (TLS) |
| **12)** Pre-processing: Please mention and describe the downsampling of point clouds (cf. line 427), i.e., which algorithm and parameters were used. Was any filtering performed, e.g. in the RiSCAN Pro Import Wizard by reflectance or deviation, or outlier filtering? If so, please add this information. | The data processing section has been revised for clarity and conciseness. The highlighted detail was also added to item 1 in subsection TLS datasets: 1. Raw terrestrial LiDAR data from each scan ==(no filtering was applied in RiSCAN PRO)==, stored in the RXP data stream format developed by RIEGL. |
| **12)** Can you explain why the files are provided as .ply and not as .las/.laz? | Files are provided in **PLY format** as it supports full 3D object representation, including polygons and geometric primitives, in addition to point data. This is essential for storing quantitative structure models (QSMs), which go beyond point clouds to describe tree geometry. In contrast, LAS/LAZ formats are designed specifically for point cloud data and do not accommodate object-level information. The PLY format is open, widely supported in Python and R, and the CloudCompare software and can be converted to/from LAS/LAZ when only point data are required. An explanation has been added to the TLS datasets subsection as item 7 shown below: 7. We provide pre-processed and segmented terrestrial LiDAR data in PLY format as it supports full 3D object representation, including polygons and geometric primitives, in addition to point data. This is essential for storing quantitative structure models (QSMs), which go beyond point clouds to describe tree geometry. The PLY format is open, widely supported in Python and R, and can be converted to LAS/LAZ when only point data are required. |
| **13)** Are additional point cloud attributes provided (e.g., Intensity/Reflectance, GPSTime, etc.)? When | Description of these fields have been added in subsection TLS datasets items 3a, 3b and 5 as shown below: 3. Pre-processed terrestrial LiDAR data: |

| | |
|---|---|
| checking the dataset, I noticed that the .ply files have further fields, e.g., scan position index, reflectance, deviation and for the segmented point clouds the segmentation results and classification probabilities. This is very valuable information that should be included in the dataset descriptor. | a. full-resolution 10m tiled plot point clouds including attributes such as XYZ, scan position index, reflectance, deviation, etc. stored in polygon PLY format.

a. downsampled 10m tiled plot point clouds including attributes such as XYZ, scan position index, reflectance, deviation, etc. stored in polygon PLY format.

4. Wood-leaf separated tree-level point clouds including segmentation results and classification probabilities for each point are stored in polygon PLY format. |
| **14)** Line 429: Is that classified point cloud also downsampled? | No, downsampling is only performed once on during pre-processing as now clarified in the revised TLS data processing section. |
| **15)** Line 443f.: "GNSS coordinates [...] for all scans …" ? do you mean the coordinates of the scan positions, i.e., the scanner location? Please clarify. | Yes, for clarification, item 10 in subsection TLS datasets has been revised as: 10. GNSS coordinates (geographical coordinate system: WGS84 Cartesian) for all scan positions stored in KMZ zip-compressed format. These files are available for the seven French Guiana and Gabon FBRMS plots.

This is also clarified in the revised TLS data processing section. |
| **16)** Figure 11: So TLS almost consistently overestimates the DBH compared to the census? Can you state this explicitly in the text/caption? | The caption for Figure 11 has been revised: **Figure 11:** Quantile-Quantile (QQ) plots comparing the distribution of diameter at breast height (DBH) measurements collected by tree census and TLS methods at each of the 10 ForestScan FBRMS 1 ha plots. TreeQSM measures DBH at the standard height of 1.3 m for each TLS-extracted tree, whereas census DBH measurements are routinely adapted to account for tree buttresses found among larger trees. Generally, census and TLS DBH measurements are in good agreement but consistently overestimated by TLS. Deviations for larger DBH values can be improved by adapting the DBH extraction of large buttressed trees once these trees are matched to their census counterparts. The 1:1 reference line (dotted black line) represents perfect agreement between census and TLS-extracted DBH measurements. |
| **17)** In which coordinate system is the point cloud data provided? | Assuming this comment refers to the GNSS scan position coordinates, pls see reply to comment 15). |
| **UAV-borne laser scanning (UAV-LS)** | |
| **18)** Lines 499-508: Is this relevant for the data publication? Please either make the connection clearer (e.g., because | This paragraph has been removed. |

| | |
|---|---|
| flights have not been possible as planned due to restrictions) or leave this section out. | |
| **19)** L. 522 / Table 6: maybe add an explanation like "a double grid plus an additional double grid at a 45° angle" to the term "criss-cross" and refer to Figure 12. | The captions for both Figure 10 (previously Figure 12) and Table 7 (previously Table 6) have been revised for clarity:

**Figure 12:** UAV-based LiDAR (UAV-LS) flight trajectories over the FBRMS-01 site at Paracou, showing coverage of experimental 4 ha plot 6 (see Fig 1; red dashed line). The criss-cross flight pattern results from multiple flight lines oriented in different directions (e.g., N–S, E–W, NE–SW) to improve point density and reduce occlusion in dense tropical forest canopies. The background depicts a digital surface model (DSM) with elevation values (m). The inset map shows the regional location of Paracou within French Guiana (© OpenStreetMap contributors, available at https://www.openstreetmap.org).

**Table 7:** Overview of 2019 VUX-1 UAV-LS flights at FBRMS-01 (Paracou), including plot ID, acquisition date/time, flight height above ground level (AGL), velocity, and pulse repetition rate. Flight patterns refer to the orientation of flight lines: N–S (north–south), E–W (east–west), NE–SW (northeast–southwest), and "criss-cross" indicates multiple orientations flown over the same area as seen in Fig. 12. All flights listed can be considered part of one acquisition, they are provided as individual point clouds in this dataset. Users may merge them according to their needs. |
| **20)** The plot IDs from the tables vs. in the text are confusing. Is FG6c2 the same as "P6". What does 200 and 100 mean in Table 6? | FG6c2 and P6 are partly the same. In Paracou, there are 15 experimental 4 ha plots with 4 subplots each containing four 1 ha subplots numbered 1 – 4, the three 1 ha ForestScan FBRSM are subplots FG5c1, FG6c2 and FG8c4 correspond to subplots 1, 2 and 4 in plots 5, 6 and 8, respectively. This is now fully explained in the revised manuscript in the caption for Figure 1 displaying the Paracou site map (see below), and in subsections TLS: FBRMS-01: Paracou, French Guiana and UAV-LS: FBRMS-01: Paracou, French Guiana

**Figure 1:** Multi-scale map depicting the location and spatial distribution of research plots at Paracou Research Station, French Guiana. (a) Location of French Guiana (green) within South America. (b) Location of Paracou Research Station (green) within French Guiana. (c) Detailed site map showing the spatial distribution of research plots with treatment-specific colours, UAV-LS coverage (orange), and ALS coverage (yellow). The map displays 15 experimental 4 ha plots, each containing four 1 ha subplots numbered 1 - 4 (60 subplots in total; plots 1 - 12: silvicultural treatments; plots 13 - 15: Biodiversity monitoring), one large 40 ha Biodiversity plot (plot 16; red), and 10 GuyaFlux plots (solid green). Treatment categories include: Biodiversity monitoring plots (plots 13, 14, 15, 16; red), T0 Control (plots 1, 6, 11; green), T1 Selective logging (plots 2, 7, 9; dark blue), T2 Selective logging + thinning by timber stand improvement (TSI; plots 3, 5, 10; cyan), and T3 |

| | |
|---|---|
| | Selective logging + TSI + fuelwood harvesting/FW (plots 4, 8, 12; pink). The three FBRMS-01 subplots -FG5c1 (subplot 1 of plot 5), FG6c2 (subplot 2 of plot 6), and FG8c4 (subplot 4 of plot 8)- are shown in solid orange and were surveyed using terrestrial laser scanning (TLS) with corresponding tree census data. The GuyaFlux tower location is indicated by a black triangle with radiating transmission waves, and the Base Camp location is marked with a white square. Scale bar: 800 m. Map data: Natural Earth 10 m cultural vectors. Satellite imagery basemap: Imagery ©2024 Google. Map projection: WGS84 (EPSG:4326). |
| **21)** Table 6 and 7: Please make clearer which flights belong together and can be considered one acquisition (meaning that resulting point clouds are merged to one point cloud before further processing) | Please note these two tables have been renumbered to Table 7 & Table 8. Their captions now clarify which flights can be considered part of a single acquisition while being provided as individual point clouds which users can merge according to their needs (see table captions below). We have also added the extra characteristic "Flights merged into single acquisitions" to Table 6: UAV-LS sensor systems used at ForestScan FBRMS-01 and FBRMS-02.

**Table 7:** Overview of 2019 VUX-1 UAV-LS flights at FBRMS-01 (Paracou), including plot ID, acquisition date/time, flight height above ground level (AGL), velocity, and pulse repetition rate. Flight patterns refer to the orientation of flight lines: N–S (north–south), E–W (east–west), NE–SW (northeast–southwest), and "criss-cross" indicates multiple orientations flown over the same area as seen in Fig. 12. All flights listed can be considered part of one acquisition, they are provided as individual point clouds in this dataset. Users may merge them according to their needs.

**Table 8:** Overview of UAV-LS flights using a YellowScan Vx20 system (RIEGL Mini-VUX scanner and Applanix 20 IMU) mounted on a DJI M600 during the 2019 mission at the FBRMS-01 site. Automated flight plans were performed using flight plans with the UgCS route planning software in grid mode. The table lists plot ID, acquisition date/time, flight parameters (direction, interline spacing, altitude and speed). Altitude values are reported as specified during flight planning with some missions using Above Ground Level (AGL), while others used Above Mean Sea Level (AMSL) due to differences in mission planning and operational requirements. These original specifications are retained to accurately reflect acquisition parameters. Pulse repetition for the RIEGL Mini-VUX scanner is fixed at 100kHz. Flights cover multiple experimental plots: 4 & 5 (single flight), 6 (8 flights), 7, 8, 10, 15, and the Tower plot (two flights) within the Paracou Research Site. All listed flights are provided individually; users may merge flights covering the same plot if needed for analysis.

For clarification, the characteristic "Flights merged into single acquisition" was also added to Table 6 covering the three UAV-LS sensor systems used at ForestScan FBRMS plots. |

| | |
|---|---|
| **22)** Please harmonize Tables 6 and 7, i.e., use the same "Date & Time" format (check ESSD author guidelines for preferred format), the same term for the velocity, and ideally the same specifications for the pattern (e.g., directions, interline distance, ..) | Tables 7 & 8 (they have been renumbered) have been standardised to:
Date: UTC ISO 8601
Direction: in degrees
Interline
Alt
Speed
Pulse repetition

Their captions have been revised for clarity as detailed in comment 21. |
| **23)** L 548ff.: Please improve the documentation of the LAStools processing steps and also explain what the mentioned commands do. | The text included in subsection UAV-LS data processing has been revised to include the requested details:
All collected raw data underwent processing with standard tools. For VUX-1UAV data, this included processing recorded global navigation satellite system (GNSS) and base station data to flight trajectories with POSPac Mobile Mapping Suite 8.3 (Applanix, Richmond Hill, Ontario, Canada), laser waveform processing to discrete returns and geolocation in world coordinates with RIEGL RiProcess 1.8.6. For miniVUX-1UAV, waveform processing is performed online in the sensor. Point cloud processing and geolocation was performed with the CloudStation software (Yellowscan, Montpellier, France), using the Strip Adjustment option. For all UAV-LS data, only points with a reflectance larger than -20 dB were kept for further processing. Points with reflectance smaller than -20 dB consist mainly of spurious points caused by water droplets under high humidity conditions (Schneider et al., 2019).

LiDAR point clouds were processed using the *LAStools* suite (rapidlasso GmbH). First, a 1-m resolution digital surface model (DSM) was generated with **lasgrid** using the highest return within each cell. Ground points were then classified with **lasground** (wilderness settings, 15-m step), and a 1-m digital terrain model (DTM) was derived from ground-classified points using **las2dem**. Heights were normalized by subtracting ground elevation with **lasheight**, producing a set of height-normalized point clouds. A 1-m canopy height model (CHM) was computed with **lascanopy**, retaining the maximum height in each grid cell after removing noise and low-confidence classes. Finally, a point density map (1-m resolution) was created using **lasgrid** with the *counter* option. This workflow produced consistent DSM, DTM, CHM, and density layers suitable for subsequent ecological analyses. These UAV-LS datasets are freely available via the Centre for Environmental Data Analysis (CEDA) with DOIs provided in section 5. Data access. |
| **24)** L 572ff: This paragraph is quite general and it is not clear what "must" or "can" be done | This text has been revised to:
Flight trajectories were reconstructed using GNSS/IMU measurements and adjusted with differentially corrected base station data in Applanix POSPac software. The corrected flight paths and |

| | |
|---|---|
| and what was actually done for the specific published dataset. Did you use RiProcess? | laser data were then integrated using the RIEGL software package, RiPROCESS, to generate the initial three-dimensional point cloud. Residual trajectory errors—such as discrepancies in GPS tracking and elevation—were corrected by using small buildings as reference points to refine the relative position and orientation of individual flight lines and scans. Further adjustments were made using ground control points: square targets (1–2 m²) composed of alternating black and white material arranged in a checkerboard pattern. This process resulted in a LiDAR-derived point cloud with a geometric accuracy of 1.8 cm. All elevation data were calculated as ellipsoidal heights (m) within the UTM 32S coordinate system. Each flight was processed separately, and all datasets were merged prior to export. Subsequent point cloud processing was carried out using elements of the lidR package (v3.1.0; Roussel et al., 2020). This UAV-LS dataset is freely available via the Centre for Environmental Data Analysis (CEDA) with DOIs provided in section 5. Data acquisition characteristics can be found in Table 6. |
| **25)** L 582ff: Please make it clearer which steps CAN be performed and which WERE actually performed (e.g., ground filtering, individual tree detection). This is currently not clear from the text. | |
| **26)** In which coordinate reference system is the point cloud data provided? | |

**Airborne laser scanning (ALS)**

| | |
|---|---|
| **27)** L589ff.: This information could be summarised in a table, then it would be easier to find. | A new Table 9 has been added and referenced. Its Table caption is: **Table 9:** Comparison of ALS acquisition characteristics for two ForestScan sites: FBRMS-01:Paracou, French Guiana and FBRMS-03: Kabili-Sepilok, Malaysian Borneo. These key flight and sensor characteristics can support alignment and comparability across sites. |
| **28)** L617 – Section 3. Aligning and matching datasets: It is unclear if any of that was done. It should be stated more clearly what was done by you and what you recommend can be done with the dataset (by others) – Maybe also change the section header to something like "Recommendation ..." | Aligning and matching of these datasets has not been done. We have changed the name of this section to "3. Recommendations for aligning and matching datasets" to help clarify this. |
| **29)** L664-670: This sections seems unrelated to the alignment topic, please clarify (or leave out). | Section 3. Has now been renamed to: Recommendations for aligning and matching datasets which should help clarify that users are free to align these datasets according to their needs. Both paragraphs in subsection 3.3 Aligning TLS and UAV-LS to ALS data referred to in this comment have been revised for clarity:
Aligning ALS data with TLS and UAV-LS datasets presents significant challenges. Despite the use of high-quality GNSS positioning, meter-scale geolocation discrepancies between sensors can occur. Co-locating LiDAR datasets acquired at different scales - TLS, UAV-LS, and ALS- remains complex, with no standard or "turn-key" solution currently available. Manual intervention is often |

| | |
|---|---|
| | required, and the approach varies by site and sensor combination. While plot-level AGB estimation is relatively tolerant to these discrepancies, finer-scale applications (e.g., matching to tree-level census data) demand more precise alignment. This can be partially addressed through manual co-registration using common tie points across datasets.

Achieving meaningful alignment also depends on the internal characteristics of ALS point clouds. Acquisition parameters such as point density, scan angle distribution, and footprint size influence comparability and should be controlled as far as possible. Post-processing can regularise point density and scan angles within or across campaigns, improving consistency. Homogeneous scanning geometry enables more stable structural metrics and enhances AGB prediction performance. Similarly, parameters such as transmitted pulse power (which co-varies with pulse repetition rate) and flight altitude (affecting footprint size and canopy penetration) should be standardised across acquisitions to minimise bias (Vincent et al., 2023). These steps are critical for reducing alignment errors and ensuring robust comparisons between TLS, UAV-LS, and ALS datasets. |
| **30)** For further use of the data, can you refer to free software and tools to visualize and analyse the point cloud data (e.g., LAStools, CloudCompare, etc.)? This might be especially relevant for the QSM data, which is provided in a rather non-standard format (.mat files). Here, it would be interesting for the user, how they can open and further analyse the files (e.g., to export the tree models as .OBJ, etc.). It would also be helpful to guide the users how to transform the TLS data into georeferenced coordinate using the .DAT files and/or GNSS coordinates of the scan positions. | For clarification, we have added the below paragraph at the end of the TLS datasets subsection:
QSMs can be converted to PLY format using ope-source tools such as *mat2ply* (Wilkes and Yang, 2025b) and then read by various tools such as the widely-used free GUI tool CloudCompare (CloudCompare Development Team, 2025; https://www.cloudcompare.org), via Python using PDAL (PDAL Contributors, 2025; https://zenodo.org/records/4031609) or O3d (Open3D Development Team, 2025; https://www.open3d.org/docs/0.9.0/tutorial/Basic/file_io.html#mesh), or via the R Geomorph package (Adams et al., 2025; https://rdrr.io/cran/geomorph/man/read.ply.html). In the Geomorph R package, the function Read mesh data (vertices and faces) from PLY files can be used to read three-dimensional surface data in the form of a single PLY file (Polygon File Format; ASCII format, from 3D scanners). Vertices of the surface may then be used to digitise three-dimensional points. The surface may also be used as a mesh for visualising 3D deformations using the warpRefMesh function. The function opens the PLY file and plots the mesh, with faces rendered if file contains faces, and coloured if the file contains vertex colour. Vertex normals allow better visualisation and more accurate digitising with digit.fixed. The KMZ files containing the GNSS scan position coordinates can be uploaded to Google Earth or read into a GIS tool such as QGIS (QGIS Development Team, 2025; https://qgis.org).

We're not quite sure what the last part of this comment means in that the GNSS scan position coordinates are georeferenced and provided in KMZ files which can be uploaded to Google Earth or read into a GIS tool such as QGIS (https://qgis.org). |

| | |
|---|---|
| **31)** Please add a "Conclusion" section as per the author guidelines (https://www.earth-system-science-data.net/submission.html#manuscriptcomposition) and check if a "code availability" section is needed. | Thank you for your suggestions. Regarding the **Conclusion** section, ESSD data papers do not necessarily require a separate conclusion, as their primary purpose is to describe the dataset and its accessibility rather than present research findings. Including such a section would add length without providing additional value for readers. Furthermore, another reviewer specifically recommended shortening the manuscript due to its length, thus, adding a new conclusion would conflict with that advice. We believe the current structure, which aligns with ESSD's emphasis on data description and availability, is appropriate for this data paper.

As for the **Code Availability** section, the three repositories (rxp-pipeline, TLS2trees and PDAL) relevant to this work have now been cited in the first paragraph of the TLS data processing subsection in the manuscript and included in the reference list. These scripts are utilities not required to access or use the dataset. Therefore, we do not deem a separate Code Availability section is not necessary. This approach is consistent with ESSD guidelines, which recommend including this section only when code is essential for data use, and also helps us respect the other reviewer's recommendation to keep the manuscript concise. |
| **32)** Can you make it clearer that all entries in Table 8 are included in the dataset collection from the first row (if that is the case)? | The Data Type for the ForestScan Collection was required and generated by the CEDA archive. It has been included in the Table 10 in section 5. Data Access. For clarity, its Data type is: Collection (multi-type composite of all ForestScan CEDA datasets) |

**TECHNICAL CORRECTIONS**

| **General** | |
|---|---|
| **1)** Specification of coordinates: Please specify the coordinates always in the same format, i.e., either using minutes, seconds (like in lines 120-121) or using decimals (like in line 156). | Done, both coordinates are in Degrees, Minutes, Seconds (DMS) format without seconds. |
| **2)** Make sure to add spaces before units consistently (where applicable). | Done |
| **3)** When referring to Figures and Tables in the text, consider omitting the word "below". | Done |

| | |
|---|---|
| **4)** L. 231 and 236: Is there the same DOI on purpose or is this an error? | The first URL has been revised to ([https://dataverse.cirad.fr](https://dataverse.cirad.fr)) |
| **5)** L. 240: Do you mean "10 x 1 ha" here? | No, 4 x 1 ha is correct as it refers to the 4 ForestScan FBRMS Lope plots |
| **6)** L. 244-245: The plot sizes are confusing here. Were there 9 x 4 ha plots and additionally 3 x 1 ha plots and 1 x 2 ha plot? | This paragraph was revised to clarify the size of the plots and the mention of a 2ha plot which was not included in the ForestScan project was removed:
In the Kabili-Sepilok FBRMS, tree census data was collected during 2020 - 2022 for a total of 9 x 4 ha plots (IDs RP291-1, RP292-3, etc. see Fig. 3) each containing four 1 ha subplots numbered 1 – 4 and covering most of the long-term plots at this site. The three FBRMS subplots SEP-11 (subplot 2 of plot RP292-3, sandstone soil), SEP-12 (subplot 2 of plot RP292-1, alluvial soil) and SEP-30 (subplot 3 of plot RP508-4, kerangas soil) were scanned using TLS during March 2017 and tree census for all subplots was collected in Jan, Mar of 2020 and Jun 2021. The 2020-2022 census was overdue as these plots had not been censused since 2013. |
| **Terrestrial laser scanning (TLS)** | |
| **7)** Terminology: Decide for either "Terrestrial LiDAR Scanning" or "Terrestrial laser scanning" | Terrestrial laser scanning is now used throughout the manuscript. |
| **8)** Fig. 5:
8.1) Please call "Figure 5" and not "Figure 5a & 5b".
8.2) Labels for "upright" and "tilt" scan seem do not match the images.
8.3) Increase label font size.
8.4) Please, fix the labels in the caption, where 5b and 5c (which does not exist) are referred to.
8.5) The subplots a and b seem redundant (since only the labels differ). Can they be combined into one figure? | 8.1) Figure has now been renamed Figure 5 and a and b are now referred as panels in the caption.

8.2) The labels for the upright and tilt scans have been removed to avoid confusion.

8.3) The figure is now larger with label fonts also larger. Labels cannot be larger as labels will then overlap.

8.4) See response for 8.1)

8.5) Panel (a) and (b) cannot be combined as they are not redundant, (a) keeps track of the line positions and (b) keeps track of the scan position number, both of which are part of the chain sampling protocol. |
| **9)** Chapter "TLS data processing" (Line 333ff): Using Arabic numbers for the subsections might be confusing to the reader | This entire section has been revised for clarity and conciseness and this numbering has disappeared. |

| | |
|---|---|
| (cf. Section numbering), so please consider using letters (a, b, c) or Roman numbering (I, II, III) instead. | |
| **10)** The TLS data acquisition section has a lot of repetition: Maybe the three sites can be summarized into one section (and the tables into one table) | This section has now been shortened and a new Table 1 (as requested in comment 10 in the TLS section of the SPECIFIC COMMENTS section. As the Paracou site has different columns according to specific forest treatment categories, we can't join the 3 FBRMS site tables and we prefer to maintain these tables separate for clarity. |
| **11)** Can you add acquisition dates to the tables? | These dates have not been added to these tables to avoid redundancy as acquisition dates have already been added to the each of the site figures (Figures 1 -3 in the Methodology section) and for each dataset in all three tables in section 5. Data Access. |
| **12)** Table 3: The caption seems incomplete ("Note: subplot 2 was"). | This text has been deleted. |
| **13)** Fig. 6: Add scale bar or colour bar legend. | In response to both our reviewers' comments, we have removed this figure from the manuscript, as it was deemed unnecessary and contributed little to the overall paper. Figure numbers have been amended accordingly. |
| **14)** Fig. 7: Add scale bar; add labels a) and b) and some descriptive captions. The colours are rather difficult to distinguish, can you adapt the style? | Thank you for the suggestion, the figure caption has been revised so it is clear that no continuous colour scale is required as colours represent discrete classes rather than a quantitative variable.

**Figure 7:** Tree-level point cloud of the largest *Baillonella toxisperma* (Maobi) tree (~40 m tall with an almost circular canopy ~50 m wide) in plot LPG-01, FBRMS-02: Lopé, Gabon. Points are classified and displayed by category only: **wood points in brown and leaf points in green**. |
| **15)** Figs. 8 and 9: It would be great if the point clouds and the derived QSMs used the same colour scheme for the tree instances, so clouds and QSMs can be better connected by the reader. | Please note these figures have been renumbered to 7 and 9. Thank you for your suggestion regarding colour consistency. We fully appreciate the rationale; however, the figures are designed to illustrate distinct aspects of the dataset. Figure 7 presents individual tree-level point clouds segmented by tree instance, whereas Figure 8 displays the corresponding QSMs derived from the segmented point clouds. The differing colour schemes are intentional, serving to emphasise that these are separate data representations. Harmonising the colour schemes would require additional processing and would not materially enhance the scientific interpretation, as the figures are primarily illustrative of structure and complexity. We believe the current approach is clear and aligned with the objectives of the paper. |

| | |
|---|---|
| **16)** Table 4: Can you add the numbers from the caption to the table columns, so that it is clearer for the reader what you are referring to? | To improve clarity, the caption for this table has been revised and the numbers from the caption added to the table:
**Table 5:** Summary statistics for 10 FBRMS ForestScan TLS plot datasets. AGB estimates use wood density values from the DRYAD global database (Zanne et al., 2009): (1) TLS2Trees pantropical mean, (2) Tropical Africa mean (TAF, Gabon), (3) South-East Asia mean (TS-EA, Malaysia), (4) Tropical South America mean (TSA, French Guiana), (5) Guyana community mean (GF, French Guiana), and (6) allometric AGB estimates based on Chave et al. (2014). |
| **17)** Figures 10, 11: Please increase the font size for the y-axis labels. | Done |

**UAV-borne laser scanning (UAV-LS)**

| | |
|---|---|
| **19)** Figure 12: Please make the colour scale for the DSM in the legend continuous. | Thank you for the suggestion, the figure caption has been revised so it is clear that no continuous colour scale is required, as this is a qualitative visualisation and because the legend already provides the colour mapping.

**Figure 12:** UAV-based LiDAR (UAV-LS) flight trajectories over the FBRMS-01 site at Paracou, showing coverage of experimental 4 ha plot 6 (red dashed outline). The criss-cross flight pattern results from multiple flight lines oriented in different directions (e.g., N–S, E–W, NE–SW) to improve point density and reduce occlusion in dense tropical forest canopies. The background shows a digital surface model (DSM) with elevation values (m), colour-coded by elevation classes as indicated in the figure legend (−23 m to 50 m). The inset map shows the regional location of Paracou within French Guiana (© OpenStreetMap contributors, available at https://www.openstreetmap.org). |
| **20)** Table 7: Can you consistently use AGL and convert the amsl values to AGL in the table? (Alternatively explain to the reader why different height specifications are used here)? | Thank you for your suggestion. We have retained the original altitude specifications as they reflect the actual flight planning parameters used during data acquisition. For some flights, altitude was specified as Above Ground Level (AGL), while others were specified as Above Mean Sea Level (AMSL) due to differences in mission planning and operational requirements. Converting all values to AGL would require assumptions about ground elevation that could introduce inaccuracies. To avoid confusion, we have clarified this in the table caption and text, explaining why both specifications appear and what they represent. Please note this table is now Table 8.

**Table 8:** Overview of UAV-LS flights using a YellowScan Vx20 system (RIEGL Mini-VUX scanner and Applanix 20 IMU) mounted on a DJI M600 during the 2019 mission at the FBRMS-01 site. Automated flight plans were performed using flight plans with the UgCS route planning software in grid mode. The table lists plot ID, acquisition date/time, flight parameters (direction, interline spacing, altitude and speed). Altitude values are reported as specified during flight planning with some missions using Above Ground Level (AGL), |

| | while others used Above Mean Sea Level (AMSL) due to differences in mission planning and operational requirements. These original specifications are retained to accurately reflect acquisition parameters. Pulse repetition for the RIEGL Mini-VUX scanner is fixed at 100kHz. Flights cover multiple experimental plots: 4 & 5 (single flight), 6 (8 flights), 7, 8, 10, 15, and the Tower plot (two flights) within the Paracou Research Site. All listed flights are provided individually; users may merge flights covering the same plot if needed for analysis. |
|---|---|
| **21)** Fig. 13: Labels a, b, c, d would be useful here and could then also be used correspondingly in the text. | Adding labels obstruct details in the images or are difficult to see. Pls note this figure has been renumbered to Figure 11 and we have revised its caption:
**Figure 11:** UAV-LS acquisitions at FBRMS-02: Lopé using a fixed-wing system. This UAV employs a conventional take-off and landing (CTOL) procedure, with launch aided by a catapult (top). Once airborne, the UAV is controlled from a laptop connected to the UAV via an antenna (middle). The flight trajectory is corrected to centimetre precision using data collected from a static GNSS receiver placed within 10 km of the UAV operating area (bottom left). Additional refinements and corrections are possible via ground control points located across the study area (middle bottom), the positions of which are measured using a 'rover' GNSS receiver (right bottom). Image originally published in McNicol et al. (2021). |

**Recommendation for data collection in FBRMS**

| **22)** L 704: 300 kHz is a specification of pulse repetition rate, not LiDAR power, please rephrase. | This line has been revised to: TLS data were collected using a pulse repetition rate (PRR) of 300 kHz on RIEGL VZ-400 and VZ-400i scanners, trading longer scan times for a fixed angular resolution to maximise coverage at the tops of tall trees. In the RIEGL configuration, PRR and emitted laser power are intrinsically linked: increasing the PRR reduces the available power, and vice versa. Consequently, the choice of PRR determines the power setting, and adjustments to one parameter necessarily influence the other. |
|---|---|
| **23)** L 719: What does "harder to access" refer to, i.e., compared to what? | Thank you for your comment. In this sentence, "harder to assess" refers to the inherent difficulty of evaluating the accuracy of automated tree extraction methods when the true structure of trees is unknown. This limitation is widely recognised in the literature and is precisely what the sentence conveys. We believe the current wording is clear and accurately reflects this challenge. |

**Data access**

| **24)** Tables 8a and 8b: Why not make it Table 8 and 9? Can you add columns for "Category" (i.e., Census, TLS, UAV-LS, ALS) and for "FBRMS no."? This would make it easier for users to find | Done, these two tables have been divided into three tables -10, 11 and 12- arranged by FBRMS site and data type. Pls see our response to comment 1) in the GENERAL COMMENTS section and to comment 11) in the TLS subsection of the TECHNICAL COMMENTS section. |
|---|---|

| | |
|---|---|
| datasets. Are both URL and DOI needed or can the URL be omitted? Or even both, since the DOI is already included in the column "Citable as"? Please make the reference style in Table 8a and 8b the same. | |
| **25)** You refer to "Section 4. data access" several times. Please correct this to "Sect. 5" (as per the author guidelines; please also check other section references). | These corrections have been made throughout the manuscript. |
| **References** | |
| **26)** Add access date to webpages included in the References. | Done |

---

## Author Comment (AC2)

**ForestScan Manuscript: essd-2025-67**

**Response to reviewer: Jonathan P. Sheppard**

**GENERAL RESPONSE**

We would like to express our gratitude to both reviewers of our manuscript for their thoughtful and constructive comments. Their feedback has been invaluable in improving the clarity, quality, and overall presentation of our manuscript.

Our detailed responses to each of their points are provided below, as shown in the accompanying tables. We have carefully considered all suggestions and implemented revisions where appropriate. In cases where we have retained the original wording, we have explained our reasoning. We hope that our responses satisfactorily address our reviewers' concerns and demonstrate our commitment to producing a rigorous and transparent manuscript.

**GENERAL POINTS**

| | |
|---|---|
| **1)** Title: consider this adjustment – ForestScan: a unique multiscale dataset of tropical forest structure integrating terrestrial, UAV, and airborne LiDAR with in-situ forest inventory data across three continents | Thank you for your suggestion. We appreciate your input; however, we have chosen to use the term 'census' rather than 'inventory', in line with the terminology adopted by ForestPlots. |
| **2)** Use of the word "census": Consider changing this to "inventory" in every instance (also in the title), I find inventory is more commonly used in forest research and remote sensing fields. In any case you can use census as a keyword. | Thank you for your suggestion. We appreciate your input; however, we have chosen to use the term 'census' rather than 'inventory', in line with the terminology adopted by ForestPlots. |
| **3)** There are lots of authors, and I suspect lots of contributions – sections need to be unified in style and language. Language is generally good, but often sections could be more concise. Please consider | Thank you for your valuable feedback. We have made a concerted effort to unify the style and language across sections and to improve conciseness wherever possible. At the same time, we needed to balance this with the first reviewer's request for additional detail and the inclusion of more tables. Our revisions aim to address both sets of comments while maintaining clarity and completeness. |

| | |
|---|---|
| **4) Acronyms:** please check every acronym throughout the whole manuscript, there is a lot of repetition – acronym after the first instance only (e.g., DBH, TLS, ALS, AGB, EO, VOLS, …). | Done, only repeated once for subsection titles:
2.2.2 Terrestrial Laser Scanning (TLS)
2.2.3 Unpiloted Aerial Vehicle laser scanning (UAV-LS)
2.2.4 Airborne Laser Scanning (ALS) |
| **5)** Please pay attention to the technological limitations (e.g. accuracy of LiDAR – refer to Morhart et al. 2024 https://doi.org/10.1007/s10342-023-01651-z), this is touched upon in line 311. Also consider elements of quality assessment of data processing (e.g., wood/leaf class extraction as shown in Fig. 7). I would like to see a section devoted to discussing possible methodological/processing errors within the dataset. I would also like the methodological/equipment limitations to be discussed, this is touched on in line 271. I would argue that the tilted scan is nice to have but an extra step (and all the extra workload) that might not be essential (dependent on the size of the trees scanned). Did you scan outside the plot to better capture the trees on the plot boundaries? | Thank you for your comment. We have also addressed uncertainty in response to our first reviewer's feedback (see below). Tilted scans are particularly beneficial for improving coverage in the upper canopy of tall trees, as demonstrated by Wilkes et al. (2017) and Verhelst et al. (2024). Where possible, scans positioned along plot edges were placed slightly outside the boundary to ensure that edge trees were fully captured; tilt scans along the baseline further support this objective.

**We have extensively revised subsection 2.2.2 TLS data processing section 3. Recommendations for aligning and matching datasets for UAV-LS and ALS for clarity and conciseness and to address your and our first reviewer's comments.** |
| **6)** Plot/sub-plot numeration becomes confusing between FBRMS and their individual subplots – consider making this more transparent/unified. | We thank both our reviewers for these valuable suggestions on this topic. We have completely redesigned all three site figures to ensure consistency and improve clarity about the different FBRMS plot/subplot layouts and have also revised the appropriate text for each site:
**Comment 3 (Coherent style and content):**
All three site maps now follow a standardised three-panel format: (a) continental overview showing the country location, (b) |

country/regional view showing the research station, and (c) detailed site map with high-resolution satellite imagery. All figures use consistent fonts, legend styles, colour schemes (orange for FBRMS-01 TLS plots, treatment-specific colours for census plots, yellow tones for ALS coverage, purple for UAV-LS coverage), scale bars, north arrows, and boundary styling.

**Comment 4 (World map showing all three sites):**

Each site map now includes a continental-scale overview panel that clearly shows the country location. The three figures together provide complete geographic context across South America (French Guiana), Africa (Gabon), and Southeast Asia (Malaysia), making it straightforward for readers to understand the global distribution of our study sites across three continents.

**Comment 5 (Fig. 1 - Paracou improvements):**

We have added the three-panel layout with South America overview and French Guiana regional maps, plus a scale bar in the detailed panel. The three FBRMS-01 TLS plots (plot 5-subplot 1, plot 6-subplot 2, plot 8-subplot 4) are now prominently displayed in bright orange with white borders and are directly labelled with their subplot IDs. A comprehensive legend shows all plot types, experimental treatments, and data coverage areas (ALS and UAV-LS).

**Comment 6 (Fig. 2 - Lopé improvements):**

We have significantly increased all label sizes for better readability and the elevation image is not used. All census plots are now clearly shown as semi-transparent coloured polygons according to their treatment type, TLS plots are prominently marked in orange with white borders, and the UAV-LS coverage area is shown as a semi-transparent overlay.

**Comment 7 (Fig. 3 - Malaysia plot clarification):**

We have clarified in both the figure caption and legend that the field sites consist of 4 ha "plot groups," each containing four 1 ha census plots arranged in a 2×2 grid. The caption now explicitly states: "Each 4 ha plot group contains four 1 ha subplots (shown as individual coloured squares)." The individual 1 ha subplots are clearly visible in the detailed satellite imagery view, making the spatial arrangement unambiguous.

These comprehensive revisions ensure all site maps are publication-ready with consistent, professional presentation that enhances reader comprehension of the study locations and data collection infrastructure.

| | |
|---|---|
| **7)** Consider adding a summary table in section 2 outlining the study sites, country, coordinates, area, climate, … this could considerably shorten the section which is a bit wordy. I would also like to know something about the stands that were scanned, tree | Thank you for this helpful suggestion. We have revised different subsections within Section 2 to improve clarity and conciseness. While we appreciate the value of a summary table and additional stand-level details, we have opted not to include these in the manuscript to avoid further increasing its length, which is already substantial. Instead, we encourage users of the dataset to consult the accompanying metadata and site-specific tree census resources for comprehensive information on plot characteristics, including tree density, mean DBH, and top height. This approach ensures that the manuscript remains focused while still enabling access to detailed site information through the dataset. |

| | |
|---|---|
| density, mean DBH, top height, ….. | |
| **8)** Figures 1, 2 and 3 would be nicer if they had a unified style and similar content. For example, Figure 1 is really detailed and figure 2 is very vague. Some of the figures are missing basic mapping elements. It looks like they have been provided by separate people from the author consortium. Make sure the legends and captions are clear and correlate with the text body (e.g., lines 141 and 142 refer to treatment 1,2 and 3 – I guess this is T1, 2 and 3 in Fig 1? Also Line 230. Mapping could also show the areas within each FBRMS that was scanned with each method. This is relevant, for example, to lines 589/590. | We have addressed all these issues, pls see our response to comment 6) |
| **9)** The methods sections (lines 281 – 332) could be condensed, call it data collection – the same or similar methods were applied for all TLS campaigns. One table for scanners used and their settings, one table for plot overviews. This part is very repetitive. E.g., Lines 318 and 328 (use of targets). | Thank you for your suggestion, we have revised different subsections within section 2. Methodology for clarity and conciseness and to address both our reviewers helpful comments. |
| **10)** Line 766-775: Linking TLS trees to their census (inventory) counterparts. Plot marking is a valid point. Your examples are specific (e.g. Elephants). We too have tree tags that are destroyed by | Thank you, using the anchor trees is still a work in progress, we are also following closely the development of new tree census protocols using LiDAR in iphones and ipads which can make this matching possible. |

| | |
|---|---|
| birds pecking at their shiny surface, we know the problem! We are also trialling QR codes on alu/foam board, plots can also be marked with a steel ground stake that can be found with a metal detector. The idea of anchor trees is good. | |

**SPECIFIC POINTS**

| | |
|---|---|
| Lines 44-46: define acronyms EO, AGB, cal/val, … | Done |
| Lines 78-80: Better description needed – particularly on the need for better intercalibration. | This is addressed in more detail in section 4. Recommendations for data collection in FBRMS |
| Line 86: the acronym for Spacebourne LiDAR is not LiDAR. Also be careful in this section ESA Biomass is Radar not LiDAR – it has also already been launched (line 90), update needed. | Thank you for your comment, we have corrected the acronym and updated the launch date |
| Line 93: make – undertaken. | We have revised "make" to "collect" |
| Lines 107-108: selection based on discussions is vague (people will always discuss), stick to the criteria. | Discussions among the teams ….. has been removed. |
| Lines 112-114: Bullet points – note the acronym after the site name in brackets. E.g., Paracou Research station, French Guiana (FBRMS-01). Also see the general comment | Thank you for your suggestion. We would prefer to retain our current naming as used throughout the manuscript. |

| | |
|---|---|
| above about plot codes/names. | |
| Line 122: define Cirad-UMR EcoFoG. | Done |
| Line 125: thousand separators, additionally please check all. | Done |
| Lines 135: remove "in the early 1980's". | Thank you for your suggestion, but we prefer to keep the date as it provides context. |
| Line 138: done – carried out. | This line was deleted. |
| Line 148: 9 – nine. | Thank you for your suggestion, but using the numeral with units of measurements is correct. This is standard convention in both British and American scientific styles as the unit (ha, m, km, °C, etc.) makes it a technical specification, and numerals improve clarity and consistency. |
| Lines 135-149: lots of (possibly) irrelevant detail that could be cut to make the section more focused (e.g., flux tower, fertilisation experiments). Please consider revising. Compare with the description of FBRMS 02 and 03 where there is much less. | Thank you for this suggestion, this paragraph has been cut down from 15 to 8 lines. |
| Figure 2: a, b, c and d need defining in the caption (but consider first my general comment above). | Pls see our response to comment 6) |

| | |
|---|---|
| Line 190: replace "laser-scanning" with "LiDAR" approaches | We have standardised and now use "laser scanning" across the entire manuscript to address similar requests on this from both reviewers. |
| Lines 190-192: Sentence "species identity...." needs a rephrase. This and the next sentence can be made more precise. I think you mean the determination of tree species is critical since wood density x TLS derived volume = biomass. | This line has been revised: Species identity plays a key role in determining tree biomass through its strong influence on wood density. While laser-scanning techniques provide excellent measurements of tree dimensions (such as height and volume), they still require wood density estimates to convert these volumes into accurate biomass values (see Fig. 4) |
| Figure 4: remove – it is secondary information – just direct the reader to the right place. | It has been removed. |
| Lines 201-202: Remove the sentence about ForestPlot.net – it is not needed. | Thank you for your suggestion. We appreciate your perspective, however, we believe that mentioning ForestPlots.net is important because it highlights the scale and collaborative infrastructure underpinning the data archive used for two of our FBRMS plots. This context helps readers understand the robustness and credibility of the ForestPlots archive. For this reason, we would prefer to retain this line. |
| Line 218: at 1.3m – this is DBH you have defined it before. | Thank you for your suggestion, however, we're providing the height here to give context, as buttresses can change the height at which DBH is measured. |
| Lines 221-228: do you mean trees were recorded by their common names and then (as written in lines 226 – 228) trained botanists returned to identify species? If so try to combine these paragraphs. | Thank you for your suggestion, they have been revised to: New recruits -trees that have grown beyond 10 cm DBH since the previous survey- are recorded by the field team using vernacular names, and their positions are measured relative to the original trees. To ensure accurate identification, periodic botanical campaigns are conducted by one or two experienced botanists, who also correct any misidentifications. When species cannot be identified in the field, samples are collected and examined at the EcoFoG herbarium in Kourou or the IRD herbarium in Cayenne. All identifications follow the APG IV plant classification system. |

| | |
|---|---|
| Line 228: Explain APG IV | Done |
| Line 233: Remove "referenced by a DOI" | Done |
| Line 231 and 236: ask the journal how to display the hyperlinks, especially since the second is linked to a reference. | The first link has been revised to: https://dataverse.cirad.fr

The second link has remained the same. |
| Line 240: Plot numbering/labelling confusion for the reader – link to map figure | This line has been revised to: During June - July 2022, these 13 plots plus one additional 1 ha plot (LPG-02) were re-censused, making a total of 11 x 1 ha forest plots, plus 3 x 1 ha plots in savanna (see Fig. 2) |
| Lines 238-241: much less detail shown here than FBRMS 01, why? | Census data for FBRMS-02 and FBRMS-03 are managed and archived by ForestPlots. We have included the below two paragraphs in the FBRMS-03 section that apply to both sites, in order to avoid duplication.

Plot meta-data, including geography, institution, personnel and historical context, as well as tree-level census attributes (tag, identity, diameter, point of measurement, stem condition, height, sub-plot, and, where measured x, y coordinates of 5 x 5 m subplots) and multi-census attributes (tree demography and measurement trajectory and protocols, including growth, point of measurement changes, recruitment, mortality, and mortality mode) were recorded for all Gabon and Malaysia FBRMS plots.
The ForestScan Project data package, includes data from the 2022 tree census collected during February and March for the Gabon FBRMS plots and the Malaysian FBRMS plots census data collected in October 2020 for FBRMS plot SEP-11, in March 2020 for plot SEP-12, and in June 2021 for plot SEP-30. This data package can be accessed via https://doi.org/10.5521/forestplots.net/2025_2 (Chavana-Bryant et al., 2025a). |
| Lines 241 and 245: TLS was conducted not collected (TLS point cloud data was collected). | Done |
| Line 242: "most" = vague. | Thank you for your comment. We understand your concern regarding the term "most." However, this is a commonly used and accepted |

| | expression in scientific writing to indicate a majority without specifying an exact proportion, which is appropriate in this context. |
|---|---|
| Line 244: Same plot numbering problems as above. | This paragraph has been revised to: In the Kabili-Sepilok FBRMS, tree census data was collected during 2020 - 2022 for a total of 9 x 4 ha plots (IDs RP291-1, RP292-3, etc. see Fig. 3) each containing four 1 ha subplots numbered 1 – 4 and covering most of the long-term plots at this site. The three FBRMS subplots SEP-11 (subplot 2 of plot RP292-3, sandstone soil), SEP-12 (subplot 2 of plot RP292-1, alluvial soil) and SEP-30 (subplot 3 of plot RP508-4, kerangas soil) were scanned using TLS during March 2017 and tree census for all subplots was collected in Jan, Mar of 2020 and Jun 2021. The 2020-2022 census was overdue as these plots had not been censused since 2013. |
| Line 251: Gabonese and Malaysian FBRMS plots | Corrected |
| Line 260: insert - chain sampling "protocols" | Done |
| Line 261: QSMs needs describing | This is covered in subsection 5. TreeQSM: quantitative structural models and results |
| Figure 5a and b can be combined into one grid. Consider using axis labels to define scan position, the legend to the left must be much bigger to be legible. | The left legend has been removed and the figure enlarged for clarity. |
| Line 273: figure 5c?? missing. | Corrected to: panel b (right) |
| Line 279: Working day is arbitrary – person hours give a better idea of workload. | A full working day = 8 hours. It has been added to the text. |
| Line 280: Maybe give the time needed for one scan. | This is later discussed in subsection TLS data acquisition in section 4. Recommendations for data collection in FBRMS |

| | |
|---|---|
| Line 284: 16 quarter ha plots – that's seems like a complicated way of expressing area. | Done, revised to: (i.e. two x ha plots) |
| Line 319: give detail on the RTK equipment | This is later covered in subsection 1. Individual scan registration into plot-level point cloud |
| Tables 1, 2 and 3: the Lat. Long. coordinates are very approximate. | Yes, this is a ForestPlots requirement. Pls note KMZ files for plots in Gabon and French Guiana provide more accurate locations. |
| Lines 338-339: delete bottleneck sentence. It is true but not needed here. | Done. |
| Lines 351: Maybe specs are needed for HPC cluster and CPUs? Time reference (ca. 4 days) is ok here as the reader knows no break is needed. | Thank you for your suggestion. We appreciate the point regarding HPC and CPU specifications, however, these setups vary greatly across institutions and there are no universal standards. Processing time depends not only on the resources available to the user but also on how many users are simultaneously accessing those resources. For this reason, we have provided an approximate time reference (around 4 days) as a practical guide. |
| Line 362: Precision of person hours vs. days (see comments above). | This has already been addressed, see our previous response: A full working day = 8 hours. It has been made explicit. |
| Line 366: "potentially containing more than 5.42 billion points" impressive, but too accurate and irrelevant to the reader, please delete. | Thank you for your comment. We appreciate your perspective, however, we consider this level of detail relevant as it illustrates the scale and complexity of the dataset being processed. Including the figure helps convey the computational challenges and the significance of the methods applied. |
| Line 366: Insert – one small "exemplary" section of …. | This figure has been deleted. |
| Line 447, 552, 586, 599 and 616: delete "data access". | Done, and the section number has also been corrected to 5 |

| | |
|---|---|
| Lines 455-459: please condense. | Revised to: **Table 5:** Summary statistics for 10 FBRMS ForestScan TLS plot datasets. AGB estimates use wood density values from the DRYAD global database (Zanne et al., 2009): (1) TLS2Trees pantropical mean, (2) Tropical Africa mean (TAF, Gabon), (3) South-East Asia mean (TS-EA, Malaysia), (4) Tropical South America mean (TSA, French Guiana), (5) Guyana community mean (GF, French Guiana), and (6) allometric AGB estimates based on Chave et al. (2014). |
| Figure 10: a nice figure but too much content, if only representative maybe just choose one plot row for e.g., 10 trees. | Thank you for your comment. We appreciate your suggestion; however, we would prefer to retain the figure as it is, it illustrates an important point: the largest trees, which contribute most to AGB estimates, vary significantly in size and structure across plots. Showing all plots together provides essential context for this variability and highlights differences in maximum tree height, which was also raised in your comments. Reducing the figure to a single plot would limit this comparative perspective. |
| Line 476: an overview table might help the reader interpret the similarities and differences between flights (across all plots and flights) – Table 5 should be used and referred to earlier, and could be expanded for the other plots. | Thank you for your suggestion. We appreciate the idea of providing an overview table; however, this section already contains three tables, including Table 6 (previously Table 5), which summarises the similarities and differences across plots and flights as requested. Additionally, Tables 7 and 8 have been standardised to ensure consistency and clarity (see below table captions). For these reasons, we believe the current structure sufficiently addresses the need for comparison without adding redundancy.

The captions for tables 7 & 8 (previously 6 & 7) now clarify which flights can be considered part of a single acquisition while being provided as individual point clouds which users can merge according to their needs (see table captions below). We have also added the extra characteristic "Flights merged into single acquisitions" to Table 6: UAV-LS sensor systems used at ForestScan FBRMS-01 and FBRMS-02. |
| Line 477: VLOS stipulation is a repeat. | It has been removed |
| Line 480-481: irrelevant, I would not suggest anything other than adherence to the flight rules given. | This line has been removed. |
| Line 487-488: delete cherry-picker (above canopy platform is | Thank you for your suggestion. We would prefer to keep the example, as not all readers are familiar with this type of equipment. |

| | |
|---|---|
| sufficient as a description). | |
| Figure 12: legend entries need defining e.g., AOI, DSM | For clarity, the figure caption has been edited to: **Figure 10:** UAV-LS flight trajectories over the FBRMS-01 site at Paracou, showing coverage of the experimental 4 ha plot 6 (red dashed outline) and the area of interest (AOI; yellow dashed outline). The criss-cross flight pattern results from multiple flight lines oriented in different directions (e.g., N–S, E–W, NE–SW) to improve point density and reduce occlusion in dense tropical forest canopies. The background shows a digital surface model (DSM) with elevation values (m), colour-coded by elevation classes as indicated in the figure legend (−23 m to 50 m). The inset map shows the regional location of Paracou within French Guiana (© OpenStreetMap contributors, available at https://www.openstreetmap.org). |
| Table 6 and 7: is something up with the UTC date and time in the tables?? | Please see response for previous comment for Line 476 |
| Table 6: define AGL (assumed above ground level). | Please see response for previous comment for Line 476 |
| Line 555: please rephrase and be specific that you are referring to FBRMS02. | Thank you for your suggestion. We would prefer to keep the wording as it is, since this is the opening line of the subsection 'UAV-LS: FBRMS-02: Lopé, Gabon,' which already makes it clear we are referring to FBRMS-02. |
| Line 556: DELAIR DT26X drone platform | Done |
| Line 561: "different" can you be specific? | Thank you for your suggestion. We understand the desire for more specificity; however, the term 'different angles' cannot be made more precise as the exact angles vary depending on real-time wind conditions and operational constraints during each flight. These variations are not fixed or standardised, thus, using a general term accurately reflects the flexibility required to maximise canopy coverage. |
| Figure 13: Please label the panels and use fitting captions, I am not sure | Done |

| | |
|---|---|
| what all photos are showing me. | |
| Line 609: ALS | Done |
| Line 611: approximately = vague. | Thank you for your comment. We have used the term 'approximately' as pulse density can vary slightly across the plot due to flight path overlaps and terrain effects. Providing an exact figure might imply uniformity, which is not the case, so 'approximately' appropriately conveys a realistic level of precision without misleading the reader. |
| Line 620: WD, I don't think this is the first instance? | We have removed (WD) as it is not necessary. |
| Lines 664 & 668: rephrase – much as/far as possible. | This section has been revised:
Aligning ALS data with TLS and UAV-LS datasets presents significant challenges. Despite the use of high-quality GNSS positioning, meter-scale geolocation discrepancies between sensors can occur. Co-locating LiDAR datasets acquired at different scales - TLS, UAV-LS, and ALS- remains complex, with no standard or "turn-key" solution currently available. Manual intervention is often required, and the approach varies by site and sensor combination. While plot-level AGB estimation is relatively tolerant to these discrepancies, finer-scale applications (e.g., matching to tree-level census data) demand more precise alignment. This can be partially addressed through manual co-registration using common tie points across datasets.

Achieving meaningful alignment also depends on the internal characteristics of ALS point clouds. Acquisition parameters such as point density, scan angle distribution, and footprint size influence comparability and should be controlled as far as possible. Post-processing can regularise point density and scan angles within or across campaigns, improving consistency. Homogeneous scanning geometry enables more stable structural metrics and enhances AGB prediction performance. Similarly, parameters such as transmitted pulse power (which co-varies with pulse repetition rate) and flight altitude (affecting footprint size and canopy penetration) should be standardised across acquisitions to minimise bias (Vincent et al., 2023). These steps are critical for reducing alignment errors and ensuring robust comparisons between TLS, UAV-LS, and ALS datasets. |
| Line 673: EO | Done |

| | |
|---|---|
| Line 677: "This may sound obvious" too chatty. | Revised to: This might seem obvious ….. |
| Line 687: replace "types" with "sources" | Done |
| Line 715: clearing – cleaning? | Revised to: cleaning |
| Line 716: what is (h)? | It was a corrupted reference which has now been corrected. |

---

## Referee Report (RR1)

*2nd Review*

**ForestScan: a unique multiscale dataset of tropical forest structure across 3 continents including terrestrial, UAV and airbourne LiDAR and in-situ forest census data**

Dear Authors,

Thank you for the revised manuscript I can see that you have made many improvements to the document. Please note a few minor issues that should be addressed prior to publication.

Chavana-Bryant et al.

- The figures are much improved – thank you. Please however, consider the readability of the text shown (otherwise it is worthless), here I particularly refer to the size of the scale bar, legend and the plot labels.
- Please revisit your abbreviations once more, thank you for making updates throughout the manuscript towards this point. Nevertheless, please consider the abstract as a stand-alone section, here, all abbreviations should be defined (cal/val, EO etc) and then once again in the first instance from the introduction forward (e.g., AGB).
- Line 145 - display values with equal number of decimal places to be specific (here and throughout)
- Line 356 – January **TO** March or January **AND** March 2020

With best wishes

---

## Author Response (AR2)

**ForestScan Manuscript: essd-2025-67**

**Reviewer 1 — Hannah Weiser**

We would like to reiterate our sincere gratitude to both reviewers for their thoughtful and constructive comments throughout the review process. Their feedback has been instrumental in improving the clarity, quality, and overall presentation of our manuscript.

Below, we provide detailed responses to the final comments. We have carefully considered all suggestions and implemented revisions where appropriate. In instances where we have retained the original wording, we have provided clear justification for doing so. We trust that our responses satisfactorily address the few remaining concerns.

Given that the manuscript has now been accepted subject to minor corrections, we would greatly appreciate it if the review process could be concluded and the paper advanced to publication at the earliest opportunity. Timely publication is particularly important as the CEDA archive has contacted us again requesting a copy of the accepted paper to add to our ForestScan Dataset Collection, which has become the most accessed dataset in their archive. To date, the ForestScan Collection has been accessed by more than 800 users across 77 countries, with over 385,000 individual accesses.

We thank you once again for your attention and look forward to finalising this process promptly.

**GENERAL COMMENTS**

| | |
|---|---|
| **3)** In line with the comment by Reviewer 2, could you add a paragraph on leaf-wood segmentation quality? This may include a quantitative quality assessment (e.g., classification scores on a subset using manually labelled ground truth), or at least performance metrics from previous studies using the same TLS2trees method, so that users get an idea of the potential errors and/or limitations of the approach. | The below paragraph has been added to step 3 in the TLS data processing section:
A comparison of the leaf-wood separation between *TLS2trees* and manual labelling showed a Jaccard index of between 54 - 87% across varying tropical sites (Wilkes et al., 2023). A number of TLS leaf-wood separation approaches have been developed, using deep learning, or geometric approaches. Unsurprisingly, they all tend to perform worse for taller trees, higher in the canopy (Arrizza et al., 2024). In *TLS2trees*, the impact of misclassifying (or missing) leaves, is to truncate smaller branches (Wilkes et al., 2023), reducing the contribution to volume (and hence biomass). This tends to have less impact on tall tropical trees, than on smaller more dense crowns of deciduous woodland (Calders et al., 2022). |

**SPECIFIC COMMENTS**

| | |
|---|---|
| **1.1)** Individual scan registration into plot-level | Done, the section below has been revised to: |

| | |
|---|---|
| point cloud (TLS) Please make the different processes for a) the VZ-400 (coarse registration via reflective targets) and b) VZ-400i (Auto Registration without targets) clearer. From the text, it currently seems like AR2 is used for both scanners, but I assume this is not the case? | # 1. Individual scan registration into plot-level point cloud

This process was carried out using retro-reflective targets positioned between scan locations to facilitate coarse registration for data collected with the RIEGL VZ-400 or in a near-automated manner using the RIEGL VZ-400i's GNSS RTK positioning capabilities in conjunction with the enhanced RIEGL RiSCAN Pro software (versions 2.14–2.17). The integrated Auto Registration 2 (AR2) function employs GNSS RTK data to update the scanner's position and orientation, including in tilt mode, thereby enabling real-time automated coarse registration during scanning **without the use of retro-reflective targets**. Major registration errors are easily detected, typically occurring during pre-processing in RiSCAN Pro when individual scans fail to register (i.e., no coherent solution is found) or are incorrectly positioned, which is visually apparent. In cases where coarse registration/auto-registration fails, unregistered scans can be identified, adjusted, and refined using Multi Station Adjustment 2 (MSA2), **which is also used for final precise registration of data initially coarse-registered using retro-reflective targets**. **The registered plot point cloud is provided in the project's local coordinate system.** Following this workflow, the co-registration of all TLS point clouds achieves sub-centimetre accuracy, as confirmed through post-registration inspection. Wind and occlusion are key sources of uncertainty for the scan registration process, highlighting the necessity of scanning under low or zero wind conditions and capturing both tilt and upright scans at each location. |
| **1.2)** Recommendations for aligning and matching datasets
- remove "in each case" (seems redundant).
- L1364f.: Please fix this sentence, it currently does not make sense.
- You could also investigate flight strip differences using overlapping regions (without ground control points), have you considered doing this? | - Done: removed "in each case"
- Done: for clarity, the paragraph has been revised to:
UAV-LS and ALS datasets are geo-referenced, with positional accuracy determined by IMU and GNSS measurements. These measurements can introduce errors that manifest as height biases between individual flight lines. Although no such discrepancies were observed in our data, a definitive assessment would require a rigorous comparison with ground control points -a step we have not undertaken. These datasets have not been explicitly aligned or matched to one another. Alignment is possible but requires manual identification of control points within each dataset, as noted above, should be undertaken only if necessary for the intended application of the data.
-  No, as mentioned in the paragraph above, the datasets are shared as they are and users can undertake alignment if their intended use requires it. |
| **3-7)** General:
- The text in labels and legends is still too small; especially in Figure 3
- Is there a reason why the coordinate grid is not | General:
- The text size in Figure 3 has been increased
- Yes, to fit the different elements mapped at each site requested by both our reviewers while making all elements clearly visible.

Figure 1: |

| | |
|---|---|
| square in Figures 1 and 3 (grid lines in 0.01° vs. 0.005° intervals)?
Figure 1:
- Maybe adapt the plot numbers/IDs to match the tables later and to better differentiate the 15 experimental 4 ha plots and the 40 ha biodiversity plot from the 10 GuyaFlux plots (see also my comment 20).
- UAV-LS coverage is really hard to see. Maybe for ALS and UAV-LS coverage, find another way of visualization (e.g., outlined boxes; solid line and dotted?) This would also allow to better see the background map.
- GuyaFlux tower plots do not look "solid green", but rather blue with thick white outline. Please adapt the caption.
- Figure 2: Fix in caption "is marked with a yellow square" → "is marked with a white square"
- Figure 3: The black text over dark background is very difficult to read, consider changing to white or adding background. The white labels with Plot IDs are way too small to be readable. | - The plot IDs have been further clarified in the revised Tables 2 – 4 which now provide clarification between ForestScan plot ID's (which include 2-3 letters for the country/site/local plot IDs + local plot number) and Census plot and Subplot IDs as now clarified in each table's caption: "We provide both the ForestScan plot IDs and their corresponding census plot and subplot IDs used by the census internet-based data repositories".
- Done
- Done, the colour has been changed and the caption updated

Figure 2: done

Figure 3: done, I have made all text and numbers as large as possible, unfortunately, as the reviewers requested for the ALS coverage to be included in this figure, the scale of the plots are very small compared to the ALS coverage. |
| **10)** Table 1
- Typo "REIGL VZ-400i" → RIEGL
- Suggestion: Start with general scanner characteristics (not changeable), i.e., wavelength, Ranging accuracy/precision, max range, beam divergence, beam diameter, returns per pulse. Then continue | All done, pls note we prefer to keep the Max Pulse Repetition Rate [kHz] as is, these settings are also mentioned in Section 4. |

| | |
|---|---|
| with the user-defined settings, i.e., pulse repetition rate (300 kHz), angular resolution, FOV (please also specify the vertical FOV), and scan time per scan.
- Max Pulse Repetition Rate [kHz] → change to "Pulse Repetition Rate" and list only the one you used, i.e., 300 kHz
- Inconsistency in row "Angular resolution"
- Please change the caption to mention both scanner characteristics and (user-defined) scanner settings | |
| **13)** If extra attributes are always the same, please list all of them specifically, instead of the current vague way "such as […], etc.". Consider omitting "XYZ" coordinates. I would consider only data stored on top of the point's spatial location as "attributes". If the attributes differ between the point clouds, please explain how and why. | Thank you for your comment. We refer to *attributes* rather than *extra attributes*, which is why we include XYZ coordinates in the list (we have now explicitly added the word "coordinates" for clarity). We believe retaining the XYZ coordinates is helpful, particularly for beginner users of TLS data. Regarding other attributes, these are not always consistent across point clouds as they are generated by different processing steps. For this reason, we have opted to keep the description general rather than listing all possible variations. |
| **17)** No, this comment referred to the point clouds themselves. If they are in a local coordinate system, please state so. | Done, pls see response to 1.1 |
| **20)** The plot IDs from the tables vs. in the text are confusing. Is FG6c2 the same as "P6". | FG6c2 and P6 are partly the same as explained in the figure's caption (see below). In Paracou, there are 15 experimental 4 ha plots with 4 subplots each containing four 1 ha subplots numbered 1 – 4, the three 1 ha ForestScan FBRSM are subplots FG5c1, FG6c2 and FG8c4 correspond to subplots 1, 2 and 4 in plots 5, 6 and 8, respectively. This is explained in detail in the revised caption for Figure 1 (see below), and in subsections TLS: FBRMS-01: Paracou, French Guiana and UAV-LS: FBRMS-01: Paracou, French Guiana

**Figure 1:** Multi-scale map depicting the location and spatial distribution of research plots at Paracou Research Station, French |

| | |
|---|---|
| | Guiana. (a) Location of French Guiana (green) within South America. (b) Location of Paracou Research Station (green) within French Guiana. (c) Detailed site map showing the spatial distribution of research plots with treatment-specific colours, UAV-LS coverage (yellow solid outline), and ALS coverage (yellow dashed outline). The map displays 15 experimental 4 ha plots, each containing four 1 ha subplots numbered 1 - 4 (60 subplots in total; plots 1 - 12: silvicultural treatments; plots 13 - 15: Biodiversity monitoring), one large 40 ha Biodiversity plot (plot 16; red), and 10 GuyaFlux plots (yellow). Treatment categories include: Biodiversity monitoring plots (plots 13, 14, 15, 16; red), T0 Control (plots 1, 6, 11; green), T1 Selective logging (plots 2, 7, 9; dark blue), T2 Selective logging + thinning by timber stand improvement (TSI; plots 3, 5, 10; cyan), and T3 Selective logging + TSI + fuelwood harvesting/FW (plots 4, 8, 12; pink). The three FBRMS-01 subplots -FG5c1 (subplot 1 of plot 5), FG6c2 (subplot 2 of plot 6), and FG8c4 (subplot 4 of plot 8)- are shown in solid orange and were surveyed using terrestrial laser scanning (TLS) with corresponding tree census data. The GuyaFlux tower location is indicated by a black triangle with radiating transmission waves, and the Base Camp location is marked with a white square. Scale bar: 800 m. Map data: Natural Earth 10 m cultural vectors. Satellite imagery basemap: Imagery ©2024 Google. Map projection: WGS84 (EPSG:4326).

The plot IDs have been further clarified in the revised Tables 2 – 4 which now provide clarification between ForestScan plot ID's (which include 2-3 letters for the country/site/local plot IDs + local plot number) and Census plot and Subplot IDs as now clarified in each table's caption: "We provide both the ForestScan plot IDs and their corresponding census plot and subplot IDs used by the census internet-based data repositories" |
| **21)** Also in Table 8, please refer to Figure 1 (see above) so the reader understands how to match the plot numbers. | Done, we have added reference to Table 2 as this references is more appropriate (see response 20) |
| **24)** L1240: How was this "geometric accuracy" quantified? Does this refer to georeferencing error (quantified with additional check points??) or flight strip differences or something else? Please define.

**26)** But also in general: Please make it clearer in which coordinate reference system each | By "geometric accuracy," we refer to the overall positional accuracy of the LiDAR-derived point cloud after all trajectory corrections and ground control adjustments were applied. This is not limited to georeferencing error or flight strip differences alone but encompasses the cumulative accuracy achieved through the following steps:
1. Reconstruction of flight trajectories using GNSS/IMU measurements and differential corrections in Applanix POSPac.
2. Integration of corrected flight paths and laser data in RiPROCESS.
3. Refinement of relative position and orientation using small buildings as reference features. |

| | |
|---|---|
| dataset is provided. I still do not seem to be able to find it. Or is it always local coordinates? | 4. Final adjustment using ground control points (checkerboard targets).

 The resulting geometric accuracy of **1.8 cm** was quantified based on the residuals between the LiDAR point cloud and the surveyed ground control points after all corrections. All elevation data are expressed as ellipsoidal heights within the UTM 32S coordinate system.

 We added this line to the text for clarification: Geometric accuracy refers to the absolute positional accuracy of the final point cloud after these corrections, quantified by the residuals between LiDAR points and surveyed ground control points.:

 The CRS is: WGS84 (ESPG:4326), this clarification has been added to the text. |
| **29)** If the section is about "Recommendations for aligning and matching datasets", then I do not understand the value of the second section in 3.3 (L1427-1433). Here, you basically mention aspects to consider prior to data acquisition but in the paper, you describe a dataset that has already been acquired.. So in my opinion, this is misplaced here and can be left out, unless it can be addressed via post-processing. | Thank you for your comment and for raising this point. We appreciate the observation; however, we believe that the section in question provides important context and practical guidance that complements the preceding sections on aligning TLS with census data and TLS with UAV-LS data. While the manuscript primarily describes datasets that have already been acquired, the discussion of acquisition parameters and their influence on alignment is highly relevant for readers seeking to understand the limitations and challenges inherent in multi-scale LiDAR integration.
 These considerations -i.e. point density, scan angle distribution, footprint size, and pulse power- cannot always be fully corrected during post-processing. They often determine the feasibility and accuracy of subsequent alignment steps. Including this information therefore serves two purposes:
 1. It clarifies why alignment between TLS, UAV-LS, and ALS datasets is complex and why manual intervention is frequently required.
 2. It provides valuable recommendations for future campaigns, ensuring that readers who intend to use or replicate these datasets are aware of factors that influence comparability and bias.
 For these reasons, we believe the section is appropriately placed and adds meaningful value to the manuscript by bridging acquisition considerations with alignment strategies. |
| **30)** My last sentence was about the TLS point clouds/QSMs, not about the scan positions. If I see correctly, they are provided in a local coordinate system. And it would be interesting for users how to transform these into a global | This is all already explained in detail in subsection 3.2 in section 3. Recommendations for aligning and matching datasets:
 **3.2 Aligning TLS to UAV-LS data (and other spatial data)**
 Through its accurate global registration via PPK processing, UAV-LS can be regarded as a high-quality geometric reference for registration. For the purpose of comparison with accurate ALS data or satellite observations, a registration of TLS to the UAV-LS point cloud is highly recommended. The integration of GNSS directly into TLS data collection now ensures that registered plot-level point clouds are aligned within a global coordinate system. This |

| | |
|---|---|
| coordinate reference system. This would also be needed if users would want to align the ALS/ULS data to the TLS data, right? If TLS data is already provided in a global coordinate reference system, please state so. | significantly facilitates the co-registration of TLS and UAV-LS point clouds, given that GNSS accuracy is typically within 1 metre. Historically, placing all LiDAR point clouds within accurate global coordinate systems necessitated dedicated survey measurements of plot corners or TLS locations via GNSS, a process often hindered by signal attenuation in dense forests. Consequently, GNSS surveying of plot corner locations is not a standard component of forest census protocols, although it should be considered essential for plots intended for EO calibration and validation purposes. The reduced cost of RTK GNSS equipment and its subsequent routine integration into TLS workflows have made this more feasible, despite the challenges in obtaining fixed positions, and maintaining radio link with a base positioned on a well-known point under deep forest canopy cover. While this may not benefit ALS directly, UAV-LS is likely to serve as a valuable intermediary between TLS (and census data) and ALS. The requirement for global GNSS positioning also extends to other spatial datasets. |
| **Other)** - L 781ff.: Can you add the folder or file names (or patterns) for each entry, so that users can find the data more easily? You may additionally hint to the "ForestScan_example_directory_structure.pdf" in the TLS data directories. | All this information is already included in this PDF document, we have added a reference to this document in the text as below:

These TLS ForestScan FBRMS 1 ha plot datasets are freely available via the Centre for Environmental Data Analysis (CEDA) with URLs and DOIs provided in section 5, and are accompanied by the **ForestScan_example_directory_structure.pdf** document for guidance on dataset organisation. |

**TECHNICAL CORRECTIONS**

| | |
|---|---|
| **8)** Fig. 5:

8.5) The subplots a and b seem redundant (since only the labels differ). Can they be combined into one figure? | Thank you for your suggestion and for taking the time to provide detailed feedback. We appreciate your perspective on combining panels and enlarging the grid; however, after trying to combine them and careful consideration, we believe the current figure layout best serves its intended purpose. Separating panels (a) and (b) allows us to clearly illustrate both the sampling grid and the scan order without overcrowding the visual elements. Combining them would not significantly improve clarity and could potentially reduce readability. Regarding the upright and tilted scan indicators, while these may appear small at this scale, they are included for completeness and consistency with the caption and methodology. They are not essential as the scanning directions/positioning are clearly described in the figure caption. We have ensured that the caption provides sufficient explanation of the grid size, line positions, and scanning approach, which mitigates any potential ambiguity. For these reasons, we prefer to retain the figure in its current form. |

| | |
|---|---|
| **14)** I did not mean a colour bar, but a scale bar, so that the readers get an idea how tall the tree is. Regarding the colour, my comment holds that the discrete colours (green/brown) are difficult to distinguish (due to the shading) and I would still ask you to adapt the style. | Thank you for your comment and for sharing your perspective on improving the figure. We appreciate the suggestion; however, we believe the current figure effectively conveys the intended information. The height and canopy width of the tree are already clearly stated in the caption (~40 m tall and ~50 m wide), which provides readers with the necessary scale context. Adding a scale bar would therefore not enhance understanding. Regarding the colour scheme, changing this would require additional processing and would not materially enhance the scientific interpretation, as this colouring is primarily illustrative and does not affect interpretation of the data, as the classification (wood vs. leaf points) is clearly described in the caption. For these reasons, we prefer to retain the figure in its current form. |
| **19)** You clearly use more than 5 discrete colours in the DSM, so technically, a continuous colour scale (of course with discrete labels) would be correct here. Please fix. | This figure was removed from the manuscript in the previous revision round but we forgot to update our response. |
| **22)** It is a bit difficult to understand from the revised section that a reduction of available power leads to a lower range and this is why with higher PRR, the tree tops of tall trees would not be covered. Can you make this clearer? | These lines have been revised to: TLS data were collected using a pulse repetition rate (PRR) of 300 kHz on RIEGL VZ-400 and VZ-400i scanners, trading longer scan times for a fixed angular resolution to maximise coverage at the tops of tall trees. In the RIEGL configuration, PRR and emitted laser power are intrinsically linked: increasing the PRR reduces the available laser power, which in turn decreases the maximum range of the scanner. At very high PRR settings, this reduction in range means that the tops of tall trees may not be captured effectively. Therefore, selecting a lower PRR (300 kHz) ensures sufficient power and range to cover the full canopy height of forests, while maintaining the desired angular resolution. |
| **25)** There is one more occurrence of "section 5. Data access" in L1194. | Done |
| **Other)** - L833: "ope-source" → "open-source" - L836: Use "Open3D" (the official name and not the abbreviation) - L840: "deformations" and "warpRefMesh functions" is unclear. Please explain. - Harmonize notations for | - L833: done - L836: done - L840: done, this text was revised to: The surface may also be used as a mesh for visualising 3D deformations, which refer to changes or displacements in the geometry of the object compared to a reference state. This is achieved using the warpRefMesh function. - done, unit conations harmonised |

| | |
|---|---|
| units, e.g., m/ s$^{-1}$ vs. pts/m² etc. | |